# BRAF activation by metabolic stress promotes glycolysis sensitizing *NRAS^Q61*-mutated melanomas to targeted therapy

Kimberley McGrail [1], Paula Granado-Martínez [1], Rosaura Esteve-Puig [1,9], Sara García-Ortega [1], Yuxin Ding [1], Sara Sánchez-Redondo[1,10], Berta Ferrer[1,2], Javier Hernandez-Losa [2], Francesc Canals[3], Anna Manzano [4], Aura Navarro-Sabaté [4], Ramón Bartrons[4], Oscar Yanes [5,6], Mileidys Pérez-Alea[1,11], Eva Muñoz-Couselo[1,7], Vicenç Garcia-Patos [1,8] & Juan A. Recio [1] ✉

*NRAS*-mutated melanoma lacks a specific line of treatment. Metabolic reprogramming is considered a novel target to control cancer; however, *NRAS*-oncogene contribution to this cancer hallmark is mostly unknown. Here, we show that *NRAS^Q61*-mutated melanomas specific metabolic settings mediate cell sensitivity to sorafenib upon metabolic stress. Mechanistically, these cells are dependent on glucose metabolism, in which glucose deprivation promotes a switch from CRAF to BRAF signaling. This scenario contributes to cell survival and sustains glucose metabolism through BRAF-mediated phosphorylation of 6-phosphofructo-2-kinase/fructose-2,6-bisphosphatase-2/3 (PFKFB2/PFKFB3). In turn, this favors the allosteric activation of phosphofructokinase-1 (PFK1), generating a feedback loop that couples glycolytic flux and the RAS signaling pathway. An in vivo treatment of *NRAS^Q61* mutant melanomas, including patient-derived xenografts, with 2-deoxy-D-glucose (2-DG) and sorafenib effectively inhibits tumor growth. Thus, we provide evidence for *NRAS*-oncogene contributions to metabolic rewiring and a proof-of-principle for the treatment of *NRAS^Q61*-mutated melanoma combining metabolic stress (glycolysis inhibitors) and previously approved drugs, such as sorafenib.

As the deadliest form of skin cancer, cutaneous melanoma is a very heterogeneous and complex disease. Despite advances in melanoma treatment during the last decade, long-lasting therapeutic responses are limited, and metastatic cancer remains a difficult type of cancer to treat. In melanoma, RAS pathway activation is an important event in the development and maintenance of the disease. Activating mutations in *BRAF* and *NRAS* are present in 50% and 25% of melanomas, respectively, and NF1, a protein that inhibits the RAS pathway, may also

[1]Biomedical Research in Melanoma-Animal Models and Cancer Laboratory, Vall d'Hebron Research Institute (VHIR), Vall d'Hebron Hospital Barcelona-UAB, Barcelona 08035, Spain. [2]Anatomy Pathology Department, Vall d'Hebron Hospital Barcelona-UAB, Barcelona 08035, Spain. [3]Proteomics Laboratory, Vall d'Hebron Institute of Oncology (VHIO), Barcelona 08035, Spain. [4]Department of Physiological Sciences, University of Barcelona, Bellvitge Biomedical Research Institute, Barcelona, Spain. [5]Universitat Rovira i Virgili, Department of Electronic Engineering, IISPV, Tarragona, Spain. [6]CIBER on Diabetes and Associated Metabolic Diseases (CIBERDEM), Instituto de Salud Carlos III, Madrid, Spain. [7]Clinical Oncology Program, Vall d'Hebron Institute of Oncology (VHIO), Vall d'Hebron Hospital Barcelona-UAB, Barcelona 08035, Spain. [8]Dermatology Department, Vall d'Hebron Hospital Barcelona-UAB, Barcelona 08035, Spain. [9]Present address: MAJ3 Capital S.L, Barcelona 08018, Spain. [10]Present address: Microenvironment & Metastasis Group, Molecular Oncology Program, Spanish National Cancer Research Centre (CNIO), Madrid, Spain. [11]Present address: Advance Biodesign, 69800 Saint-Priest, France. ✉e-mail: juan.recio@vhir.org

be inactivated in another 14% of melanomas[1,2]. Although activation of these molecules within the RAS pathway leads to the activation of ERK1/2, *BRAF*- and *NRAS*-mutated melanomas represent two different clinical and biochemical entities, as they exhibit different signaling patterns and biological responses[3]. In that regard, while BRAF^V600E signaling activates ERK1/2 independently of RAS, mutated NRAS activates ERK1/2 through CRAF[4] and can also signal through the PI3K pathway[5], which is involved in the regulation of metabolism[6].

The discovery of *BRAF^V600E* activating mutations in melanoma led to the development of specific BRAF^V600E inhibitors that currently, in combination with MEK inhibitors, represent the first-line of treatment for *BRAF* mutant melanomas[7,8]. Alongside this, immunotherapies have dramatically improved outcomes[9–11]. However, there is not specific line of treatment for *NRAS* mutant melanomas. The historic difficulties targeting mutated RAS proteins have hampered the treatment of this subset of melanoma tumors, and thus 25% of cutaneous melanomas have no specific line of treatment.

Given the achieved success in targeting sustained proliferative signaling and immune evasion in cancer treatment, there is a strong rationale for considering other hallmarks of cancer, such as reprogramming energy metabolism, as potential targets for therapeutic strategies. It is known that cancer cells show an increased glucose consumption rate, in which metabolic reprogramming represents a clear advantage for proliferation and survival. Increasing evidence shows that oncogene alterations target specific components in metabolic pathways that contribute to this metabolic rewiring. These alterations are essential to support uncontrolled growth, evasion of inhibitory signals for tumor growth, cell migration and metastasis to distant tissues. Regarding RAS pathway activation, it has been demonstrated that BRAF^V600E contributes to melanoma cell survival under oxidative metabolic stress conditions through PCG1α and MITF[12–15] and to metabolic rewiring[16], in which the *BRAF^V600E* mutant melanomas seemed to downregulate the expression of proteins related to fatty acid metabolism[17]. In addition, upon BRAF^V600E activation, RSK1 promotes glycolytic metabolism through the phosphorylation of PFKFB2, probably in a cell type-dependent manner[18]. Furthermore, there is a supporting role for metabolic reprogramming in resistance to RAS pathway inhibition in *BRAF^V600E*-mutated melanoma, in which the expression of glycolytic enzymes was restored upon the ectopic expression of *NRAS^Q61K*, and pharmacological inhibition of glycolysis resensitized these resistant cells to BRAF inhibitors[19]. Despite the advances in *BRAF^V600E*-mutated melanomas, there is considerably less known about the role of mutant *NRAS* in reprogramming metabolism in this disease. RAS oncogenes have been demonstrated to have a significant role in cancer cells, increasing glycolytic metabolism and glucose uptake[20]. In addition, mutated *KRAS* has also been shown to alter several other processes, such as glutamine metabolism, mitochondrial metabolism, redox homeostasis, ribose biosynthesis and the hexosamine biosynthesis pathway[21–25].

Due to the substantial body of evidence supporting the role of metabolic oncogenic reprogramming in melanoma, understanding the contributions of the *NRAS^Q61* oncogene to melanoma metabolism is essential to delineate new therapeutic strategies against a subset of melanomas without a specific line of treatment. Here, we unveil specific metabolic settings of *NRAS^Q61* mutant melanomas and their potential therapeutic impact when repurposing old drugs for melanoma treatments. Our data suggest that *NRAS^Q61* mutant melanomas particularly rely on glucose metabolism. Metabolic profiling shows that upon glucose starvation (GS), *NRAS^Q61* mutant melanomas are less flexible than *BRAF^V600E*-mutant melanomas, using other fuel energy sources. Mechanistically, GS induces the activation of PKA and AKT that promotes the use of BRAF instead of CRAF, resulting in the simultaneous hyperactivation of the RAS pathway and the regulation of key glycolytic enzymes (PFKFB2/PFKFB3), which in turn, foster the activation of PFK1 connecting the RAS pathway and glucose

metabolism. Interestingly, metabolic stress sensitizes *NRAS^Q61* mutant melanomas to sorafenib, establishing a therapeutic strategy for the specific treatment of these tumors.

## Results

### Glucose deprivation promotes ERK1/2 hyperactivation and sensitizes *NRAS^Q61* mutant cells to sorafenib treatment

Previously, we described that *BRAF^V600E* mutant melanoma cells showed limited sensitivity to metabolic stress. This was achieved by uncoupling AMPK and LKB1, which was promoted by the subsequent activation of ERK1/2[26]. This discovery led us and others to investigate the effect of the combination of RAF inhibitors (sorafenib) with AMPK modulators such as metformin[27]. We observed that compared with *BRAF^V600E* mutant cells, the cells harboring *NRAS^Q61* mutations appeared to be more sensitive to the combination (Supplementary Fig. 1a). Thus, we studied the possible relationship between oncogenic alterations in the RAS pathway (*NRAS^Q61* and *BRAF^V600E* mutations) and metabolic stress. As described for some metabolic stressors, we observed that in response to GS, *NRAS^Q61*-mutated melanoma cells, including patient-derived cells, exhibited a quick hyperactivation (within 30 min) of the RAS-ERK1/2 pathway. This occurred much less frequently, or was absent, in the *BRAF^V600E* mutant cells (Fig. 1a). Hyperphosphorylated ERK1/2 accumulated in both the nucleus and the cytoplasm. The rapid hyperactivation of the pathway and the activity or induction of ERK1/2 phosphatases, such as PP2A and DUSP1, eliminated the possibility that the lack of phosphatase activity was the cause for the observed effect (Supplementary Fig. 1b, c). Interestingly, the metabolic stress generated by GS sensitized *NRAS^Q61* mutant cells to the sorafenib treatment (Fig. 1b). To circumscribe the role of RAF proteins in the mechanism, due to the multikinase inhibitor role of sorafenib, we investigated whether other inhibitors targeting the other kinases that were inhibited by sorafenib reproduced the observed result. Sorafenib, regorafenib, salirasib, the MEK1/2 inhibitors U0126 and trametinib, and CCT196969 to some extent, but not sunitinib, axitinib, lenvatinib, tipifarnib, vemurafenib and dabrafenib, were able to consistently inhibit the pathway completely upon glucose deprivation. These data eliminated the possible contribution of upstream receptor tyrosine kinases (PDGFR, cKIT, VEGFR) to the sorafenib-mediated effect (Fig. 1c and Supplementary Fig. 1d) and suggested the involvement of RAF proteins in the mechanism. Interestingly, sorafenib promoted a significant increase in cell death (40-45%, after 1 h. treatment) upon GS in *NRAS^Q61* mutant cells but not in *BRAF^V600E* mutant cells (Fig. 1d and Supplementary Fig. 1e). Thus, these results show evidence supporting the differential molecular settings in *NRAS^Q61*- and *BRAF^V600E*-mutated melanoma cells, directing the distinct molecular response to metabolic stress and suggesting a therapeutic strategy repurposing the use of sorafenib for the treatment of *NRAS^Q61*-mutated melanoma.

### Hyperactivation of ERK1/2 upon metabolic stress is *NRAS* oncogene-dependent and promotes a switch in the use of BRAF instead of CRAF

Since the observed response was restricted to *NRAS^Q61* mutant melanomas, we investigated whether this effect was *NRAS* oncogene-dependent. To dysregulate the pathway, oncogenic *NRAS^Q61* must be recruited first to the membrane and then loaded with GTP. We observed that GS promotes *NRAS^Q61* activation (increase in *NRAS^Q61*-GTP), allowing the downstream activation of the pathway (Fig. 2a). Furthermore, downregulation of *NRAS* by siRNAs impeded the downstream hyperactivation of ERK1/2 in response to GS (Fig. 2b). In agreement with the above results, the inhibition of NRAS signaling by salirasib upon glucose deprivation promoted apoptosis in *NRAS^Q61* mutant cells but not in *BRAF^V600E* mutant cells. Apoptotic cell death was

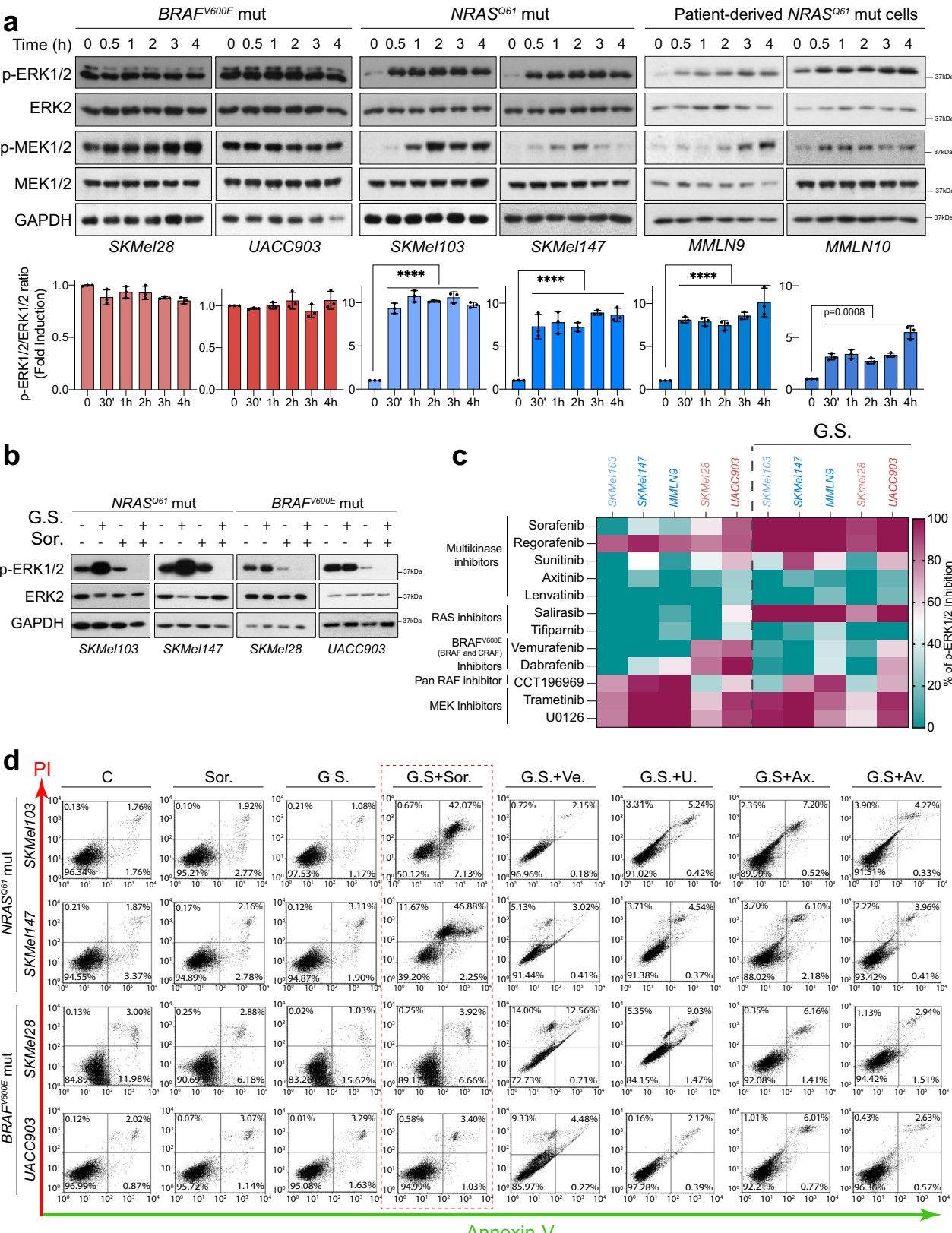

confirmed according to the regulation of cleaved-caspase 6 and PARP1 (Supplementary Fig. 2a).

Contrary to what has been reported using metabolic stressors[28], hyperphosphorylation of ERK1/2 upon GS was concomitant with increased phosphorylation of the CRAF^Ser259 and CRAF^Ser289/296/301 residues, which was related to the inhibition of CRAF activity. This

treatment did not alter the amounts of NRAS molecules that were bound to immunoprecipitated CRAF; however, GS promoted the phosphorylation of BRAF^S445 and the recruitment of BRAF molecules to NRAS, suggesting the concurrent activation of BRAF (Fig. 2c). *NRAS* mutant melanoma cells preferentially use CRAF instead of BRAF (the preferred isoform for signaling in normal melanocytes). Among other

**Fig. 1 | Glucose starvation (GS) induces ERK1/2 hyperactivation and sensitizes *NRAS^Q61* mutant melanomas to sorafenib-mediated cell death. a** *BRAF^V600E* and *NRAS^Q61* mutant melanoma cells were subjected to GS for the indicated time points. Representative images of immunoblotting analysis using the indicated antibodies are shown. The graphs show the quantification of p-ERK1/2 (mean ± SD, $n = 3$ biologically independent samples; ****$p < 0.0001$, unpaired two-sided $t$ test). **b** Representative immunoblots showing the response of *BRAF^V600E* and *NRAS^Q61* mutant melanoma cells to GS for 4 h in the absence or presence of sorafenib. $n = 3$ biologically independent samples. **c** Heatmap representing the quantification of

ERK1/2 inhibition induced by the indicated inhibitors under normal conditions and in response to 4 h of GS in *NRAS^Q61* and *BRAF^V600E* mutant melanoma cells (data source Supplementary Fig. 1d). **d** Cell death detection by flow cytometry analysis of *BRAF^V600E*- and *NRAS^Q61*-mutant melanoma cells stained with propidium iodide (PI) and Annexin-V-GFP. Cells were treated for 4 h with sorafenib, vemurafenib, U0126, Axitinib and Avastin in the presence or absence of glucose (GS) ($n = 2$ biologically independent experiments). C Control, GS glucose starvation; Sor. Sorafenib, Ve. Vemurafenib, U U0126, Ax. Axitinib, Av. Avastin.

mechanisms, this is promoted by the increase in phosphodiesterase activity, which degrades cAMP, thereby preventing the inhibition of CRAF by PKA[4]. It is known that GS promotes the activation of other relevant pathways linked to metabolic regulation such as PKA[29,30] and PI3K-AKT[31,32], both involved in the inactivation of CRAF[33,34]. The PI3K-AKT pathway is also involved in the phosphorylation of BRAF^S445 [34] which contributes to BRAF activation by elevating basal and consequently RAS-stimulated activity[35]. Both the PKA and PI3K-AKT pathways were also activated by GS in *NRAS^Q61* mutant melanoma cells (Supplementary Fig. 2b, c). GS promoted the inactivation of CRAF (phosphorylation at Ser259 and Ser289/296/301)[4,33,36] and phosphorylation of BRAF^S445 through PKA and/or AKT, since AKT and PKA inhibitors abolished the CRAF and BRAF phosphorylation induced by GS (Supplementary Fig. 2c). Treatment with PKA and AKT inhibitors upon GS also promoted the apparent degradation of CRAF. However, it has been demonstrated that dephosphorylation of CRAF^S259 regulates its association with the membrane[37]. Thus, the inhibition of CRAF^S259 phosphorylation promoted CRAF association with the insoluble part of lysates and not its degradation (Supplementary Fig. 2c, d). These results established that BRAF is an important piece of the mechanism. RAF isoforms form homo- and heterodimers to signal, in which the heterodimers signal more potently than the homodimers. We did not observe an increase in CRAF-BRAF heterodimer formation. However, we observed an increase in the immunoprecipitated BRAF amount in response to GS (as early as 15 min after treatment initiation), which was compatible with the formation of BRAF homodimers (Supplementary Fig. 2e). In fact, we observed a switch in the kinase activity of the RAF isoforms in response to glucose deprivation, which was consistent with the observed increased binding of BRAF to NRAS molecules, the increase in BRAF^S445 phosphorylation and homodimer formation. While under basal conditions, CRAF was the kinase signaling, and BRAF was barely active; under metabolic stress, BRAF became active, while CRAF was turned off (Fig. 2d). Notably, the readdition of glucose after 2 h of GS restored the basal amounts of p-ERK1/2, as well as the basal phosphorylation state of CRAF (reflecting its reactivation status) and the original p-BRAF^S445 amounts, confirming the proposed mechanism and its reversibility (Fig. 2e). Moreover, the accumulation of p-CRAF^S338 occurred under normal conditions in the presence of sorafenib, reflecting the use of CRAF under this circumstance. This observation did not occur under metabolic stress (Fig. 2e). Interestingly, GS plus sorafenib treatment led to the disappearance of CRAF from lysates. Again, this observation was due to the inhibition of CRAF^S259 phosphorylation, which promoted the association of CRAF with the non-soluble cell lysate fraction and not with its proteasome-mediated degradation[37] (Supplementary Fig. 2f, g). The described mechanism was genetically validated by knocking down either or both *CRAF* and/or *BRAF* isoforms. While depletion of CRAF did not prevent the hyperactivation of p-ERK1/2 after GS, either BRAF alone or BRAF and CRAF knockdown partially blocked this effect, confirming the participation of BRAF in the proposed mechanism (Fig. 2f, g).

Altogether, these data support the notion that *NRAS^Q61* mutant melanoma cells harbor a specific molecular setting that determines the response to metabolic stress. This mechanism is *NRAS* oncogene-dependent and involves a switch to use BRAF instead of CRAF for signaling.

## *NRAS^Q61* mutant melanomas exhibited differences in mitochondrial capacities and the use of alternative fuel energy sources upon glucose deprivation

To obtain new insights into the metabolic settings that shape the differential response to glucose deprivation, we metabolically profiled *NRAS^Q61* and *BRAF^V600E* mutant cells by using seahorse technology. We measured the oxygen consumption rate (OCR) as a mitochondrial respiration indicator. The percentage of OCR variation at the different conditions in relation to the basal line showed a diminished capacity for spare respiration (Fig. 3a) and ATP-linked respiration production (Fig. 3b) in *NRAS^Q61* mutant melanoma cells with respect to *BRAF^V600E* mutant cells, including two patient-derived cell samples. Moreover, the basal extracellular acidification rate (ECAR) was significantly higher in the assayed *BRAF^V600E* mutant cells than in the *NRAS^Q61* mutant cells (Fig. 3c). Mitochondria can compensate for the lack of a particular energy source by using other pathways to fuel mitochondrial oxidation. Thus, we measured the ability of cells to increase the oxidation of either glucose, glutamine or fatty acids to compensate for the inhibition of the other alternative fuel pathways in basal conditions and upon GS As expected for tumor cells, the cells were heterogeneous for using different fuel energy sources (Supplementary Fig. 3). However, compared to *BRAF^V600E* mutant cells, *NRAS^Q61* mutant melanoma cells were less flexible using alternative fuel energy sources in the absence of glucose (Fig. 3d), suggesting that glucose metabolism was particularly relevant for *NRAS^Q61* mutant melanomas.

## Glucose but not pyruvate or glutamine rescues sorafenib-induced cell death under metabolic stress conditions

We next measured glucose uptake in *NRAS^Q61* and *BRAF^V600E* mutant cell lines in the presence or absence of sorafenib and after GS (1 h). Under basal conditions, glucose uptake was heterogeneous among the assayed cell lines and was more pronounced in SKMel28 cells (consistent with their larger size). However, the addition of sorafenib partially inhibited glucose uptake in *BRAF^V600E* mutant cells, while it increased or had no effect on *NRAS^Q61* mutant cells (Fig. 3e). Furthermore, glucose uptake after 1 h of GS was significantly increased in *NRAS^Q61* compared to *BRAF^V600E* mutant cells (Fig. 3f), supporting the glucose dependency of this subtype of melanomas and their resistance to sorafenib in normal growing conditions.

To further delimitate the steps within glycolysis that contribute to the specific molecular response observed in *NRAS^Q61* mutant melanomas, we investigated whether the addition of increasing amounts of either glucose or pyruvate could reconstitute the molecular response and the natural resistance of *NRAS^Q61* mutant melanomas to sorafenib. Figure 3g shows that in the presence of glutamine, the addition of glucose and not pyruvate reconstituted the resistance of *NRAS^Q61* mutant cells to sorafenib. Consequently, the inhibition of hexokinase with increasing concentrations of 3 bromopyruvate (3-BP) in the presence of sorafenib sensitized cells to the drug, inducing cell death (Fig. 3g). Furthermore, the addition of glucose and not pyruvate in the presence of the drug reconstituted the status of ERK1/2 activation to basal conditions and the constitutive resistance of *NRAS^Q61* mutant cells to sorafenib (Fig. 3h). Altogether, the results confirmed that glucose metabolism was

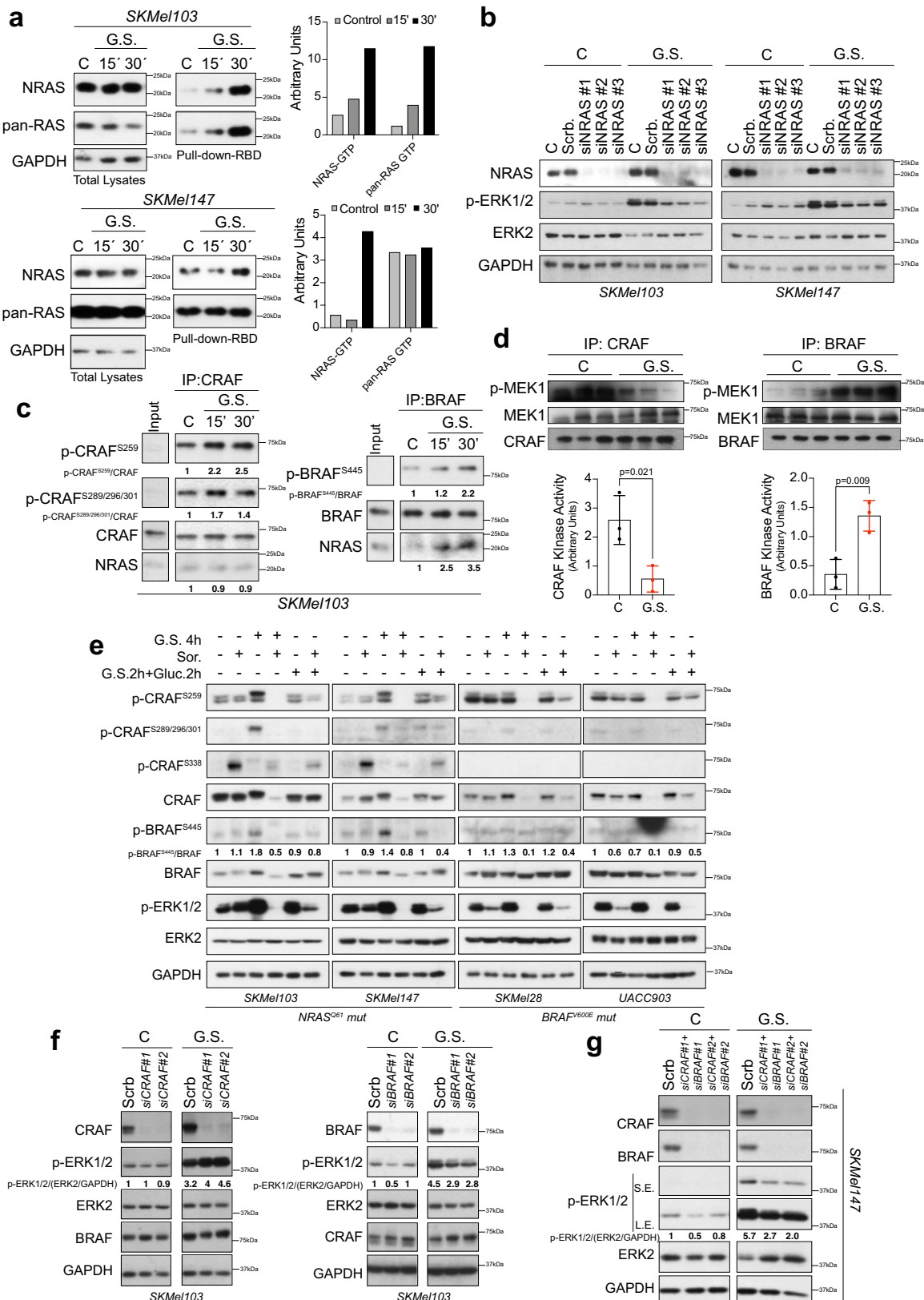

particularly relevant for *NRAS^Q61* mutant melanomas (not to fuel mitochondrial oxidation but probably to generate lactic acid and/or macromolecule intermediates). They also suggest the presence of a molecular sensor between the glycolysis steps and RAS pathway activation.

## *NRAS^Q61* mutant melanomas exhibited an increased glycolytic flux that was sustained under glucose starvation

To investigate how glucose was metabolized in *NRAS^Q61* mutant cells compared to *BRAF^V600E* mutant cells, we labeled melanoma cells with [U-^13C6] glucose for 40 min and traced the carbon distribution into metabolic products by mass spectrometry under basal conditions, in

**Fig. 2 | ERK1/2 activation upon metabolic stress is *NRAS* oncogene-dependent and promotes a switch to use BRAF instead of CRAF. a** Immunoblots showing the amount of activated RAS (RAS-GTP) after GS for 15 and 30 min in *NRAS$^{Q61}$* mutant melanoma cells. The total lysates from the input samples used for the pull-downs with the CRAF binding domain (RBD) are shown. The graphs show the quantification of the indicated pulled down proteins. *n* = 2 independent biological experiments in 2 different cell lines. **b** Immunoblot showing the activation of ERK1/2 upon 4 h of GS in *NRAS* knockdown cells (*NRAS$^{Q61}$* mutant melanoma cells). *n* = 2 independent biological experiments in 2 different cell lines. **c** Representative immunoblots (*n* = 3 independent biological experiments) showing the immunoprecipitated RAF protein isoforms from SKMel103 cells in a time-course manner upon GS The phosphorylation status of the indicated residues and the amount of NRAS present in the immunocomplexes are shown. The numbers indicate the fold change with respect to the control. **d** CRAF and BRAF kinase assays

obtained from SKMel103 cells grown in normal conditions and upon 1 h GS The graphs show the quantification of the assay (mean ± SD, *n* = 3 independent biological experiments, unpaired two-sided *t* test). **e** Representative immunoblots showing the phosphorylation status of CRAF and BRAF in *NRAS$^{Q61}$* and *BRAF$^{V600E}$* mutant melanoma cells upon 4 h GS For recovery, the cells were starved for 2 h. Then, glucose was added for another 2 h. (*n* = 3 independent biological experiments). Numbers indicate fold induction. **f** Representative immunoblots showing ERK1/2 phosphorylation in response to GS in *CRAF* or *BRAF* knockdown SKMel103 cells (*n* = 2 independent biological experiments). The numbers indicate the fold change with respect to the control (scrb). **g** Representative immunoblots showing ERK1/2 phosphorylation in response to GS in *CRAF* and *BRAF* knockdown SKMel147 cells (*n* = 2 independent biological experiments). The numbers indicate the fold change with respect to the control (scrb). C Control, GS glucose starvation, Sor. Sorafenib, scrb scrambled siRNA.

the presence of sorafenib or after 1 h of GS (Fig. 4a, b and Supplementary Fig. 4). In agreement with the above results, the analysis of the identified metabolite isotopologues (M + 0, M + 1, M + 2, and M + 3) and their relative abundance (Supplementary Fig. 4) supported an elevated glycolytic flux in *NRAS$^{Q61}$* compared to *BRAF$^{V600E}$* mutant cells (Fig. 4c). Consistent with the glucose uptake data (Fig. 3e), glycolytic flux inhibition by sorafenib was more pronounced in *BRAF$^{V600E}$* than in *NRAS$^{Q61}$* mutant melanomas (Fig. 4d). Additionally, aspartate was generated from pyruvate through pyruvate carboxylase in both the *NRAS$^{Q61}$* and *BRAF$^{V600E}$* mutant cells. Aspartate production from pyruvate (M + 3) decreased in both melanoma subtypes after GS (Supplementary Fig 4), however, its relative abundance was sustained or increased upon this condition, particularly in *NRAS$^{Q61}$* mutant cells (Fig. 4e). Interestingly, serine (M + 3) was only detected in *NRAS$^{Q61}$* mutant melanomas, in which the production increased after GS (Supplementary Fig. 4). Moreover, the comparison of the relative abundance of pyruvate, lactate, serine, aspartate and most likely alanine (derived from pyruvate) between the basal condition (control) and cells labeled after 1 h in GS suggested that *NRAS$^{Q61}$* and not *BRAF$^{V600E}$* mutant cells tended to sustained and/or increased the glycolytic flux under GS (Fig. 4e and Supplementary Fig. 4). Overall, these data confirm that glucose metabolism is particularly relevant for *NRAS$^{Q61}$* mutant melanomas, generating lactate and macromolecule intermediates (amino acids), even under unfavorable conditions (i.e.: glucose deprivation).

## *NRAS$^{Q61}$*- and *BRAF$^{V600E}$*-mutant melanomas differentially regulate metabolism-related genes in response to metabolic stress, including *PFKFB2*

Although a prompt response to metabolic stress mostly involves the rewiring of metabolic pathways, we investigated whether this response to glucose deprivation was associated with differential gene transcriptional regulation, especially with metabolic genes. Gene ontology analysis of the top 400 differentially expressed genes under basal conditions between the investigated *NRAS$^{Q61}$* and *BRAF$^{V600E}$* mutant melanoma cell lines (Supplementary Data 1) showed the association of these genes with metabolism-related categories, including cellular component organization or biogenesis, metabolic processes and detoxification (Fig. 5a). As expected, there was a limited global transcriptional response to glucose deprivation at early time points, in which the responses of the studied *NRAS$^{Q61}$* mutant cells to gene transcriptional regulation were less than that of *BRAF$^{V600E}$* mutant cells (Supplementary Fig. 5a). Since glucose metabolism appeared to be critical for the *NRAS$^{Q61}$* mutant cell response to metabolic stress, we analyzed the transcriptional regulation (log$_2$FC > 0.265 and <−0.265) of the glycolytic enzymes involved in the conversion from glucose to pyruvate and in the derivation of the intermediates to branching pathways in response to glucose deprivation (Supplementary Fig. 5b and Supplementary Data 2). Overall, *BRAF$^{V600E}$* mutant cells showed a tendency to downregulate multiple glycolytic enzymes. However,

*NRAS$^{Q61}$* mutant melanoma cells displayed a modest upregulation of genes related to the metabolization of glucose to pyruvate and the derivation of glycolysis intermediates to branching pathways (*PGAM2, PFKFB2, G6PD, G6PC2, FBP2, FBP1* and *ENO2*), together with the downregulation of *PKLR, PFKFB4, MPC2, LDHC* and *ALDOB*, which included the genes involved in the use of pyruvate in the mitochondria (OXPHOS) or the conversion of pyruvate to lactate (Fig. 5b and Supplementary Data 2). Regulated genes (log$_2$FC > 0.265 and < −0.265) in *NRAS$^{Q61}$* mutant cells were more thoroughly investigated regarding their expression with respect to the *NRAS* and *BRAF* mutational status of human samples using the TCGA database (Firehose Legacy study, 287 complete samples, Supplementary Fig. 5c). 6-Phosphofructo-2-kinase/fructose-2,6-bisphosphatase 2 (*PFKFB2*) upregulation showed a significant tendency to co-occur with *NRAS* mutations (Fig. 5c). *NRAS* and *PFKFB2* are located on the same chromosome (Chr 1). Most *NRAS$^{Q61}$*-mutated samples either showed a significant positive copy number correlation between both genes or the copy number of *PFKFB2* appeared to be amplified. This observation was not evident in the *BRAF$^{V600E}$* mutant cells (Supplementary Fig. 5d). *PFKFB2* mRNA expression analysis in the CCLE database also showed that the melanoma cell lines are among the tumor types that express higher amounts of *PFKFB2*, which correlated with low DNA methylation detection at the *PFKFB2* promoter (Supplementary Fig. 5e). In agreement with this finding, *NRAS$^{Q61}$* mutant cells, including patient-derived cells, expressed elevated amounts of PFKFB2 protein (Fig. 5d). This result was validated by IHC in an independent subset of human samples showing that *NRAS$^{Q61}$* mutant melanomas expressed significantly higher amounts of PFKFB2 protein than that of the *BRAF$^{V600E}$* mutant melanomas (Fig. 5e). Altogether, the data indicate that *PFKFB2* is overexpressed in *NRAS$^{Q61}$* mutant melanomas and might be involved in their specific response to GS.

## PFKFB2 regulation is sensitized to sorafenib in response to metabolic stress

PFKFB1-4 are involved in both the synthesis and degradation of fructose 2,6-bisphosphate (F2,6-BisP), a metabolite that allosterically affects the activity of phosphofructokinase-1 (PFK1), the major control point in glycolysis. When phosphorylated by several protein kinases, PFKFB2 increases the synthesis of F2,6-BisP (Supplementary Fig. 6a)[38]. PFKFB2 was constitutively phosphorylated at Ser$^{483}$ in both *NRAS$^{Q61}$* and *BRAF$^{V600E}$* mutant cells; however, GS induced the further phosphorylation of this residue in a time-course and cell line-dependent manner (Supplementary Fig. 6b and Fig. 6a). Interestingly, sorafenib and not trametinib completely abolished PFKFB2$^{Ser483}$ phosphorylation, but only upon glucose deprivation (Fig. 6b), suggesting a regulatory link between glycolysis and the RAS-ERK1/2 pathway through the BRAF protein. Although this effect was observed in both *NRAS$^{Q61}$*- and *BRAF$^{V600E}$*-mutant cells, sensitivity was greater in the *NRAS$^{Q61}$*-mutant cells than in their *BRAF$^{V600E}$*-mutant counterparts (Fig. 6c). Although the results were inconsistent and cell line-dependent, we also

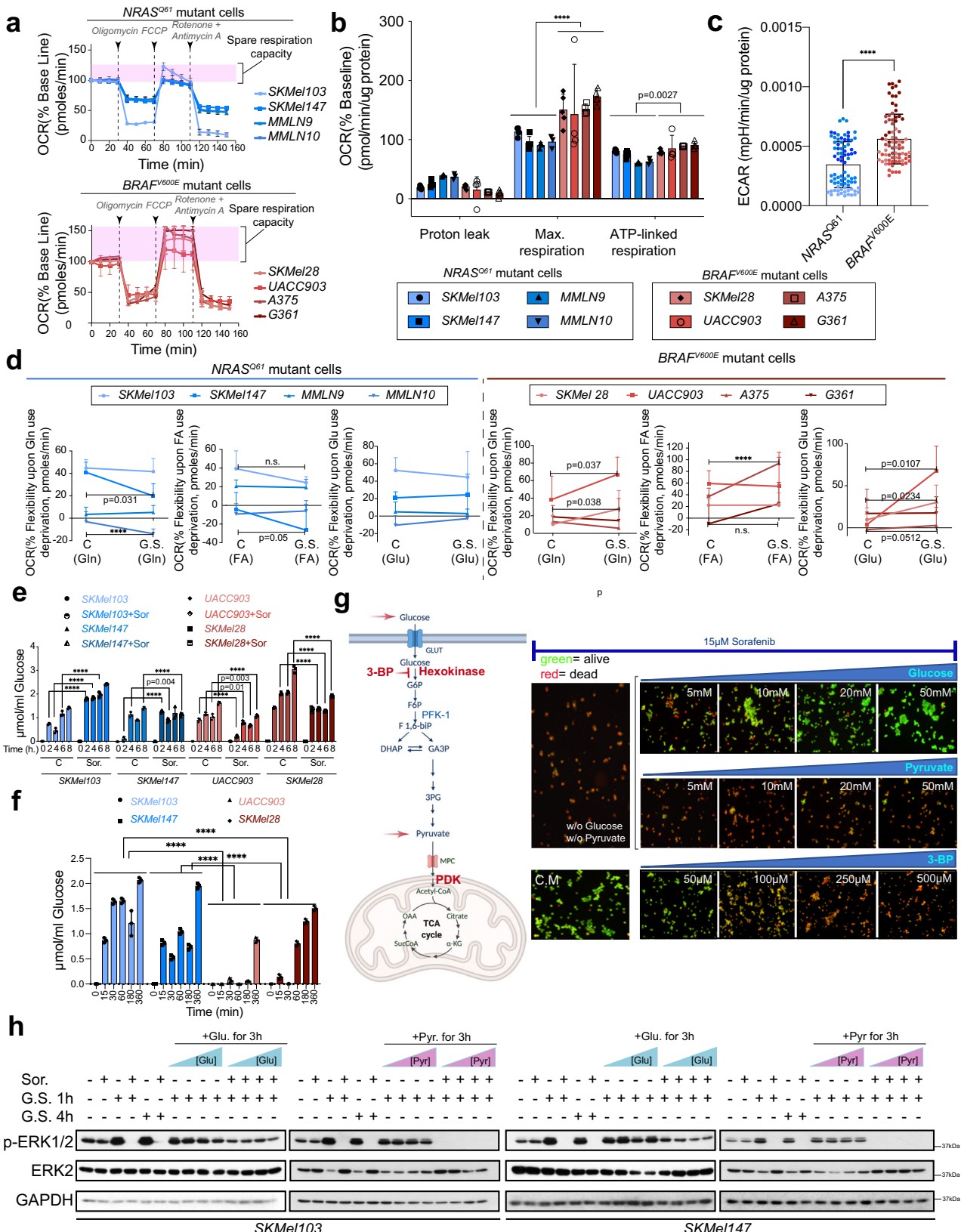

observed the disappearance of PFKFB2 upon GS and sorafenib treatment at late time points. As described for CRAF, this observation was related to the association of nonphosphorylated PFKFB2 with the insoluble fraction of the cell lysates and not to protein degradation (Supplementary Fig. 2g). To circumscribe the participation of RAF (BRAF) proteins in PFKFB2$^{Ser483}$ phosphorylation, we performed both

pharmacologic and genetic experiments. Salirasib (RAS inhibitor) and the RAF inhibitors sorafenib, regorafenib and CCT, but not trametinib or U0126 (MEK inhibitors), inhibited the phosphorylation of PFKFB2$^{Ser483}$ upon GS (Fig. 6d). In agreement with this and with the role of BRAF in the mechanism, knockdown of *NRAS* and *BRAF*, but not *CRAF* or *MEK1/2*, promoted the decrease in PFKFB2$^{Ser483}$

**Fig. 3 | $BRAF^{V600E}$ and $NRAS^{Q61}$ mutant melanoma cells show different metabolic profiles, in which glucose but not pyruvate or glutamine rescues sorafenib-induced cell death under metabolic stress conditions. a** Metabolic profiling of $BRAF^{V600E}$ and $NRAS^{Q61}$ mutant melanoma cells. Five different experiments per condition and per cell line were used to determine the oxygen consumption rate (OCR) (mean ± SD). Arrows in the graph indicate the time points where the drugs were added. **b** Indicated parameters associated with mitochondrial respiration (mean ± SD; $n = 5$ (SKMel103, SKMel147, SKMel28), $n = 4$ (UACC903, A375, G361), $n = 3$ (MMLN9, MMLN10) biologically independent samples; unpaired two-sided $t$ test, ****$p < 0.0001$). **c** Extracellular acidification rate (ECAR). (mean ± SD; $n = 5$ biologically independent samples measured 4 times = 20 measurements per cell line (4 $NRAS^{Q61}$-mutated and 4 $BRAF^{V600E}$-mutated); one-way ANOVA for max respiration; ****$p < 0.0001$, unpaired two-sided $t$ test. **d** Metabolic flexibility of $BRAF^{V600E}$ and $NRAS^{Q61}$ mutant melanoma cells when subjected to GS and mitochondrial use deprivation of fatty acids (FA), glutamine (Gln) or glucose (Glu)) (source data Supplementary Fig. 2b; mean ± SD; $n = 5$ biologically independent samples; ****$p < 0.0001$, unpaired two-sided $t$ test. **e** Graph showing glucose consumption of $BRAF^{V600E}$ and $NRAS^{Q61}$ mutant melanoma cells in the presence and absence of sorafenib (mean ± SD; $n = 3$ independent biological experiments; ****$p < 0.0001$, unpaired two-sided $t$ test. **f** Graph showing glucose consumption of $BRAF^{V600E}$ and $NRAS^{Q61}$ mutant melanoma cells after 1 h of GS (mean ± SD; $n = 3$ independent biological experiments; ****$p < 0.0001$, unpaired two-sided $t$ test. **g** Cell viability assay showing the effects of the addition of glucose or pyruvate under GS conditions or 3-bromopyruvate (3-BP) in complete medium (CM) in the presence of sorafenib in $NRAS^{Q61}$ mutant melanoma cells ($n = 3$ independent biological samples). A schematic cartoon of glycolysis and the involved molecules is shown on the left. **h** Representative immunoblot showing the effects on ERK1/2 activation and sorafenib sensitivity induced by the addition of glucose (Glu.) and pyruvate (Pyr.) for 3 h. after 1 h. of GS in $NRAS^{Q61}$ mutant melanoma cells. The colored triangles (blue and purple) represent increasing concentrations of the indicated compounds (5 mM, 10 mM, 20 mM and 50 mM) ($n = 4$ independent biological experiments). C Control, GS glucose starvation, Sor. Sorafenib.

phosphorylation upon GS, including in patient-derived cells (Fig. 6e and Supplementary Fig. 6c).

Next, we investigated whether PFKFB2 increased phosphorylation upon metabolic stress affected PFK1 activity. While in the absence of glucose $BRAF^{V600E}$ mutant cells showed a 40–50% decrease in PFK1 activity, $NRAS^{Q61}$ mutant melanoma cells tended to sustain PFK1 activity rates (approximately 80%) (Fig. 6f). Moreover, $NRAS^{Q61}$ mutant melanoma cells upheld between 50-80% of F2,6-BisP and 50–100% of F1,6-BisP amounts with respect to basal conditions, while $BRAF^{V600E}$ mutant cells showed a 75% and 90% decrease in F2,6-BisP and F1,6-BisP, respectively, under the same conditions (Fig. 6f and Supplementary Fig. 6d–f). In this way, it is known that high PFK1 activity is associated with the formation of PFK1 tetramers that are associated with F-actin[39], which is induced among others by F2,6-BisP (Supplementary Fig. 6g). Glucose deprivation promoted the colocalization of PFK1 to F-actin in $NRAS^{Q61}$ but not in $BRAF^{V600E}$ mutant melanoma cells (Fig. 6g), suggesting that PFKFB2-increased kinase activity was associated with simultaneous PFK1 activation in an attempt to maintain the glycolytic pathway. The resulting PFK1 activity leads to a sustained production of fructose 1,6-bisphosphate (F1,6-BisP) (Supplementary Fig. 6e). It is known that F1,6-BisP couples glycolytic flux to activate RAS through SOS1/2 in a conserved mechanism from yeast to mammals[40]. Indeed, F1,6-BisP promoted the phosphorylation of ERK1/2 in both low glucose and GS conditions (Fig. 6h). This activation occurred at earlier time points (Fig. 6h and Supplementary Fig. 6h) and was abolished in SOS1/2 knockdown cells (Fig. 6i). Altogether, these data suggest the existence of a feedback loop upon metabolic stress in $NRAS^{Q61}$ mutant melanomas between the RAS pathway and the glycolytic route, connecting BRAF, PFKFB2, PFK1, SOS1/2 and NRAS.

### PFKFB2 heterodimerizes with PFKFB3 and becomes phosphorylated by BRAF, mediating the response to metabolic stress

The above findings link the attempt to sustain glucose metabolism by PFK1 activation, increasing the amount of F1,6-BisP with the activation of the RAS-ERK1/2 pathway. Additionally, PFKFB2 phosphorylation is sensitized to sorafenib upon metabolic stress. Since PFKFB2 activity is mainly governed by posttranslational modifications, we aimed to investigate the protein interaction network, as well as possible relevant and new PFKFB2 phosphorylated residues, by analyzing PFKFB2 immunocomplexes under different conditions by mass spectrometry. In silico analysis of putative phosphorylation sites in the PFKFB2 sequence that were identified previously described phosphorylated residues and the kinases responsible for these modifications. Additionally, RAF proteins were identified as putative kinases for residues S466 and S483, and as previously suggested, T468 and S493 were identified as possible ERK targets[41] (Supplementary Fig. 7a). To isolate the PFKFB2-protein complexes, we used SKMel103- and UACC903-infected cells with a His-tagged-

PFKFB2 inducible expression lentiviral construct. The cells were subjected to GS in the presence and absence of sorafenib (Supplementary Fig. 7b–d). Mass spectrometry analysis of the PFKFB2 protein complexes identified the presence of proteins that are known to modify or bind to PFKFB2, such as AMPK, 14-3-3β and HSP90. Moreover, ERK2, PFKFB3 and ACTB were identified as new partners that increased their presence in the complexes upon metabolic stress (Fig. 7a; Supplementary Fig. 7e and Supplementary Data 3). Analysis of PFKFB2-associated proteins by western blotting not only confirmed the presence of increased amounts of 14-3-3β and p-ERK1/2 in response to GS but also showed constitutive binding to PFKFB3 (Fig. 7b). Interestingly, BRAF was found only in $NRAS^{Q61}$ mutant cells and was stabilized within the complexes by sorafenib, as seen with RAF heterodimer formation (Fig. 7b). PFKFB2 phospho-peptide analysis and fragmentation allowed us to identify phosphorylated residues S466, T468, S483, S486 and S493. These modifications were detected as a single phosphorylation site and as double phosphorylated peptides S483 + S493 (Supplementary Fig. 8a, b). Semiquantitative analysis of the phosphorylated residues with respect to the unphosphorylated peptides according to the treatment demonstrated the induced phosphorylation of residues S466, T468, S483 and S493 in response to GS Moreover, upon metabolic stress, phosphorylation of these residues, including S483 and S493, was sensitive to sorafenib to a different extent and was more pronounced in $NRAS^{Q61}$ than in $BRAF^{V600E}$ mutant cells (Fig. 7c). Immunoprecipitation of either BRAF or p-BRAF$^{S445}$ in His-tagged PFKFB2-expressing cells confirmed the presence of PFKFB2 in the immune complexes, that was increased upon GS in the case of p-BRAF$^{S445}$ (Fig. 6d). This binding was better observed in sorafenib-treated cells with the stabilized complexes in both parental and His-tagged PFKFB2-expressing cells (Supplementary Fig. 8c). This interaction was confirmed in vitro using recombinant proteins (Fig. 7e). In vitro kinase assays using recombinant proteins confirmed that BRAF, and to some extent ARAF but not CRAF, phosphorylated PFKFB2$^{S483}$; however, none of them phosphorylated PFKFB2$^{S466}$ (Fig. 7f). PFKFB1–4 display distinct properties, including tissue expression profiles, the ratio of their kinase/phosphatase activities, and their response to protein kinases, hormonal and growth factor signals[42,43]. The intracellular steady-state concentration of F2,6-BisP is controlled by a family of homodimeric and bifunctional enzymes PFKFB, in which PFKFB3 has been described to have a 710/1 kinase/phosphatase activity ratio[44], favoring the generation of F2,6-BP and PFK1 activation. Reciprocal and consecutive immunoprecipitation of His-PFKFB2 and Flag-PFKFB3 complexes in $NRAS^{Q61}$ mutant cells confirmed the direct or indirect binding of PFKFB2 and PFKFB3, suggesting the formation of heterodimers or heterotetramers (Supplementary Fig. 9a, b and Fig. 7g). The direct binding between PFKFB2 and PFKFB3 and their

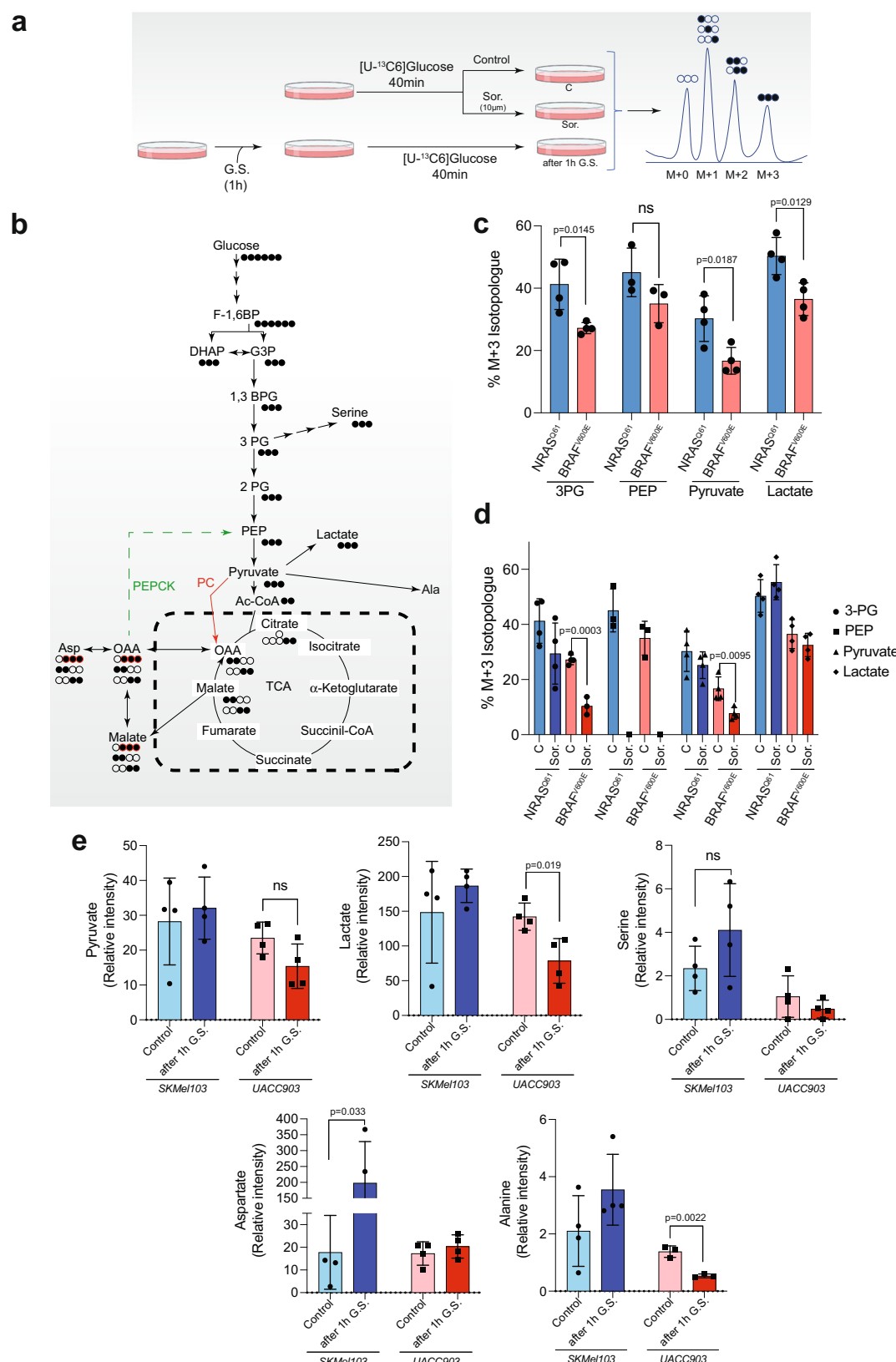

interaction with BRAF was confirmed in vitro using recombinant proteins (Fig. 7h). While PFKFB3 is known to be overexpressed in tumor cells[45], it was not expressed in normal melanocytes, and it tended to be upregulated only in *NRAS^Q61*-mutated patient-derived cells (Fig. 7i). This piece of data was validated in an independent set of human samples (Fig. 7j). To confirm the participation of PFKFB2

or PFKFB3 in glucose deprivation-mediated ERK1/2 hyperphosphorylation, we knocked down either *PFKFB2* or *PFKFB3* by siRNA technology. *PFKFB2* and *PFKFB3* knockdown almost completely abolished p-ERK1/2 hyperactivation in *NRAS^Q61* mutant cells upon GS, while it had no effect in *BRAF^V600E* mutant cells (Fig. 7k). Altogether, these data confirmed the contribution of PFKFB2/PFKFB3 to

**Fig. 4 | *NRAS^Q61* mutant melanoma cells remodel glucose into macromolecules and exhibit a higher glycolytic flux than that of the *BRAF^V600E* mutant cells resistant to sorafenib treatment. a** Scheme showing the labeling of cells with [U-$^{13}$C6 glucose and the posterior identification of the possible intermediate isotopologues by mass spectrometry. **b** Glycolytic pathway and tricarboxylic acid cycle (TCA) diagram showing the distribution of the labeled carbons from [U-$^{13}$C6] glucose into the possible intermediate isotopologues. **c** Graph showing the amount of the indicated glycolytic intermediate M + 3 isotopologs in *BRAF^V600E-* and *NRAS^Q61-* mutant melanoma cells under normal growth conditions. Bars represent the mean ± SD (n = 4 biologically independent samples; unpaired two-sided *t* test, ns = not significant). **d** Graph showing the amount of the indicated M + 3 isotopologues glycolytic intermediates in sorafenib-treated and nontreated *BRAF^V600E* and *NRAS^Q61* mutant melanoma cells. Bars represent the mean ± SD (n = 4 biologically independent samples; unpaired two-sided *t* test). **e** Graphs showing the relative intensity of the indicated molecules under basal conditions and after 1 h of GS in *BRAF^V600E-* and *NRAS^Q61-*mutant melanoma cells. Bars represent the mean ± SD (n = 4 biologically independent samples; unpaired two-sided *t* test, ns not significant). C Control; GS glucose starvation; Sor. Sorafenib.

the survival response to metabolic stress in *NRAS^Q61* mutant melanomas, revealing a link between the RAS and glycolytic pathways, which implies the regulation of PFKFB2 by BRAF and probably ERK proteins.

## The combination of the glycolysis inhibitor 2-DG and sorafenib is effective against *NRAS^Q61* mutant melanomas

The described mechanism suggests an effective and expeditious therapeutic strategy to combat *NRAS^Q61* mutant melanomas by combining metabolic stress and sorafenib (an FDA-approved drug). To translate the metabolic stress to an in vivo setting, we used 2-DG, a glucose analog that cannot undergo glycolysis. Furthermore, 2-DG has already been tested in clinical trials, it was well tolerated and showed no significant effects on tumor growth as a single agent[46]. As reported for other cell systems[28], treating *NRAS^Q61* mutant melanoma cells with 2-DG not only induced ERK1/2 phosphorylation but also PFKFB2 phosphorylation, which was abolished upon sorafenib treatment (Fig. 8a). As expected, the combination of 2-DG with sorafenib resulted in AIF-dependent necroptosis (Fig. 8b–d). Moreover, the combination of both compounds showed a significant additive effect, reducing tumor growth in vivo in an *NRAS^Q61* mutant melanoma model and not in a *BRAF^V600E* mutated xenograft (Fig. 8e). Importantly, these results were also observed in an *NRAS^Q61*-mutated patient-derived xenograft (PDX), in which two out of five tumors totally regressed. Immunohistochemical analysis of samples showed an increased amount of phosphorylated ERK1/2 in *NRAS^Q61*-mutated melanoma upon 2-DG treatment, together with areas that stained positive for cleaved-caspase 3 surrounding necrotic tissue in tumors treated with the combination, in which intact cells stained negative for p-ERK1/2 (Fig. 8e). These results support the in vivo efficacy of the combination of 2-DG and sorafenib for *NRAS^Q61* mutant melanomas, repurposing the use of molecules that are well tolerated for melanoma treatment.

## Discussion

Although *NRAS* was the first oncogene identified in melanoma[47], targeting RAS-mutated tumors has been very difficult thus far. In this matter, the links between RAS signaling and altered cellular metabolism are of particular interest due to the potential use of RAS-related vulnerabilities to treat RAS-driven cancers. *NRAS*-mutated melanoma is distinct from *BRAF*-mutated melanoma in clinical presentation and prognostic features. Here, we show that *NRAS^Q61*-mutated melanomas display distinct metabolic settings that determine the molecular response to glucose deprivation. Under this condition, *NRAS^Q61*mutant tumors modify the use of RAF isoforms, establishing a link between the RAS pathway and glucose metabolism in an attempt to sustain glycolysis through the regulation of key metabolic enzymes (PFKFB2/ PFKFB3). As a result, the hyperactivation of the RAS pathway occurs through a feedback loop, which mediates the initial survival of *NRAS^Q61*mutant tumors under glucose restrictions. Importantly, this condition generates a vulnerable situation in which these tumors become sensitive to sorafenib treatment.

The activation of ERK2 by ROS generation upon extended periods of GS (9–24 h), or the use of metabolic stressors has been documented[28,48]. Our results show that *NRAS^Q61* mutant melanomas, including patient-derived cells, specifically respond to GS by rapidly

hyperactivating the RAS-ERK1/2 pathway (within 30 min.). This observation suggested the existence of specific metabolic settings in this type of tumor related to glucose metabolism and connected to the RAS pathway. Interestingly, this response to GS also generates a vulnerable situation, making these tumors sensitive to an old drug, such as sorafenib. Why sorafenib is the only drug that can generate a significant amount of apoptosis compared to that of the other molecules that inhibit some of the components of the pathway is still under investigation, but it might be related to the class Type II inhibitor in which sorafenib and regorafenib belong. Moreover, these data also discard the participation of upstream receptor tyrosine kinases (PDGF, VEGF, cKIT) in the activation of ERK1/2 and focus the sensitization to sorafenib on RAF proteins. Contrary to previous reports using some metabolic stressors[28], the mechanism seems to be *NRAS^Q61* oncogene-dependent. In this matter, metabolic stress promoted the activation of RAS (NRAS-GTP) and knocking down *NRAS* abolished the effects of GS on ERK1/2 activation. Furthermore, the inhibition of farnesyl-transferase activity upon GS led to the apoptosis of *NRAS^Q61* mutant cells, supporting a protective role for the oncogene in the survival response to metabolic stress. *NRAS^Q61* mutant melanomas preferentially use CRAF to signal downstream pathways. This is promoted by the disruption of CRAF inhibitory mechanisms, such as cAMP signaling[4]. In addition to PKA, AKT is also able to inhibit CRAF by phosphorylating Ser259[36], and both AKT and PKA become activated in *NRAS^Q61* mutant cells in response to GS, resulting in CRAF^Ser259 [33], CRAF^Ser289/296/301 and BRAF^S445 phosphorylation, the consequent inhibition of CRAF and the increased activity of BRAF[4,33–35]. These results support the specific contribution of the oncogene to the response, including the switch in the RAF isoform use (from CRAF to BRAF), a feature that can only be observed in *NRAS^Q61* mutant melanomas. This hypothesis was supported by changes in the phosphorylation codes of both CRAF and BRAF that were linked to the recruitment of BRAF (and not CRAF) to NRAS molecules and the functional consequences in the formation of BRAF homodimers, as well as in the kinase activity of the proteins. Importantly, these changes were reversed by the addition of glucose, confirming the mechanism itself and its reversibility. These results highlight specific *NRAS^Q61* oncogene-dependent features related to glucose restrictions, raising questions related to the metabolic settings of these tumors in contrast to *BRAF^V600E*-mutated melanomas. In relation to this, RAS-driven cancer cells, mostly referred to *KRAS*-mutant tumors, appear to function optimally when the nutrient supply is favorable but undergo rapid bioenergetic collapse when starved of glucose or glutamine because their demands for energy cannot be met in the absence of sustained glycolysis or glutaminolysis[49]. In our study, even though *NRAS^Q61* mutant cells seemed to have an intact mitochondrial network, they showed a diminished spare capacity of mitochondrial respiration and ECAR compared to that of *BRAF^V600E* mutant cells. It is becoming clear that tumors with similar genetic alterations show different metabolic profiles according to their tissue origin[50], also suggesting the importance of the tumor microenvironment in the regulation of cancer cell metabolism. *NRAS^Q61* mutant melanomas appeared to be heterogeneous regarding the preferential fuel energy source at basal conditions;

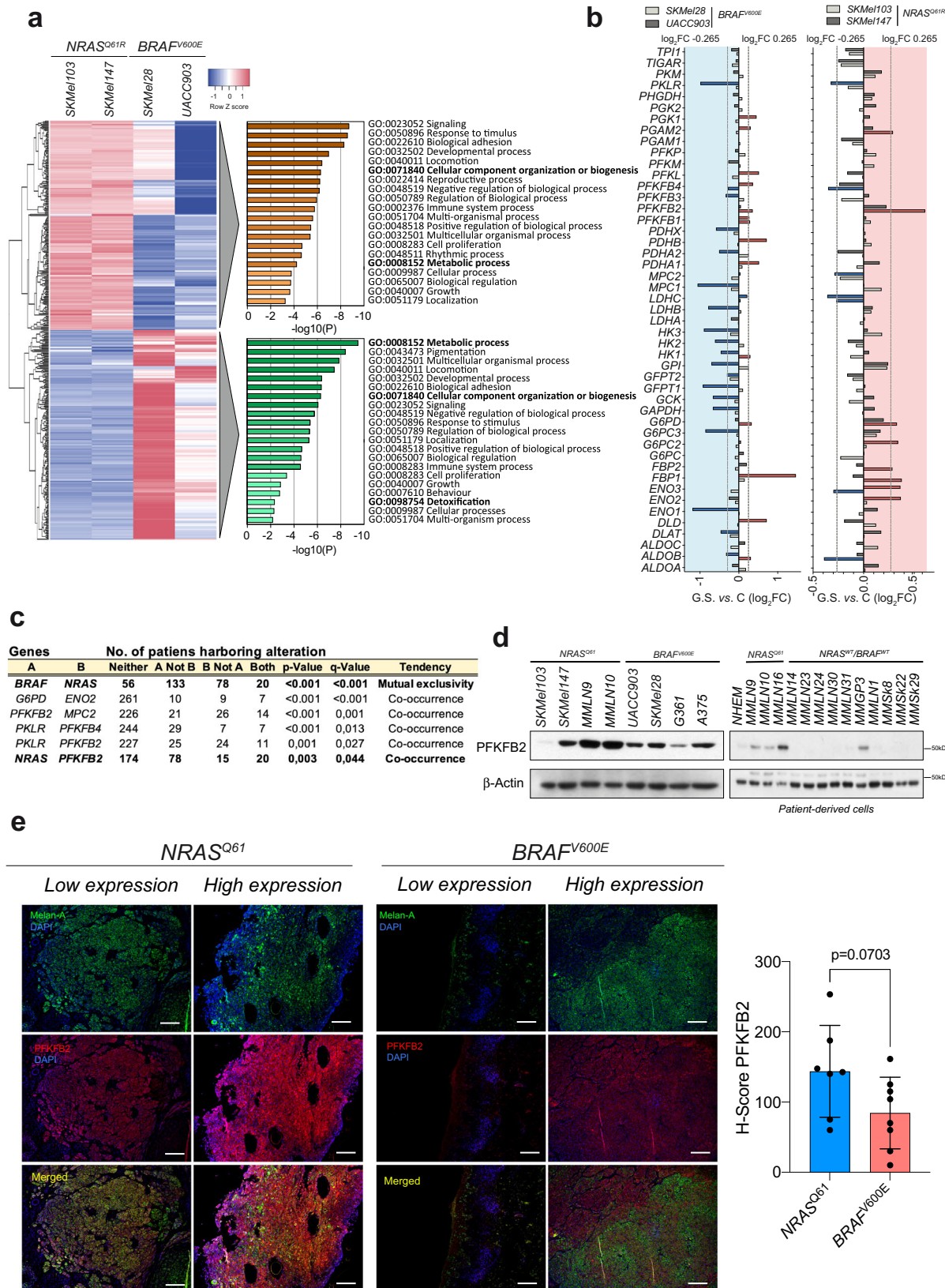

however, their flexibility was also less than that of *BRAF^V600E* mutant melanomas that use alternative mitochondria fuels under GS, suggesting that *NRAS^Q61*-mutated tumors are particularly dependent on glucose metabolism, most likely to generate ATP, lactate, metabolic intermediates and precursors required to sustain cell growth and proliferation. Indeed, the extinction of *KRAS^G12D* in a murine pancreatic cancer mouse model resulted in decreased glucose uptake and lactate production[25]. In relation to this, isotope labeling tracing experiments confirmed the elevated glycolytic flux in *NRAS^Q61* mutant cells, the resistance of this metabolic pathway to sorafenib and the diversion of glucose to macromolecule precursors. Furthermore, the addition of glucose and not pyruvate or

**Fig. 5 | *PFKFB2* upregulation preferentially co-occurs with *NRAS^Q61*-mutated melanomas. a** Heatmap showing the unbiased hierarchical clustering of the top 400 genes differentially expressed (log₂FC > 0.265) under basal conditions in *BRAF^V600E*- and *NRAS^Q61*-mutant melanoma cells. On the right, the graphs show the enriched terms across the input gene lists, colored by p-value (hypergeometric test; https://metascape.org/). *n* = 3 independent biological experiments per cell line. **b** Graph showing the log₂FC variations of glucose metabolism-related genes when subjected to GS for 1 h. in *BRAF^V600E*- and *NRAS^Q61*- mutant melanoma cells. The red bars indicate upregulated genes (log₂FC > 0.265), the blue bars indicate down-regulated genes (log₂FC < −0.265). **c** Table showing the tendency of regulated genes in (b) (log₂FC > 0.265) to either co-occur or be mutually exclusive. Data obtained from TCGA database (Firehose Legacy study, 287 human samples; for *p* value Fisher exact test, for the q-value it's a Benjamini–Hochberg FDR correction procedure, http://www.cbioportal.org). **d** Immunoblot showing the amount of PFKFB2 expressed in melanoma cell lines, including patient-derived cells (*n* = 2 independent biological experiments). **e** Immunofluorescence showing the expression of PFKFB2 in human melanoma samples. Bars represent 400 μm. (*n* = 7 *NRAS^Q61*-mutated and *n* = 8 *BRAF^V600E*-mutated). The graph shows the H-score of the evaluated samples (mean ± SD; *p* value, unpaired two-sided *t* test). C Control; GS glucose starvation.

glutamine upon metabolic stress restored the activation levels of the RAS-ERK1/2 pathway and resistance to sorafenib, supporting the dependency of these cells on glucose metabolism and confirming the existence of a link between glycolysis and the RAS pathway.

As expected, a prompt response to metabolic stress involves minimal variations in metabolic gene transcriptional regulation, where transcriptional and transcript cycling reflects, rather than drives, metabolic and biosynthetic changes. Nevertheless, *PFKFB2* was modestly upregulated in response to glucose deprivation in *NRAS^Q61* mutant cells, revealing an interesting observation in which *NRAS^Q61* mutations tend to significantly co-occur with *PFKFB2* upregulation. The PFKFB family comprises four isoforms (PFKFB1-4) of bifunctional enzymes that control the amounts of F2,6-BisP[51]. The amino acids located near the N- and C-terminal ends of the protein isoenzymes are responsible for its posttranslational regulation, as they can be phosphorylated by different protein kinases[51]. Our results strongly suggest the involvement of RAF proteins in the mechanism, excluding the participation of other kinases, such as p90RSK, which are described elsewhere to regulate PFKFB2 in *BRAF^V600E* mutant cells[18]. The phosphorylation of PFKFB2^S483 by BRAF not only represents a novel link between the RAS pathway and the regulation of glycolysis and survival but also raises interesting discussions. It seems that PFKFB2^S466 phosphorylation is responsible for the increase in *Vmax*, whereas both PFKFB2^S466 and PFKFB2^S483 phosphorylation are necessary to decrease the *km* for F6-P[38]. Thus, BRAF could increase the affinity of PFKFB2 for F6-P upon GS This in turn promotes the production of F2,6-BisP, which allosterically regulates PFK1, leading to the production of F1,6-BisP. Then, in addition to the contribution of other RTKs under G. S or low glucose conditions, F1,6-BisP promoted loading with GTP of the *NRAS^Q61* oncogene through SOS1/2, which again activated BRAF and increased p-ERK1/2 signaling. By doing this, *NRAS^Q61* mutant melanomas ensure the necessary functioning of glycolysis, a distinctive feature of RAS-mutated tumors, and survival under stress conditions (Fig. 9). An interesting question is whether the regulation of PFKFB2 is restricted to *BRAF* wild type under an *NRAS^Q61* mutated context or it is extensive to *KRAS*-mutated tumors. In addition, whether BRAF^V600E can directly modify PFKFB2 must be investigated; however, it seems that BRAF^V600E does this through p90RSK[18]. Our results raise another issue as follows: PFKFB family members have been described to function as homodimers in normal cells. The data suggest the formation of PFKFB2/PFKFB3 heterodimers or heterotetramers, as is the case for PFK1[39]. This configuration would be particularly interesting for cells that maintain glycolysis at all costs, considering that PFKFB3 has a kinase/phosphatase activity ratio of 701/1[44]. This would favor the production of F2,6-BisP, which controls the activity of PFK1 and assures the production of ATP, metabolic intermediates and reduction power. Whether the presence of these heterodimers/heterotetramers is restricted to tumor cells is being investigated; however, normal melanocytes do not express PFKFB3.

The discovered mechanism offers new possible therapeutic strategies by either targeting PFKFB isoforms or combining inhibitors of glucose metabolism (intake or metabolization) with previously approved drugs, such as sorafenib and/or regorafenib. As proof of principle, we show that mimicking GS by the administration of 2-DG in combination with sorafenib is effective against *NRAS^Q61*-mutated melanomas, promoting cell death.

In summary, we show that the *NRAS^Q61* oncogene provides specific metabolic settings that determine the response to metabolic stress, which includes the generation of a feedback loop involving the activation of BRAF and crosstalk with PFKFB2 to control glycolytic flux. In addition, our results suggest the existence of molecular configurations in glycolytic key regulatory enzymes that would be beneficial for cancer cells, particularly for *RAS*-mutated tumors. Finally, this mechanism establishes a condition to confront *NRAS^Q61*-mutated melanomas by repurposing the use of drugs that are approved for human use.

## Methods

The research in this manuscript complies with all relevant ethical regulations. All research protocols has been approved by the Ehtics and Institutional Animal Care and Use Committees of Vall d´Hebron Institute of Research and Hospital.

### Reagents

Sorafenib (used at 15 μM unless otherwise indicated) was obtained from Santa Cruz Biotechnology (Santa Cruz, CA, USA). Trametinib 100 nM, regorafenib 10 μM, salirasib 25 μM, tipifarnib 50 nM, dabrafenib 10 μM, vemurafenib 5 μM, axitinib 10 μM, avastin 250 μg/ml, lenvatinib 40 μM, sunitinib 10 μM, CCT196969 10 μM, MK2206 5 μM, H-89 10 μM, necrostatin1 10 μM, zVAD-FMK 20 μM and MG132 1 μM, were obtained from Selleckchem (Houston, TX, USA) and used at the indicated concentrations. U0126 (used at 10 μM) was obtained from Cell Signaling (Danvers, MA, USA). Sodium pyruvate was purchased from Biowest (Riverside, MO, USA). Metformin 250 μM, D-glucose, D-glucose 13C6, fructose 1,6-bisphosphate (F1,6-BisP) and 2-deoxy-D-glucose (2-DG) were obtained from Sigma (St Louis, MO, USA). 3-Bromopyruvate (3-BP) was obtained from Millipore (Burlington, MA, USA). Primary antibodies against p-ERK1/2^T202/Y204 (1:1000), p-BRAF^S445 (1:500), p-CRAF^S259 (1:1000), p-CRAF^S338 (1:1000), p-CRAF^S289/S296/S301 (1:1000), MEK1/2 (1:1000), p-MEK1/2^Ser217/222 (1:1000), cleaved caspase 3 (1:400), cyclin D1 (1:1000 Western Blot (WB), 1:50 Immunohistochemistry (IHC)), PFKFB2 (1:1000 WB, 1:100 Immunofluorescence (IF), 1:50 Immunoprecipitation (IP)), p-PFKFB2^S483 (1:1000), p-AKT^S483 (1:1000), AKT (1:1000), p-PKA substrate (1:1000), Caspase 6 (1:1000) and GST (1:5000) were purchased from Cell Signaling. 14-3-3ß (1:500), ERK2 (1:1000), DUSP1 (1:200), PFK1 (1:100), Lamin A/C (1:1000), AIF (1:1000), PARP1 (1:1000), SOS1 (1:100), NRAS (1:100), ARAF (1:1000), BRAF (1:1000 WB, 1:50 IP) and CRAF (1:1000 WB, 1:50 IP) were purchased from SCBT. Melan-A (1:100) was obtained from Thermo Fisher Scientific (Whalhman, MA, USA). PFKFB3 (1:1000) WB, 1:100 IF, 1:50 IP) and GAPDH (1:10000) were obtained from Proteintech (Proteintech, Rosemont, IL, USA). Ki67 (1:200) was purchased from Abcam (Abcam, Cambridge, UK). panRAS (1:1000) and p-PFKFB2^S466 (1:1000) were obtained from Millipore. SOS2 (1:1000) was purchased from

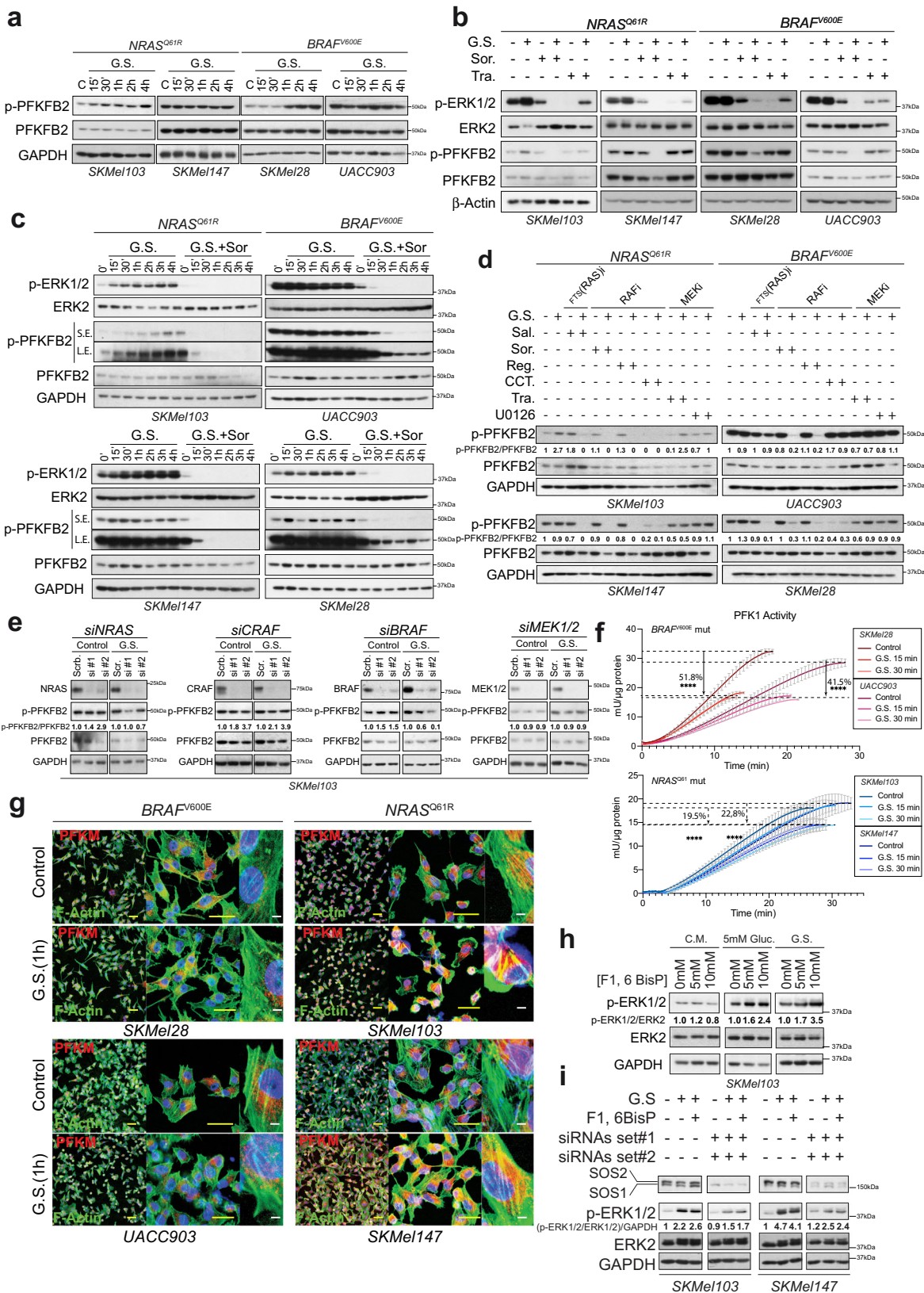

R&D systems (Minneapolis, MN, USA). β-Actin (1:10000) was obtained from Sigma. Horseradish peroxidase and secondary fluorescent antibodies were obtained from GE Healthcare (Little Calfont, UK) and Thermo Fisher Scientific, respectively. Alexa Fluor™ 488 phalloidin (F-Actin) was purchased from Thermo Fisher Scientific.

## Cell culture

SKMel103 (CVCL_6069) and SKMel147 (CVCL_3876) cells were obtained from M. Soengas (CNIO, Madrid, Spain). UACC903 (CVCL_4052) cells were a gift from J. Trent (Tgen, Phoenix, AZ, USA). SKMel28 (HTB-72), A375 (CRL-1619) and G361 (CRL-1424) were

**Fig. 6 | Metabolic stress promotes the regulation and sensitization of glycolytic enzymes to sorafenib, leading to RAS-ERK1/2 pathway activation.**
**a** Representative immunoblot showing the time-dependent regulation of PFKFB2 phosphorylation (S483) upon GS in $NRAS^{Q61}$- and $BRAF^{V600E}$-mutant melanoma cells ($n = 3$ independent biological experiments). **b** Representative immunoblot showing the sensitization of PFKFB2 phosphorylation to sorafenib and not trametinib upon 4 h GS in $BRAF^{V600E}$- and $NRAS^{Q61}$- mutant melanoma cells ($n = 3$ biologically independent samples). **c** Representative immunoblot showing PFKFB2 phosphorylation regulation (S483) over time in response to GS and sorafenib treatment in $NRAS^{Q61}$- and $BRAF^{V600E}$-mutant melanoma cells ($n = 2$ biologically independent samples). **d** Representative immunoblot showing PFKFB2 phosphorylation regulation (S483) in response to GS and RAS/ERK1/2 pathway inhibition conditions (Ras inhibitor: Salirasib; RAF inhibitors: sorafenib, regorafenib, and CCT196969; MEK inhibitors: trametinib and U0126) in $NRAS^{Q61}$ and $BRAF^{V600E}$ mutant melanoma cells ($n = 3$ biologically independent experiments). Numbers show the fold induction of the indicated ratio. **e** Representative immunoblots showing PFKFB2 phosphorylation regulation (S483) in response to 4 h. GS in $NRAS$, $CRAF$, $BRAF$ and $MEK1/2$ SKMel103

knockdown cells ($n = 3$ biologically independent experiments). Numbers show the fold induction of the indicated ratio. **f** PFK1 enzymatic activity assay in $BRAF^{V600E}$- and $NRAS^{Q61}$- mutant melanoma cells after GS for 15 and 30 min (mean ± SD, $n = 5$ biologically independent samples; unpaired two-sided $t$ test, ****$p < 0.0001$). **g** Immunofluorescence of PFK1 (PFKM) and F-actin (Alexa Fluor-488-phalloidin) in $BRAF^{V600E}$- and $NRAS^{Q61}$- mutant melanoma cells under 1 h GS Bars represent 500 µm (yellow) and 25 µm and 5 µm (white). **h** Immunoblot showing the activation of the RAS-ERK1/2 pathway in SKMel103 cells by the addition of fructose 1,6 bisphosphate (F1,6BisP) to complete medium (CM), under low glucose conditions (5 mM) and GS ($n = 3$ biologically independent samples). The numbers indicate the fold change with respect to the control. **i** Immunoblot showing the activation of the RAS-ERK1/2 pathway by the addition of fructose 1,6 bisphosphate (F1,6BisP) under GS in $SOS1/2$ knockdown $NRAS^{Q61}$-mutant melanoma cells ($n = 3$ biologically independent samples). The numbers indicate the fold change with respect to the control. C Control, GS glucose starvation, Sor. Sorafenib, Tra. Trametinib, Sal. Salirasib, Reg. Regorafenib, CCT. CCt196969, S.E. short exposure, L.E. long exposure.

purchased from the American Type Culture Collection (ATCC, Manassas, VA, USA). NHEM (C-12400) was purchased from PromoCell (Heidelberg, Germany) and cultured following manufacturer's recommendations. Patient-derived cell lines, including MMLN1, MMLN9, MMLN10, MMLN14, MMLN16, MMLN23, MMLN24, MMLN30, MMLN31, MMGP3, MMSK8, MMSK22 and MMSK29, were derived from patients after tumor surgery. All samples were obtained and used upon the informed consent of the patients and the Vall d'Hebron Hospital Ethical and Clinical Research Committee (CEIC) approval (PR(AG)115/2013). Patient-derived cells were cultured in Dulbecco's modified Eagle's medium (DMEM) (Biowest) supplemented with 20% fetal bovine serum (FBS) (Gibco, Waltham, MA, USA), 2 mM L-glutamine (Gibco), 100 U/ml penicillin, 100 µg/ml streptomycin (Gibco) and 5 µg/ml Plasmocin (InvivoGen, Toulouse, France). SKMel103, SKMel147, A375 and G361 cells were grown in DMEM. UACC903 cells were cultured in RPMI 1640 media (Biowest). SKMel28 cells were cultured in Eagle's Minimum Essential Media (EMEM) (ATCC). In all cases, the media was supplemented with 10% FBS, 2 mM L-glutamine, 100 U/ml penicillin, 100 µg/m streptomycin and 5 µg/ml Plasmocin. All cells were maintained at 37 °C in a 5% $CO_2$ incubator. For the experiments, the cells were cultured in DMEM without glucose (Gibco).

### Constructs
The *pLenti-rtTA2-IRES-H2B-GFP* doxycycline-inducible plasmid was obtained from *S. Tenbaum, HG Palmer's Laboratory* (Vall d´Hebron Institute of Oncology, VHIO). Human *PFKFB2* and *PFKFB3* sequences were obtained from *pCR4-TOPO-PFKFB2* (MHS6278-202856883) and *pBluescriptR-PFKFB3* (MHS6278-202808447) (Horizon Discovery, Cambridge, UK) to generate *pLenti-rtTA2-His-PFKFB2-IRES-GFP* and *pLenti-rtTA2-Flag-PFKFB3-IRES-GFP*, respectively. During the cloning process, a His-tag was added to the N-terminal end of the PFKFB2 protein and a Flag-tag was added to the N-terminal end of the PFKFB3 protein by PCR. The obtained constructs were validated by restriction analysis and sequencing.

### siRNA assays
*SOS1, SOS2, NRAS, BRAF, CRAF, MEK1, MEK2, PFKFB2* and *PFKFB3* siRNAs were purchased from Thermo Fisher Scientific. siRNAs were transfected using Lipofectamine® RNAiMAX (Thermo Fisher Scientific) following the manufacturer's recommendations. Cell treatments were performed at least 60 h after transfection. The siRNA sequences were as follows: *NRAS* siRNA#1: CAAGUGUGAUUUGCCAACAAGGACA; siRNA#2: CAAGAGUUACGGGAUUCCAUUCAU; siRNA#3: AGUCAUUU GCGGAUAUUAACCUCUCUA; *BRAF* siRNA#1: AAGUGGCAUGGUGAUG UGGCA; mission esiRNA#2; EHU127401 CRAF siRNA#1: UGUCCAC AUGGUCAGCACCtt; mission esiRNA#2: EHU139661; *MEK1* siRNA#1: GGAACCAGAUCAUAAGGGAtt siRNA#2: UGUUCAGUCUGGAAUUUC

Att; *MEK2* siRNA#1: GAUCAGCAUUUGCAUGGAAtt, siRNA#2: GAAC UUGACGAGCAGCAGAtt; *SOS1* siRNA#1: GUCGCAUAGUAUACAtt; siRNA#2: GCAUAUCUUUUGCAACGAAtt; *SOS2* siRNA#1: GGAAGUAU UAGGGUACAAAtt; siRNA#2: GUAGUACUCUAGAUCGAAUtt; *PFKFB2* siRNA#1: CCAAGAAACUAACACGCUA; siRNA#2: GCCUCGCACCAUUU ACCUUtt; PFKFB3 siRNA#1: GAGGAUCAGUUGCUAUGAAtt; siRNA#2:C CAUCUACCUGAACGUGGAtt.

### RAS activation assay
GTP-RAS was quantified using the RAS Activation Assay Kit (Millipore) following the manufacturer's recommendations. After the isolation of GTP-RAS, complexes were resolved by SDS−PAGE, transferred to a PVDF membrane and probed with anti-NRAS and anti-pan-RAS antibodies.

### Colony formation assay
Five hundred cells per well were seeded in six-well plates and incubated for three weeks. Media was replaced every 48 h. Colonies were stained with crystal violet.

### Cell viability
Acridine orange and ethidium bromide were purchased from Sigma. The cells were resuspended in 50 µl PBS. The cell suspension was stained with 50 µl of staining solution (5 µg/ml AO and 3 µg/ml EB in PBS) and placed on a microscope slide with a glass coverslip. Live cells appeared uniformly green, while apoptotic cells also incorporated ethidium bromide and therefore stained red.

### Apoptosis assays
Apoptosis after the cell treatments was analyzed using an Annexin-V-FITC apoptosis kit (BioVision, Milpitas, CA, USA) following the manufacturer's instructions. Data was analyze using FlowJo v10.

### Immunoblots
The cells were lysed in RIPA lysis buffer, and equal amounts of protein were subjected to SDS−PAGE and transferred to a PVDF membrane. Immunoblots were performed as previously described[52,53]. Nuclear and cytoplasmic protein lysates were obtained using the NE-PER Extraction Kit (Thermo Fisher Scientific) following the manufacturer's recommendations. For the obtention of nonsoluble proteins, samples were lysed in 80 mM TRIS (pH 6.8), 10% glycerol, 100 mM DTT and 2% SDS. Quantifications of bands were performed using ImageJ 1.53a.

### Immunoprecipitation and His-Flag pulldown
Immunoprecipitation was performed as previously described[54]. Pull down of the His-tagged PFKFB2 proteins was performed using

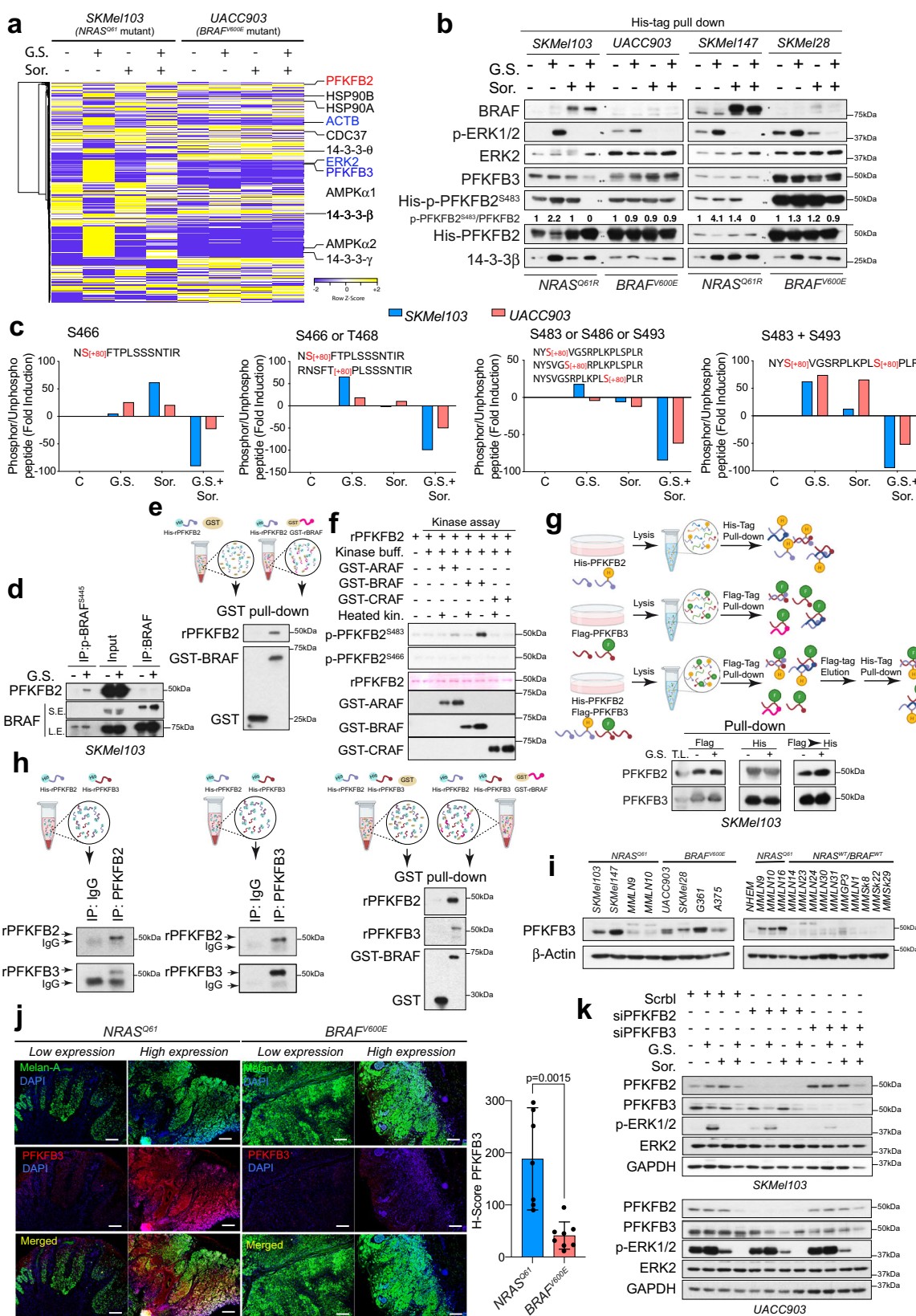

cobalt-based immobilized metal affinity chromatography (IMAC) Dynabeads™ following the manufacturer's directions (Thermo Fisher Scientific). Eluted His-tagged proteins were used for immunoblotting or mass spectrometry. To pull down Flag-tagged proteins, an anti-Flag M2 Affinity Gel Kit (Thermo Scientific) was used following the manufacturer´s instructions. After binding and

washing, bound proteins were eluted by competition with Flag-peptide. For consecutive pull-down assays, the cells were infected with both constructs (*pLenti-rtTA2-His-PFKFB2-IRES-GFP* and *pLenti-rtTA2-Flag-PFKFB3-IRES-GFP*), and a His-tag pull down procedure was performed with Flag-tagged eluted proteins. Complexes were analyzed by western blotting.

**Fig. 7 | Protein interaction network analysis links PFKFB2/3 to the RAS-ERK1/2 pathway. a** Spectral counts) of the identified proteins in the PFKFB2 protein complexes upon the indicated treatments in *NRAS^Q61*- and *BRAF^V600E*-mutant melanoma cells (*n* = 3 biologically independent experiments). Proteins already described to interact with PFKFB2 (black); new proteins described in this work (blue). **b** Representative immunoblot showing the amounts of the indicated proteins after isolation of His-tagged PFKFB2 complexes in *NRAS^Q61* and *BRAF^V600E* mutant melanoma cells. Numbers show the fold induction of the indicated ratio (*n* = 3 biologically independent experiments). **c** Graphs showing the fold change differences with respect to the control (spectral peak areas) of the indicated phosphorylated/unphosphorylated ratio peptides in *NRAS^Q61* and *BRAF^V600E* mutant melanoma cells upon the indicated treatments. Each condition represents three biologically independent experiments (*n* = 3) that were pooled before the mass spectrometry analysis. **d** Immunoblot showing the binding of PFKFB2 to BRAF and p-BRAF^S445 in *NRAS^Q61* mutant melanoma cells under 1 h GS; *n* = 3 independent experiments per treatment. **e** Representative in vitro interaction assay of GST-BRAF and recombinant PFKFB2 (rPFKFB2). Immunoblots show the amounts of the identified proteins after GST pull-down (*n* = 3 independent samples). GST protein was used as a

control. **f** Representative in vitro kinase assay (*n* = 3 independent samples) using rPFKFB2, GST-ARAF, GST-BRAF, and GST-CRAF. Ponceau S staining shows rPFKFB2. **g** Immunoblot showing the amount of the identified proteins after isolation of the protein complexes as indicated in the scheme. Total lysate 5% (T.L.), 4 h GS, and pull-down (P.D.) (*n* = 2 independent biological experiments). **h** Representative in vitro interaction assay of rPFKFB2, rPFKFB3 and GST-BRAF. The immunoblots show the amount of the identified proteins after the PFKFB2/PFKFB3 immunoprecipitation or GST pull-down (*n* = 3). Isotype IgG and GST were used as controls. **i** Immunoblot showing the amount of PFKFB3 expressed in melanoma cell lines, including patient-derived cell lines. (*n* = 2 independent biological experiments). **j** Immunofluorescence showing the expression of PFKFB3 in human melanoma samples. Bars represent 400 μm. (*n* = 7 *NRAS^Q61*-mutated and *n* = 8 *BRAF^V600E*-mutated). The graph shows the H-score of the evaluated samples (mean ± SD; *p* value, unpaired two-sided *t* test). **k** Immunoblot showing the activation of ERK1/2 upon 4 h GS and/or sorafenib treatment in *PFKFB2* or *PFKFB3* knockdown *NRAS^Q61* and *BRAF^V600E* mutant melanoma cells (*n* = 2 biologically independent experiments). C Control, GS glucose starvation; Sor. Sorafenib; Scrbl. scrambled siRNA.

## PP2A phosphatase activity

Phosphatase activity was established using the Ser/Thr Phosphatase Assay Kit 1 (K-R-pT-I-R-R) (Millipore) following the manufacturer's instructions.

## In vitro kinase assay

RAF kinase assays were performed with immunoprecipitated BRAF or CRAF as described in Andreu-Perez et al.[54]. For PFKFB2 phosphorylation analysis, GST-ARAF (Abcam), GST-BRAF (Millipore), GST-CRAF (Genescript, Piscataway, NJ, USA) and PFKFB2 (Origene, Rockville, MD, USA) recombinant proteins were incubated in 20 mM MOPS pH 7.2, 25 mM ß-glycerol phosphate, 5 mM EGTA, 1 mM sodium orthovanadate, 1 mM dithiothreitol, 250 μM ATP and 37.5 mM magnesium chloride for 30 min at 30 °C. The products of the assay were subjected to SDS–PAGE and transferred to a PVDF membrane. PFKFB2 phosphorylation was evaluated by using specific anti-p-PFKFB2^S466 and anti-p-PFKFB2^S483 antibodies.

## Metabolic profiling

Mitochondrial function and glycolytic function were assessed using Seahorse technology (Seahorse XF Cell Mito stress kit, Agilent, Santa Clara, CA, USA). Briefly, SKMel103, SKMel147, MMLN9, MMLN10, SKMel28, UACC903, A375 and G361 cells were cultured on Seahorse XFe-24 plates (Agilent) at a density of 80,000 cells per well. On the day of metabolic flux analysis, the cells were changed to unbuffered DMEM (DMEM base medium supplemented with 10 mM glucose, 1 mM sodium pyruvate, 2 mM glutamine, pH 7.4) and incubated at 37 °C in a non-CO$_2$ incubator for 1 h. All medium and injection reagents were adjusted to pH 7.4 on the day of assay. Four baseline measurements of OCR and ECAR were taken before sequential injection of mitochondrial inhibitors. Four readings were taken after each addition of mitochondrial inhibitor before injection of the subsequent inhibitors. The mitochondrial inhibitors used were oligomycin (1 μM), FCCP (0.5 μM), rotenone (0.5 μM) and antimycin A (0.5 μM). The OCR and ECAR were automatically calculated and recorded by Seahorse XF-24 software. After the assays, the plates were saved, and protein readings were measured for each well to confirm the equal cell numbers per well. The percentage of change compared with the basal rates was calculated as the value of change divided by the average value of baseline readings.

To study the dependency, capacity and flexibility of the different cell lines, we used the XF Mito Fuel Flex Kit (Agilent) following the manufacturer's recommendations. Fuel flexibility can be calculated by subtracting the fuel dependency from the fuel capacity for the pathway of interest. The following different inhibitors were used to establish the dependency, capacity and flexibility of *NRAS* and *BRAF* mutant cells for

glucose, glutamine and long-chain fatty acid metabolism: UK5099, which blocks the mitochondrial pyruvate carrier (MPC), thus inhibiting glucose oxidation; BPTES, which inhibits glutamine oxidation by inhibiting glutaminase (GLS1); and etomoxir, an inhibitor of long-chain fatty acid oxidation. Four baseline measurements of OCR were taken before and after sequential injection of inhibitors, including UK5099 (3 μM), BPTES (4 μM) and etomoxir (2 μM). Inhibitors were added following all sequence combinations to determine the different parameters for the following different assayed fuels: glucose, glutamine and long-chain fatty acids. The OCR was automatically calculated and recorded by Seahorse XF-24 software. After the assay, the plates were saved, and protein readings were measured for each well to normalize the obtained data. Dependency, capacity and flexibility parameters were calculated using the following formulas:

$$\text{Dependency}\,\% = \left[\frac{\text{Baseline OCR} - \text{Target inhibitor OCR}}{\text{Baseline OCR} - \text{All inhibitors OCR}}\right] \times 100$$

$$\text{Capacity}\,\% = \left[1 - \left[\frac{\text{Baseline OCR} - \text{Other two inhibitors OCR}}{\text{Baseline OCR} - \text{All inhibitors OCR}}\right]\right] \times 100$$

$$\text{Flexibility}\,\% = \text{Capacity}\,\% - \text{Dependency}\,\%$$

## Glucose uptake measurements

Briefly, the cell lines were at 60-80% confluence (exponential growth). For experiments performed after GS the cells were starved for 1 h in glucose-free medium. Then, all cell media was changed to medium containing 10 mM glucose in the presence or absence of sorafenib (10 μM). Glucose uptake was calculated by measuring the medium glucose consumption using a glucose assay kit (Abcam, ab65333) following the manufacturer's instructions.

## Metabolomics-isotope labeling

The samples were analyzed as previously described[55]. In brief, for isotope-labeling experiments, glucose was replaced with U-$^{13}$C6-glucose (Sigma Aldrich). SKMel103 and UACC903 were labeled for 40 min under basal conditions, in the presence of sorafenib (10 μM) and after 1 h of glucose starvation. Then, the cells were scraped, collected and frozen. Metabolites were extracted by adding 300 μl of cold methanol/water (8:1, v-v). The samples were vortexed for 30 s and immersed in liquid N$_2$ to disrupt cell membranes followed by 10 s of bath sonication. These two steps were repeated three times. Cell lysates were incubated for 20 min on ice before centrifugation (5000 × *g*, 15 min at 4 °C). Ten microliters of $^{13}$C-glycerol (150 ppm) were added to the supernatant as

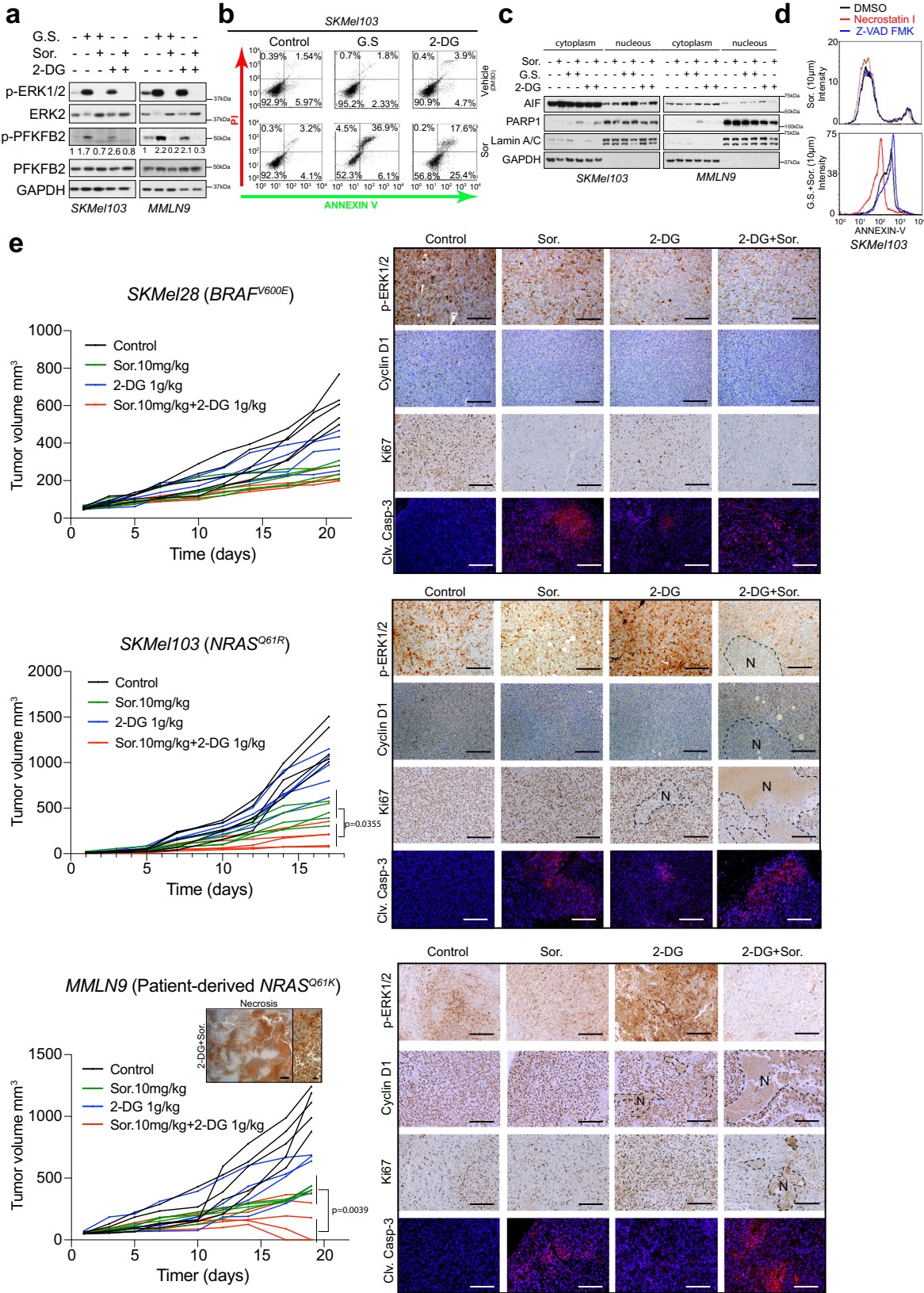

an internal standard. Next, 250 µl of each sample were dried under a stream of $N_2$ gas. Lyophilized polar extracts were incubated with 50 µl methoxyamine in pyridine (40 µg/µl) for 45 min at 60 °C. To increase the volatility of the compounds, we silylated the samples using 25 µl N-methyl-N-trimethylsilyltrifluoroacetamide with 1% trimethyl-chlorosilane (Thermo Fisher Scientific) for 30 min at 60 °C. A TRACE 1300 Series GC coupled to an Orbitrap Exactive mass spectrometer (Thermo Fisher Scientific) was used for isotopologue determination. Derivatized samples were injected (1 µl) into the gas chromatograph system with a split inlet 1:5 equipped with a TraceGOLD-5MS stationary phase column (30 m × 0.25 mm × 0.25 µm film, Thermo Scientific). Helium was used as a carrier gas at a flow rate of 1.5 ml/min. The oven

**Fig. 8 | The combination of 2-DG and sorafenib is effective against *NRAS*^Q61-mutated melanomas. a** Immunoblot showing the effects of 2-DG on RAS pathway activation and PFKFB2 phosphorylation compared to GS in *NRAS*^Q61 mutant melanoma cells. Numbers show the fold induction of the indicated ratio ($n = 3$ independent biological experiments). **b** Cell death detection by flow cytometry analysis in *NRAS*^Q61 mutant melanoma cells stained with propidium iodide (PI) and Annexin-V-GFP. The cells were treated for 4 h with sorafenib in the presence or absence of glucose (GS) and upon the addition of 2-DG (30 mM) and 10 mM glucose. A representative experiment is shown ($n = 3$ independent biological experiments). **c** Western blot showing the expression and localization of the indicated cell death-related proteins in *NRAS*^Q61-mutated cell lines after GS in the presence of sorafenib and/or 2-deoxy-glucose (2-DG). Lamin A/C and GAPDH are shown as loading and subcellular localization markers ($n = 3$). **d** Histograms showing cell death detection by flow cytometry analysis in *NRAS*^Q61 mutant melanoma cells stained with Annexin-V-GFP after the indicated treatments: Pan-caspase inhibitor zVAD-FMK and necrostatin 1 ($n = 2$ biologically independent experiments). **e** In vivo tumor growth assays showing the effect of the combination of sorafenib and 2-DG in *NRAS*^Q61- and *BRAF*^V600E-mutant melanoma cells. $n = 5$ mice per group of treatment in three different cells lines, including patient-derived cells (MMLN9); $p$ value, one-way ANOVA. Representative immunohistochemistry and immunofluorescence images against the indicated antibodies are shown. Bars represent 400 μm and 100 μm for the magnification of the graph inset. GS glucose starvation, Sor. Sorafenib, N necrosis.

temperature was programmed at 70 °C for 1 min and increased at 10 °C/min to 325 °C. The ionization performed was positive chemical ionization (CI) with isobutene as the reagent gas (flow rate at 1 mL/min). Mass spectral data were acquired in full scan mode (150–750 *m/z*), and the AGC target was set to 1 million ($1 \times 10^6$). The analyzer resolution settings were 60,000 at 200 *m/z*.

## PFK1 activity assay
PFK1 activity was measured using the Phosphofructokinase (PFK) Activity Colorimetric Assay Kit (Sigma) following the manufacturer's recommendations.

## Fructose 1,6-bisphosphate (F1,6-BisP) and fructose 2,6-bisphosphate determination (F2,6-BisP)
The cell extracts were washed twice in ice-cold PBS and lysed with either 100 mM NaOH plus 0.1% Triton X-100 (F2,6-BisP) or 6% HClO$_4$ (F1,6-BisP). After neutralization (basics extracts with AcOH/AcO-250 mM and acidic with 3 M KOH/KHCO$_3$), both basic and acidic extracts were centrifuged at 1.232 RCF (g-force) for 10 min at 4 °C. F1,6-BisP and F2,6-BisP were determined following the methodology described by Lang G. and Michal G[56]. and Van Schaftingen et al.[57]. The protein concentration was determined as described by the Bradford-based Bio-Rad Laboratories Assay.

## Immunocytochemistry (IC), immunofluorescence (IF) and immunohistochemistry (IHC)
For IC analysis, SKMel103, SKMel147, SKMel28 and UACC903 cells were seeded directly on cover slides in a 24-well plate (Sarstedt, Nümbrecht, Germany) at 50% confluence, incubated overnight and subjected to treatment the next day. Cell staining was performed as previously described[54]. For IF and IHC, 4 μm sections of formalin-fixed paraffin-embedded tumor samples were subjected to immunocytochemistry according to the manufacturer's antibody protocol. The samples were developed by using either secondary antibodies linked to horseradish peroxidase with the UltraView™ Universal DAB Detection Kit (Ventana Medical Systems) or secondary antibodies linked to fluorophores. Staining was performed either manually or on the automated immunostainer Beckmarck XT (Ventana Medical Systems, Roche, Tucson, AZ, USA). For the samples processed manually, antigen retrieval was performed using target retrieval solution pH 6.0 (Agilent). The samples were scanned (panoramic slide digital scanner) and evaluated by two independent pathologists (using 3DHistech software).

## Computational analysis of phosphorylation sites
Analysis of PFKFB2 putative phosphorylation sites was performed using *GPS 5.0* software[58].

## Gene expression analysis
RNA was purified using a Direct-Zol RNA kit (Zymo Research, Irvine, CA, USA). The isolation procedure was performed according to the manufacturer's recommendations. RNA amount and quality were assessed using a 2100 Bioanalyzer (Agilent). Each condition was performed in triplicate. Gene expression was analyzed using a *Clariom S Human Array* (Affymetrix Santa Clara, CA, USA). Raw data generated from the array were processed using the free-source software R studio. Intensity values were converted into gene expression values using Robust Multiarray Average (RMA) through the *BioConductor* package *oligo*, which consisted of background correction, logarithmic transformation, quantile normalization and probe normalization. Identification of differentially expressed genes (DEGs) was carried out using the *limma* package. A moderated *t* test was applied for each comparison based on the empirical *Bayes* method. Then, top tables were generated, with genes sorted from the most to least differentially expressed genes according to the log$_2$FC value. For functional analysis of the obtained differentially expressed genes, *Metascape* was used (metascape.org)[59]. Heatmaps were generated using *Heatmapper* (heatmapper.ca)[60].

## Mass spectrometry
After infection with *pLenti-rtTA2-His-PFKFB2-IRES-GFP*, SKMel103 and UACC903 cells were induced with doxycycline (Sigma) for 24 hours and subjected to one hour of glucose withdrawal and/or sorafenib treatment (15 μM). A His-tag pull down was performed for both untreated and treated cells following the procedure explained above. His-tagged PFKBF2 was enriched by Ni-IMAC chromatography using His-Trap columns (Sigma). The samples were concentrated and buffer exchanged to 6 M urea 50 mM ammonium bicarbonate using 0.5 ml 3KDa cutoff Amicon Ultra ultrafiltration devices (Millipore). The total protein content was quantified using an RCDC kit (Bio-Rad), and approximately 5 μg of each sample were taken for tryptic digestion. The samples were first reduced with dithiothreitol (DTT) to a final concentration of 10 mM for 1 h at room temperature and then alkylated with 20 mM iodoacetamide (IAA) for 30 min at room temperature in the dark. The carbamidomethylation reaction was quenched by the addition of N-acetyl-L-cysteine to a final concentration of 35 mM followed by incubation for 15 min at room temperature in the dark. The samples were diluted with 50 mM ammonium bicarbonate to a final concentration of 1 M urea. Modified porcine trypsin (Promega) was added at a ratio of 1:10 (w/w), and the mixture was incubated overnight at 37 °C. The reaction was stopped with formic acid to a final concentration of 0.5%. The digests were finally purified on reversed-phase C18 micro columns (ZipTip, Millipore) and maintained at −20 °C until further analysis. Phosphopeptide enrichment was performed according to Thingholm and Larsen[197], with some modifications. TiO$_2$ beads at 0.50 mg/μl were previously equilibrated in 1 M glycolic acid, 80% acetonitrile (ACN) and 1% trifluoroacetic acid (TFA). Peptides were diluted in 60% ACN with 1% TFA and added to 0.5 mg TiO$_2$. The suspension was incubated for 20 minutes at room temperature, with end-overend rotation for phosphopeptide binding. The mixture was then centrifuged at 2.082. RCF (g-force), and the supernatant containing nonphosphorylated peptides was discarded. TiO$_2$ beads with phosphopeptides were loaded on previously prepared homemade constructed stage tips (made using high-performance C18 extraction disks

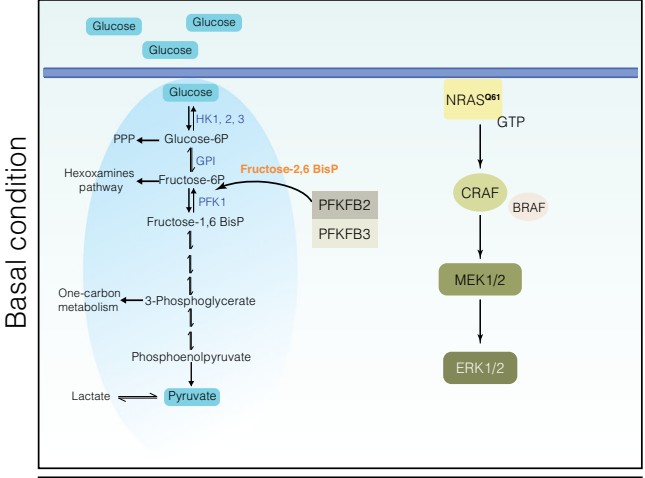

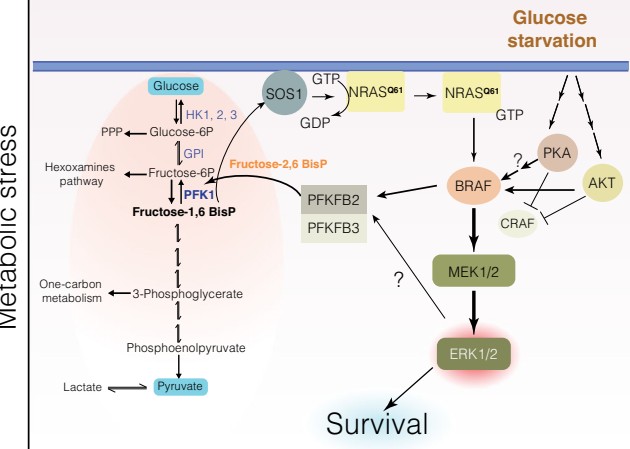

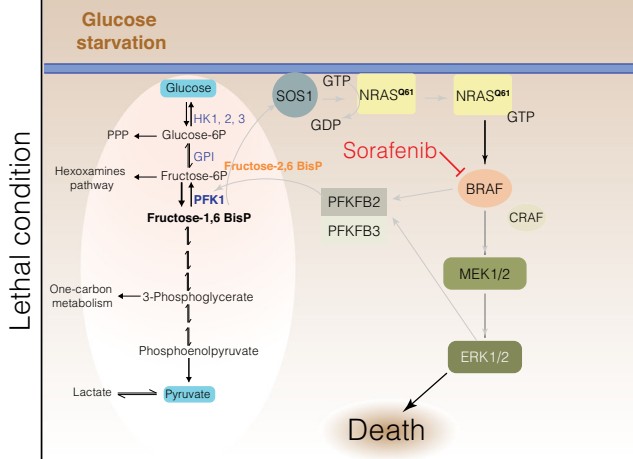

**Fig. 9 | Schematic representation of the proposed mechanism: under basal conditions,** *NRAS*[Q61] **mutant melanoma cells are dependent on glucose metabolism and preferentially use CRAF to signal through ERK1/2.** Schematic representation of the proposed mechanism: Under basal conditions, *NRAS*[Q61] mutant melanoma cells are dependent on glucose metabolism and preferentially use CRAF to signal through ERK1/2. Upon GS PKA and AKT are activated, promoting concomitant CRAF inactivation and BRAF activation and generating a feedback loop to sustain glycolysis. This involves the activation of the kinase activity of PFKFB2/3 that results in the production of fructose-2,6BisP, which in turn allosterically activates PFK1, inducing the generation of fructose-1,6BisP. This metabolite, under metabolic stress, activates SOS1, promoting the activation of *NRAS*[Q61] by exchanging GDP for GTP, leading again to the activation of BRAF and resulting in cell survival. Treating cells under this condition with sorafenib abolishes BRAF downstream signals, leading to cell death.

in pipette tips). After two successive washes with 60% ACN and 1% TFA, bound phosphopeptides were eluted first with 5% NH$_4$OH and then with 10% NH$_4$OH with 25% ACN. Eluted phosphopeptides were evaporated, resuspended in 0.1% FA and stored at −20 °C until further analysis. Tryptic digests were analyzed using a linear ion trap Velos-Orbitrap mass spectrometer (Thermo Fisher Scientific). Instrument control was performed using the Xcalibur software package, version 2.2.0 (Thermo Fisher Scientific). Peptide mixtures were fractionated by on-line nanoflow liquid chromatography using an EASY-nLC 1000 system (Proxeon Biosystems, Thermo Fisher Scientific) with a two-linear-column system. Digests (approx. 500 ng) were loaded onto a trapping guard column (Acclaim PepMap 100 nanoviper, 2 cm long, inner diameter 75 μm packed with C18, 3 μm particle size from Thermo Fisher Scientific) at 4 □l/min. Then, the samples were eluted from the analytical column (25 cm long, inner diameter 75 μm packed with Reprosil Pur C18-AQ, 3 μm particle size, *Dr. Maisch*). Elution was achieved by using a mobile phase from 0.1% FA (Buffer A) and 100% acetonitrile with 0.1% FA (Buffer B) and applying a linear gradient from 0 to 35% buffer B for 60 min at a flow rate of 300 nl/min. Ions were generated by applying a voltage of 1.9 kV to a stainless-steel nanobore emitter (Proxeon, Thermo Fisher Scientific) connected to the end of the analytical column on a Proxeon nanospray flex ion source.

The LTQ Orbitrap Velos mass spectrometer was operated in data-dependent mode. A scan cycle was initiated with a full-scan MS spectrum (from *m/z* 300 to 1600) acquired in the Orbitrap with a resolution of 30,000. The 20 most abundant ions were selected for collision-induced dissociation fragmentation in the linear ion trap when their intensity exceeded a minimum threshold of 1000 counts, excluding singly charged ions. Accumulation of ions for both MS and MS/MS scans was performed in the linear ion trap, and the automatic gain control (AGC) target values were set to $1 \times 10^6$ ions for survey MS and 5000 ions for MS/MS experiments. The maximum ion accumulation times were 500 and 200 ms in the MS and MS/MS modes, respectively. The normalized collision energy was set to 35%, and one microscan was acquired per spectrum. Ions subjected to MS/MS with a relative mass window of 10 ppm were excluded from further sequencing for 20 s. For all precursor masses, a window of 20 ppm and an isolation width of 2 Da were defined. Orbitrap measurements were performed, enabling the lock mass option (*m/z* 445.120024) for survey scans to improve mass accuracy. LC−MS/MS data were analyzed using the Proteome Discoverer software (Thermo Fisher Scientific) to generate mgf files. Processed runs were loaded into ProteinScape software (Bruker Daltonics, Bremen, Germany), and peptides were identified using Mascot (Matrix Science, London UK) to search the SwissProt database, restricting taxonomy to human proteins. MS/MS spectra were searched with a precursor mass tolerance of 10 ppm, fragment tolerance of 0.8 Da, trypsin specificity with a maximum of 2 missed cleavages, cysteine carbamidomethylation set as fixed modification and methionine oxidation, serine, threonine or tyrosine phosphorylation as variable modifications. The significance threshold for the identifications was set to *p < 0.05* for the probability-based Mascot score and a minimum ion score of 20, and the identification results were filtered to 1% false discovery rate (FDR) at the peptide level based on searches against a Decoy database. Relative quantification of the peptides corresponding to PFKFB2 phosphorylation sites was based on the integrated areas of the extracted ion chromatograms for each of the corresponding observed *m/z* values. The areas for the signals corresponding to both the unphosphorylated and phosphorylated peptides were measured for each of the samples.

**Pull-down and PFKFB2/PFKFB3 dimerization analysis**

In vitro dimerization experiments were performed using PFKFB2 (Origene), PFKFB3 (Origene) and GST-BRAF (Millipore) recombinant proteins. For the study of heterodimer formation, solutions containing both PFKFB2 and PFKFB3 were incubated in 50 mM NaCl, 50 mM Tris

pH 7.4, 1 mM EDTA and 1% NP-40 for 30 min at 30 °C and subjected to either PFKFB2 or PFKFB3 immunoprecipitation. Immunoprecipitated complexes were subjected to SDS–PAGE, transferred to a PVDF membrane and blotted with anti-PFKFB3 or anti-PFKFB2 antibodies. To establish BRAF-PFKFB2 and BRAF-PFKFB2-PFKFB3 complexes, GST-BRAF recombinant protein was incubated with PFKFB2, alone and in the presence of PFKFB3, for 30 min at 30 °C. GST-bind resin (Millipore) was used to pull down the obtained complexes.

## Animal study

Animal experiments were conducted and designed according to protocols approved by the Institutional Animal Care and Use Committee of Vall d´Hebron Institute of Research. For xenograft animal models, $5 \times 10^5$ human SKMel103, SKMel28, or MMLN9 cells were subcutaneously implanted in athymic$^{nu/nu}$ mice (female 4-6 weeks old, Harlan Laboratories). When the tumors reached sizes between 50-100 mm³, the mice were randomized into treatment groups ($n = 5$ per group). The treated groups received 1000 mg/kg 2-DG in drinking water and an intraperitoneal injection of 10 mg/kg sorafenib every day. Control groups were treated with vehicle (PBS with 5% (v/v) DMSO). Tumor volume was calculated as $L \times w \times h$, where "$L$" was the major diameter, "$w$" the minor diameter and "$h$" the third axis of the tumor mass. Maximal tumour size/burden permitted by the Vall d´Hebron Institute of Research Ethics Committee is 1000-1200 mm³. Tumor samples were collected at the end of the experiment for further analysis. The results are presented as the tumor volume mean ± SD.

## Statistics

The statistical tests used are reported in the figure legends. In summary, statistical analyses were performed in GraphPad Prism 9.0 (GraphPad Software Inc.) using a 2-tailed Student's $t$ test to compare differences between 2 groups or 1-way ANOVA with multiple-comparisons tests to compare 3 or more groups. Statistical tests used for the analyses of transcriptomes (microarrays and gene set enrichment analysis) were performed using the BioConductor package oligo R-limma package[61]

The schemes and explanatory figures were generated using Biorender (www.biorender.com).

## Reporting summary

Further information on research design is available in the Nature Portfolio Reporting Summary linked to this article.

## Data availability

The gene expression microarray datasets generated in this study have been deposited in the ArrayExpress database under accession code E-MTAB-10921. The mass spectrometry proteomics data generated in this study have been deposited in the ProteomeXchange Consortium database via the PRIDE partner repository under accession code PXD028297. The gene alteration data used in this study are publicly available at cBioPortal (Skin cancer, TCGA, Firehose Legacy; (https://www.cbioportal.org)). *PFKFB2* mRNA expression and DNA methylation data used in this study are publicly available at the CCLE database. (https://sites.broadinstitute.org/ccle). The remaining data are available within the Article, Supplementary Information or Source Data file. Source data are provided with this paper.

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

## Acknowledgements

This work was funded by Instituto de Salud Carlos III and co-funded by European Union (ERDF/ESF, "A way to make Europe"/"Investing in your future") PI14/0375-Fondos FEDER J.A.R., PI17/00043-Fondos FEDER; J.A.R., PI20/0384-Fondos FEDER; J.A.R., Euronanomed2-ISCIII (AC16/00019)-Fondos FEDER; J.A.R., Asociación Española Contra el Cancer (AECC-GCB15152978SOEN) (supported P.G.M., K.M.); J.A.R., Ramón Areces Foundation (supported K.M. and research); J.A.R. (PI17/00412)-Fondos FEDER; R.B., A.M., A.N.S. We thank A. Zorzano's laboratory for technical assistance and performance of Seahorse technology.

## Author contributions

Conceptualization: J.A.R., K.M. Investigation: R.E.P., K.M., S.G.O., S.S.R., P.G., J.H.L., R.B., A.M., A.N.S., F.C., O.Y., Y.D., M.P.A., B.F. and J.A.R.; resources: J.A.R., V.G.P. and E.M.C.; funding acqusition: J.A.R., A.M., R.B. A.N.S._. Methodology: R.B., O.Y., A.M. and A.N.S. Formal analysis: J.A.R., K.M., O.Y. and F.C. Writing-review and editing: J.A.R., K.M. Supervision: J.A.R.

## Competing interests

The authors declare no competing interests.
