## [Peer Review File · Nature Communications]

REVIEWERS' COMMENTS

Reviewer #1 (Remarks to the Author):

In this report the authors wish to show that in NRASQ61R mutant melanomas "glucose deprivation promotes a switch from CRAF to BRAF, rendering the cells sensitive to sorafenib" and BRAF sustains glucose metabolism through the phosphorylation of PFKFB2/PFKFB3 .

This manuscript is problematic in the way it is written and presented. The authors show in Figure 1 that glucose starvation enhances the levels of pERK in NRASQ61R but not in BRAFV600E melanoma cells. This response was suppressed by treatment with the tyrosine kinase inhibitor sorafenib, suggesting the activation of a kinase.

The problems start with Figure 2. In this figure the role of RAF1 and BRAF are examined. The authors conclude that the main mechanism is "concomitant activation of BRAF" because "glucose starvation promoted the phosphorylation of BRAFS445".

However, Figure 2d shows that CRAF (i.e., RAF1) protein disappears upon treatment with G.S. (glucose starvation) and sorafenib, a point not even mentioned or discussed in the manuscript. In addition, total levels of BRAF (not just pBRAF) are proportionally enhanced in each case, again, not indicated by the authors.

The authors continue:

"However, we observed an increase in the immunoprecipitated BRAF amount in response to glucose starvation (as early as 15 min after treatment initiation), compatible with the formation of BRAF homodimers (Fig. 2f)."

The reason for BRAF homodimers formation is the absence of RAF1. The authors do not show RAF1 and BRAF proteins in the precipitates but rather Fig. 2f shows p-MEK1 and MEK1 appearing in different lanes that are not marked, so we don't know what is going on.

The conclusion that "While in basal conditions CRAF was the kinase signaling and BRAF was barely active, under metabolic stress BRAF became active while CRAF was turned off (Fig. 2g)" is wrong. The absence of CRAF activity is due to the lack of protein, not lack of activity.

There is no panel marked with g in Fig. 2!

In addition, Figure 2 contains a blot with antibodies to Bax, Bim and MCL1 that is not mentioned in the discussion of this Figure, nor in the Figure legend or later on in any other place (except the purchase of the antibodies).

Likewise, several panels in Fig. 3 are not marked.

The language is awkward and needs to be corrected.

Examples: switch of RAF use isoform, from CRAF to BRAF

However, NRAS mutant melanomas lack of specific line of treatment.

Given the success targeting sustaining proliferative signaling, long-lasting therapeutic responses are limited, and if metastatic, it stands as a difficult type of cancer to treat.

Reviewer #2 (Remarks to the Author):

The authors studied the possible relationship between the oncogenic alterations of the MAPK pathway (NRASQ61 and BRAFV600E mutations) and metabolic stress in melanoma. They showed that glucose starvation induces ERK hyperactivation and sensitized NRASQ61 mutant melanomas to sorafenib-mediated cell death. ERK activation upon metabolic stress was NRAS-dependent and promoted a switch in the use of BRAF instead of CRAF to activate the MAPK pathway. BRAFV600E

and NRASQ61 mutant melanoma cells show different metabolic profiles and glucose but not pyruvate nor glutamine rescues sorafenib-induced cell death upon metabolic stress conditions. The authors suggest a link between PFKFB2/3 and RAS-ERK1/2 pathway through phosphorylation of PFKFB2 by BRAF.

The study is interesting and in general the data are clear and solid. However, several important shortcomings must be addressed as described below.

There are no indications in the manuscript of the dose of inhibitors used in all the experiments except in Figure 3C where a high dose of Sorafenib (15uM) is used. The lack of these details makes it very difficult to interpret the data correctly and could suggest nonspecific effects of high doses of sorafenib.

Figure 1A : The authors should show that the phosphorylation of MEK follows the phosphorylation of ERK to demonstrate that glucose starvation acts upstream of ERK rather than downstream of it.

Figure 2 : Figure 1A showed a rapid induction of ERK after 30 min of glucose starvation in RAS mutated cell lines. However, to decipher the molecular mechanism of this induction, all the experiments presented Figure 2 are done after 4 hr of glucose starvation. This time discrepancy makes it difficult to validate the switch of BRAF instead of CRAF as a driver of ERK induction. In these conditions, one cannot rule out that BRAF activation is a consequence and not a cause of ERK hyperactivation.

Figure 2b : It is striking and in contradiction with published data that NRAS inhibition in NRAS-mutated cells has no effect on ERK phosphorylation in the control conditions (C). It has been widely shown that MAPK activation is dependent on RAS in RAS-mutated cells. How can the authors explain this discrepancy?

Figure 2d: The most striking effect in this experiment is probably the complete inhibition of CRAF expression in the four cell lines treated with GS+sor which is surprisingly not addressed in the manuscript. Knowing that sorafenib regulates translation by inhibiting EIF4E phosphorylation, the effect of sorafenib could be explained by an inhibition of translation enhanced by glucose starvation. Is there any evidence to discredit the inhibition of translation by sorafenib in these experiments?

Figure 2f: The changes in CRAF and BRAF phosphorylations and the in vitro kinase activities suggest a switch in RAF isoform upon glucose starvation. However, because the phosphorylation of RAF on these sites is not a definitive marker of its activity and because the kinase activity is realised in vitro, the switch in RAF isoform need to be confirmed using siRNA to disrupt BRAF or CRAF and evaluate the effect on ERK phosphorylation.

Figure 4c: It is not clearly indicated which alterations have been evaluated. Did the authors consider only genetic alterations (mutations, copy-number alterations) or included mRNA expression? What is the fraction of melanoma presenting alteration of PFKFB2 in the Firehorse Legacy study used.

Figure 5A : Induction of PFKFB2 phosphorylation by glucose starvation is only seen in two out of four cell lines making it impossible to draw a general conclusion on PFKFB2 phosphorylation induced by glucose starvation.

Figure 5E: Although the in vitro phosphorylation of PFKFB2 by BRAF is convincing (figure 6f) the decrease in PFKFB2 phosphorylation with BRAF siRNA is not. The decrease in PFKFB2 phosphorylation parallels the decrease in PFKFB2 total protein. Therefore, the ration p-PFKFB2/ PFKFB2 does not seem to vary upon BRAF inhibition making it impossible to conclude on the effect of BRAF on PFKFB2 phosphorylation ex vivo.

The two points raised above show that this work does not clearly support the claim that there is a connection between glucose starvation, phosphorylation of PFKFB2 and BRAF. Additional evidences are needed to draw this conclusion.

Figure 6k: The data presented connect PFKFB2 and PFKFB3 expressions to ERK phosphorylation upon glucose starvation. There is however no explanation in the text and in Figure 8 to explain how PFKFB2 and PFKFB3 can regulate ERK activity upon glucose starvation.

Minor points:

Figure 2c : The western blot is not described in the text.

Figure 6b: Was CRAF present in the His-PFKFB2 pull down?

Figure 7b : The western blot is not described in the text.

Figures 6e/h lack control with GST alone or IgG.

Figure 6k: The opposite effects on ERK phosphorylation and ERK expression is striking. How can the authors explain the presence of phosphorylated ERK but the absence of total ERK proteins in some samples?

Reviewer #3 (Remarks to the Author):

This is an interesting study from McGrail et al., that shows that NRAS-Q61 mutated melanomas rely on glucose metabolism, and that inhibition of glycolysis through inhibitors or glucose deprivation sensitize these cells to sorafenib treatment. In particular, glucose deprivation of NRAS-Q61 mutated melanomas triggers a switch from CRAF to BRAF signalling, resulting in PFKFB2 phosphorylation and sustained glycolytic activity. A feedback loop hyperactivates the RAS-ERK1/2 pathway, which in turn sensitizes cells to sorafenib treatment.

Some major considerations:

1) The Seahorse data (Fig. 3) indicate lower levels of both ECAR and OCR in NRAS-Q61 cells as compared to BRAF-V600E cells. Based on these data, the authors speculate that glucose metabolism is particularly relevant for NRAS-Q61 cells, because it is used in pathways that branch off from glycolysis. It would be important to quantify glucose uptake in both types, e.g. using fluorescent NBD-glucose, to estimate how much glucose both cell types use. To substantiate the claim that glucose is used to generate macromolecule intermediates or reducing power, the authors should perform a tracing experiment using [U-13C]-glucose to show remodelling (and not just dampening) of glucose metabolism. Preferable, such tracing experiments are performed both in the presence and absence of sorafenib.

2) From Figure 5, the authors conclude that NRAS-Q61 but not BRAF-V600E melanomas sustain PFK1 activity rates. Are the differences in figure 5f significant? Please indicate. Also, the differences between BRAF and NRAS mutated melanoma types seem to be driven by differences in PFK1 activity under basal conditions (30 versus 20 mU/ug protein, respectively). Upon glucose starvation, the residual activity appears more or less equal (15 mU/ug protein in both types), suggesting that glucose starvation is reducing PFK1 activity to similar levels in both types. Can the authors please comment on this?

3) The authors claim from supplementary Figure 4B that the maintenance of PFK1 activity is accompanied with a sustained production of F2,6-BP and F1,6-BP. However, F2,6-BP levels are lowered by glucose starvation in all cell lines tested. To nonetheless make the claim that F2,6-BP production is sustained, the authors should include mutant BRAF cells in these experiments as a control.

4) The authors link PFKFB2/3 to the ability of NRAS-Q61 melanomas to induce ERK1/2 phosphorylation under conditions of glucose starvation (Fig 6k). The link between PFKFB2/3 phosphorylation and sorafenib sensitivity is however less convincing. If PFKFB2/3 are indeed key regulators of this process, one would expect that PFKFB2 and/or PFKFB3 overexpression sensitizes cells to sorafenib treatment. The authors should show this.

Minor comments:

1) How is ATP-linked respiration calculated in figure 3a? ATP-linked respiration is defined as the difference between the basal respiration and the proton leak, and appears larger in BRAF-V600E

cells compared to NRAS-Q61 cells (judging from the left graphs).

2) In Figure 4b, the authors zoom in on glycolysis. The reason for this is not well explained in the text. Arguably, many other metabolic and signalling pathways are also critical for the response to glucose starvation, and studying these may identify other key regulators of the response to metabolic stress. Can the authors provide additional explanation to justify their choice?

3) Several subfigures in Figures 5 and 6 are not correctly referred to in the text, e.g 5f, 5g, 5h and 6d. Please check.

4) In figure 5h, does adding F2,6-BP to cells induce similar phosphorylation of ERK1/2 under glucose deprivation conditions?

5) Figure 6b/6c: the differential effects of G.S. + Sor treatment on BRAF and NRAS melanomas seem more pronounced on western blot compared to the phosphoproteomics results: on western blot PFKFB2-S483 phosphorylation is completely inhibited in NRAS cells, but hardly affected in BRAF cells. Using phosphoproteomics, peptides containing phosphorylated S483 seem substantially downregulated in both cell lines. Can the authors comment on this apparent difference?

Reviewer #4 (Remarks to the Author):

This study investigates the response of NRAS mutant melanoma cells to glucose starvation as compared with BRAF mutation. The results show that glucose starvation induced a rapid hyper-phosphorylation of ERK1/2 in NRAS mutant cell lines, while not in BRAF mutant cells. The authors suggested that glucose starvation induced a NRAS mutant-BRAF-ERK-PFKFB2/3-PFK1-NRAS activation feedback loop to support melanoma cell survival. They also showed that glucose starvation or 2-DG sensitizes NRAS mutant cells to sorafenib treatment. It is an interesting study. However, there are many issues for this study. The conclusion of glucose starvation inducing the switch of CRAF to BRAF downstream of NRAS mutant and following feedback loop is mainly based on consumption and hypothesis, lacking experimental validation. There are also many inconsistencies in figures, and missing and wrong-labeled information. Overall, the study is premature and conclusion can not be supported by provided data.

Main issues:

1. It is unclear how glucose deprivation triggers the CRAF/BRAF isoform switch downstream NRAS mutant. The study stated that NRAS mutant activates BRAF, but there is no direct data evidence showing that NRAS mutant binds to BRAF. Fig. 2a pull-down assay used CRAF-binding domain to pull down NRAS, which showed that dramatically increased upon glucose starvation, opposite to the switch claim. BRAF-binding domain should be tested.

2. The study claims that NRAS mutant through activation of BRAF to support melanoma cell growth. In Fig. 2d, could siBRAF mimic Sorafenib cell killing effects in glucose starvation condition?

3. Fig. 2d, p-BRAF S445 band shows very minor effects in response to glucose starvation, not consistent with p-ERK1/2 levels. Fig. 2c should check caspase 3/6/9 cleavage and PRAP for apoptosis marker.

4. Fig. 4. Many panels showing the changes of p-PFKFB2 levels upon glucose starvation are very minor, difficulty to conclude that glucose starvation activates PFKFB2 in NRAS mutant cells. Fig 5e, p-PFKFB2 levels change very minor upon siNRAS or siBRAF.

5. Fig. 6a, there is no BRAF protein shown in PFKFB2 pull-down protein list, which does not support their direct association. Fig. 6d, there is no increase of PFKFB2 and BRAF binding upon glucose starvation compared with control. Fig. 6h, lacking negative pull-down control.

6. Fig. 7a, should also test PFKFB2 levels upon 2-DG treatment. Fig. 7b should test caspase 3/6/9 cleavage. There is no explaining for what is the mean for BIM, MCL1, BAX levels in the text.

7. Fig. 8. The proposed model is not strongly supported by provided data, there are too many mechanistic gaps for the feedback loop, lacking experimental data support the connection between

PFK1/SOS1/NRAS mutant activation and following pathway.

8. The study should specific indicate glucose starvation, instead to use metabolic stress in whole study as metabolic stress can include many different types of stress.

There are also many mis-information:

1. In supplementary figure 1a, "NRASQ61 mut" and "BRAFV600E mut" should be switched.
2. In supplementary figure 1b, could you explain why DUSP1 mainly expresses in nucleus in NRASQ61 mutant cells and only expresses in cytoplasm in BRAFV600E mutant cells.

4. In the main text, there are too much errors in the citations of figures.
Some samples are as follows, please check:

cited in main text Supposed version
Supplementary Fig. 1c Supplementary Fig. 1d
Fig.2e Fig.2d
Fig.2f Fig.2e
Fig.2g Fig.2f
Figure 4c after Fig.5b Fig.5c
Fig.4d after Fig.5b Fig.5d
Fig.4e after Fig.5b Fig.5e
Fig. 5c Fig.5f
Fig. 5d Fig.5g
Fig 5e Fig.5h
Second Fig. 6c Fig. 6d

5. There is no Fig. 2g.

6. please check Fig. 5e and result statement regarding F1,6-BP promoted the phosphorylation of ERK1/2..., no data found. Which cell line?

7. Almost all the inhibitors, such as sorafenib, salirasib, regorafenib and salirasib, are not labelled with the dosage in the figure caption.

RESPONSE TO REVIEWERS' COMMENTS:

Reviewer #1 (Remarks to the Author):

In this report the authors wish to show that in NRASQ61R mutant melanomas “glucose deprivation promotes a switch from CRAF to BRAF, rendering the cells sensitive to sorafenib” and BRAF sustains glucose metabolism through the phosphorylation of PFKFB2/PFKFB3 .

This manuscript is problematic in the way it is written and presented. The authors show in Figure 1 that glucose starvation enhances the levels of pERK in NRASQ61R but not in BRAFV600E melanoma cells. This response was suppressed by treatment with the tyrosine kinase inhibitor sorafenib, suggesting the activation of a kinase.

The problems start with Figure 2. In this figure the role of RAF1 and BRAF are examined. The authors conclude that the main mechanism is “concomitant activation of BRAF” because “glucose starvation promoted the phosphorylation of BRAFS445”.

We are not sure about the difficulties the referee is referring to. Chronologically in Fig. 2 we show that NRAS oncogene needs to be activated to observe the effects promoted by glucose starvation (G.S.) (Figs. 2a and 2b), furthermore inhibition of NRAS in the absence of glucose promotes the same deadly effects as the combination of G.S. and sorafenib (old-Fig 2c, new supplementary Fig 2a). Since the use of U0126 or trametinib (MEK1/2 inhibitors) under G.S. did not lead to cell death (Fig. 1d), the data suggested the involvement of RAF proteins in the mechanism. RAF proteins activity is tightly controlled by both intramolecular and extra-molecular interactions, many of them mediated by phosphorylation-dephosphorylation of particular residues. In this matter phosphorylation of CRAF Ser259 and Ser289/Ser296/Ser301 cluster is associated to the resting inactive state of the protein, while activation of the CRAF involves phosphorylation at multiple activating sites, including Ser338. BRAF Ser445 is the equivalent to Ser338 in CRAF, and even though this residue is constitutively phosphorylated, Ser445 phosphorylation still contributes to BRAF activation by elevating basal and consequently RAS-stimulated activity. We show that residues Ser259 and Ser289/Ser296/Ser301 of CRAF as well as Ser445 of BRAF become phosphorylated in response to G.S through PKA and AKT, inactivating CRAF and contributing to BRAF activation (Supplementary Fig. 2b-d). We have added the quantification of p-BRAF Ser445 normalized by the amount of BRAF/GAPDH showing an increased amount of p-BRAF Ser445 in NRAS mutant cells after G.S. In addition to this, supplemental Fig. 2e (old Fig. 2e) shows the increased amount of BRAF immunoprecipitated compatible with the formation of homodimers, beginning as early as 15 minutes after G.S. Finally, the *in vitro* kinase activity (in triplicates, as stated in the figure legend) of immunoprecipitated CRAF and BRAF showed, the activation of BRAF and the inhibition of CRAF kinase activity after G.S. Altogether, this evidence led us to conclude that G.S. promotes the inactivation of CRAF with the concomitant activation of BRAF, not only because of the phosphorylation state of p-BRAF Ser445. Nevertheless, in Fig 2c we have added a time course (early time points) of immunoprecipitated BRAF and CRAF showing the accumulation of BRAF phosphorylated at Ser445 and CRAF phosphorylated at Ser259 and cluster Ser289/Ser296/Ser301. Importantly, this experiment also shows the increased binding of NRAS to the immunoprecipitated BRAF as early as 15' after G.S., while there was no variation/decreased in the amount NRAS associated to CRAF. In addition to this, partial depletion of either BRAF or BRAF/CRAF results in a decreased response to G.S. while no effects can be observed after CRAF depletion (Fig. 2f-g). Overall, all these data support and confirm the involvement of BRAF in the proposed mechanism.

However, Figure 2d shows that CRAF (i.e., RAF1) protein disappears upon treatment with G.S.

(glucose starvation) and sorafenib, a point not even mention or discussed in the manuscript. In addition, total levels of BRAF (not just pBRAF) are proportionally enhanced in each case, again, not indicated by the authors.

We thank the reviewer comment. This is a good question that has been addressed and discussed in the manuscript. Nevertheless, we want to clarify that the mechanism described in response to G.S is independent of the apparently degradation of CRAF promoted by sorafenib under G.S. In relation to this, several years ago it was demonstrated that dephosphorylation of Ser259 in CRAF regulated it association to the membrane¹. Sorafenib treatment inhibited the G.S.-induced phosphorylation of Ser259. Thus, as showed in the below figure and Supplementary Fig 2f-g, treatment of cells with sorafenib under G.S (1h and 4h) promoted the accumulation of CRAF in the cell lysates insoluble fraction. Furthermore, CRAF appears not to be degraded via the proteasome, since treatment of cells with proteasome inhibitor MG-132 led to the accumulation of CRAF in control conditions as well as 1h or 4h after G.S. However, in the presence of sorafenib under G.S. (at 1h and 4h) CRAF amounts decreased. Thus, the disappearing of CRAF in G.S. condition in the presence of sorafenib is not due to CRAF degradation, but its traslocation to the membrane and association to the non-soluble fraction of cell lysates. This data also argues with the reviewer's comment suggesting that "The absence of CRAF activity is due to the lack of protein, not lack of activity".

(a) SkMel103 cells were subjected to the indicated treatments, G.S. (glucose starvation) Sor. (sorafenib 10 μ M) in the presence or absence of the proteasome inhibitor MG-132. Western blot show the amounts of the indicated proteins. (b) Western blot showing the indicated proteins in the soluble and insoluble lysate fractions upon the indicated treatments.

In respect to the BRAF amount, it is true that the levels BRAF accumulated in G.S. condition. This accumulation can be also observed in a cell dependent manner upon G.S.+gluc(2h) or G.S.+gluc(2h)+sorafenib. As above mentioned, it is known that S445 phosphorylation contributes to BRAF activation by elevating basal and consequently RAS-stimulated activity. In the figure the reviewer is referring to, p-BRAF^{S445} is only accumulating in G.S. treated samples (in a cell independent manner). The modification on BRAF molecules upon G.S. can also be observed by the slight band shift observed upon this condition. We have quantified and normalized the p-BRAF^{S445} respect the BRAF/GAPDH amounts and the values have been added to the figure. Additionally, and as above stated, we have added additional experiments (Fig 2c-time course of immunoprecipitated BRAF in response to G.S. and supplementary Fig 2f-g) that provide further evidence and clearly support the participation of BRAF in the proposed mechanism.

The authors continue:

"However, we observed an increase in the immunoprecipitated BRAF amount in response to glucose starvation (as early as 15 min after treatment initiation), compatible with the formation of BRAF homodimers (Fig. 2f)."

The reason for BRAF homodimers formation is the absence of RAF1. The authors do not show RAF1 and BRAF proteins in the precipitates but rather Fig. 2f shows p-MEK1 and MEK1 appearing in different lanes that are not marked, so we don't know what is going on.

We are sorry about the misunderstanding. The homodimer formation occurs in response glucose starvation in the absence of any drug. Moreover, the above explanation demonstrates that CRAF is not disappearing in G.S+Sor condition. In supplemental Fig. 2e (old Fig. 2e), we show the immunoprecipitated BRAF and CRAF in control conditions and after glucose starvation in a time course manner, in two different NRAS mutant cell lines. The right half of the blots show that cells express plenty amounts of CRAF. The expression of CRAF under glucose starvation can be also observed in Fig. 2d-2e.

In Fig 2d, BRAF and CRAF were immunoprecipitated in three different experiments per condition. Then a kinase assay was performed using recombinant MEK1 as a substrate. We have added the amounts of BRAF and CRAF immunoprecipitated in the kinase assay.

The conclusion that “While in basal conditions CRAF was the kinase signaling and BRAF was barely active, under metabolic stress BRAF became active while CRAF was turned off (Fig. 2g)” is wrong. The absence of CRAF activity is due to the lack of protein, not lack of activity.

Again, we are sorry for the misunderstanding. As above exposed, we show evidence supporting the inactivation of CRAF (phosphorylation of CRAF Ser259 and Ser289/Ser296/Ser301) and increased activity of BRAF (phosphorylation at Ser445). Moreover, as showed in Supplementary Fig. 2e (old Fig. 2e) and Fig 2d-e, the amounts of CRAF under glucose starvation do not change. In addition to this, new Fig. 2c also shows the accumulation of pBRAF^{S445} and the recruitment of NRAS to BRAF immunocomplexes 15' after G.S. supporting our statement and the mechanism.

There is no panel marked with g in Fig. 2!

ç

This typo has been corrected accordingly.

In addition, Figure 2 contains a blot with antibodies to Bax, Bim and MCL1 that is not mentioned in the discussion of this Figure, nor in the Figure legend or later on in any other place (except the purchase of the antibodies).

We understand the reviewer’s concern. This figure has become supplemental Fig. 2a. The purpose of the blot in old Fig 2c is to molecularly support the results from the flow cytometry data showing the apoptosis promoted by the combination of NRAS inhibitors and G.S. Due to the extension of the manuscript, the restricted amount of words allowed and the relevance of this piece of data contextualized within the whole manuscript, we just mentioned the relevant biological consequences related to the mechanism we are describing. We have changed the western blots and added more describing information to the text.

Likewise, several panels in Fig. 3 are not marked.

We assume that the confusion comes from Fig 3a. Initially we did not add marks to the graphs on the right part of the figure because this data is inferred from the metabolic profiles (Seahorse technology) showed on the left. Nevertheless, to satisfy the reviewer we have added new marks to the graphs (Fig 3b, and Fig 3c). Now old Fig 3b, 3c and 3d. have become new Fig 3d, 3g and 3h respectively. Text has been corrected accordingly.

The language is awkward and needs to be corrected.

Examples: switch of RAF use isoform, from CRAF to BRAF

However, NRAS mutant melanomas lack of a specific line of treatment.

Given the success targeting sustaining proliferative signaling. long-lasting therapeutic responses are limited, and if metastatic, it stands as a difficult type of cancer to treat.

We revised the text accordingly.

Reviewer #2 (Remarks to the Author):

The authors studied the possible relationship between the oncogenic alterations of the MAPK pathway (NRASQ61 and BRAFV600E mutations) and metabolic stress in melanoma. They showed that glucose starvation induces ERK hyperactivation and sensitized NRASQ61 mutant melanomas to sorafenib-mediated cell death. ERK activation upon metabolic stress was NRAS-dependent and promoted a switch in the use of BRAF instead of CRAF to activate the MAPK pathway. BRAFV600E and NRASQ61 mutant melanoma cells show different metabolic profiles and glucose but not pyruvate nor glutamine rescues sorafenib-induced cell death upon metabolic stress conditions. The authors suggest a link between PFKFB2/3 and RAS-ERK1/2 pathway through phosphorylation of PFKFB2 by BRAF.

The study is interesting and in general the data are clear and solid. However, several important shortcomings must be addressed as described below.

1. There are no indications in the manuscript of the dose of inhibitors used in all the experiments except in Figure 3C where a high dose of Sorafenib (15uM) is used. The lack of these details makes it very difficult to interpret the data correctly and could suggest nonspecific effects of high doses of sorafenib.

The reviewer is right, we have added this information accordingly. We agree with the reviewer that we could be in the high range concentration of sorafenib, however, this concentration has been widely used by the scientific community (Refs. 2, 3, 4, 5, 6, 7). Nevertheless, as showed in the figure below, the effects of sorafenib inhibiting ERK1/2 activation and promoting cell death upon G.S. conditions are exactly the same at 5uM, 10uM or 15uM, discarding any nonspecific effects of sorafenib.

SKMe103 and SKMe147 cells were treated with the indicated concentrations of sorafenib in glucose starvation (G.S.) conditions. Western blot shows the inhibition of ERK1/2 phosphorylation at the indicated concentrations of sorafenib. On the right cell death was equally promoted using all the tested concentrations.

2. Figure 1A: The authors should show that the phosphorylation of MEK follows the phosphorylation of ERK to demonstrate that glucose starvation acts upstream of ERK rather than downstream of it.

We had checked the activation of MEK1/2 in different experiments but they were not included. We have now added the blots showing the p-MEK1/2 and MEK1/2 amounts to Fig 1a. However, there were indirect evidence in the results that supported the upstream

activation of the pathway. Supplemental Fig. 1d shows that MEK1/2 inhibitors U0126 and Trametinib almost totally block the hyperactivation of ERK1/2 after G.S. Additionally, knockdown of NRAS abolished the downstream hyperactivation of the pathway in response to G.S. Altogether these data support both, the oncogene dependency for the observed effect and the hyperactivation of the pathway upstream ERK1/2.

3. Figure 2 : Figure 1A showed a rapid induction of ERK after 30 min of glucose starvation in RAS mutated cell lines. However, to decipher the molecular mechanism of this induction, all the experiments presented Figure 2 are done after 4 hr of glucose starvation. This time discrepancy makes it difficult to validate the switch of BRAF instead of CRAF as a driver of ERK induction. In these conditions, one cannot rule out that BRAF activation is a consequence and not a cause of ERK hyperactivation.

We understand the reviewer concerns. In Fig. 2 there were two experiments (Fig. 2b and old Fig. 2d -> new Fig. 2e) that were performed at 4hr of G.S. for different reasons. Although activation of RAS occurs 15-30' after G.S. (Fig. 2a and new Fig. 2c) which supports the early upstream RAF activation of the pathway, the experiment in Fig. 2b was performed 4hr after G.S. to assure that even at the longest timing investigated throughout the manuscript, the hyperactivation of ERK1/2 does not occurs without the participation of NRAS, confirming the role of the oncogene in the observed response and discarding a downstream feed-back mechanism. In Fig. 2d, the experiment was performed starving the cells from glucose for 4h, because this experiment includes the molecular recovery by the addition of glucose after 2 h of glucose starvation (2+2). We thought that applying these timings (2+2) we will assure the establishment of a strong response to metabolic stress (G.S.) and then, will give enough time (if necessary) for recovery, maintaining the longest time of treatment showed throughout the manuscript. Additionally, old Fig. 2e (new supplemental Fig. 2e) show the formation of BRAF homodimers as early as 15' after G.S. and the kinase assay was performed after 1h of G.S. Nevertheless, we have added new Fig. 2c showing a time course of immunoprecipitated BRAF and CRAF in response to G.S. The figure shows the accumulation of p-BRAF^{S445}, p-CRAF^{S259} and p-CRAF^{S289/296/301} at early time points (15' and 30'). Importantly this experiment also shows the recruitment of NRAS to BRAF as early as 15' after G.S., without any change in the NRAS molecules bound to CRAF. Altogether, this data supports the inactivation of CRAF (phosphorylation on Ser259 ; Ser289/296/301 and minimal kinase activity) and the kinase activation of BRAF (increased detection of Ser445 phosphorylation, increased binding to NRAS molecules to BRAF at 15' and the formation of homodimers at 15'). Knowing that BRAF is the member of the RAF family (ARAF, BRAF and CRAF) with the most potent kinase activity, showing the highest affinity for MEK1/2 phosphorylation^{8,9}, we believe this data rules out the switch of BRAF instead of CRAF as a driver of ERK hyperactivation.

4. Figure 2b : It is striking and in contradiction with published data that NRAS inhibition in NRAS-mutated cells has no effect on ERK phosphorylation in the control conditions (C). It has been widely shown that MAPK activation is dependent on RAS in RAS-mutated cells. How can the authors explain this discrepancy?

We understand the reviewer's concern however, there is a fair explanation for this apparently contradiction. RAS-ERK pathway mediates the signaling of a wide variety of hormones, growth factors and differentiation factors, as well as tumor-promoting substances. Thus, the different enzymes within this signaling module are in charge of collecting and funnel different upstream stimuli that result in a cell biological response. These molecules (modules) are located at different subcellular localizations to timely accomplish the cell requirements. Thus, just a very small fraction of them become activated in response to a certain stimulus or at basal level. For example, up to 3% of cellular CRAF can be found in association with Ras and just between 2-5% participate in response to a certain stimulus^{10,11,12}. It is also known that the scaffold protein

KSR1 only binds less than 5% of endogenous CRAF¹³, indicating that KSR1 affects only a subset of RAF functions, and RAF members might be present in other protein complexes. Thus, when it comes to knockdown these molecules, depletion of 80-90% of the protein might not be enough to affect the basal state of the pathway or even the response to certain stimulus, since just 3-5% of the total protein pools participate in the response. This is what is happening in Fig. 2b, even though 80-90% of NRAS protein is depleted, the 10% left over is enough to sustain the basal activity of the pathway, however, the situation is challenged when it comes to respond to glucose starvation, either by the number of molecules needed, or the absence of enough molecules at a particular subcellular location. In addition to this, since these cell lines are addicted to the *NRAS*^{Q61} oncogene, reaching the depletion limits where the basal activity of the pathway is affected, negatively select these cells, making difficult to obtain viable cells under these circumstances. Similar situation occurs with RAF proteins. As shown in the figure below, despite the depletion of RAF proteins reaches 80-90% of total protein, this does not transduce to the basal levels of p-ERK signaling (i.e. SKMel103 cells, similar results obtained in other *NRAS* mutated melanoma cell lines) .

Western blot shows ERK1/2 phosphorylation status in response to glucose deprivation after knockdown of BRAF (left) or CRAF (right) in SKMel103 cells (*NRAS*-mutated).

The addiction of these cells (*NRAS* mutant cells) to RAS pathway signaling makes challenging to perform the genetic validation of some of the mechanisms involved.

5. Figure 2d: The most striking effect in this experiment is probably the complete inhibition of CRAF expression in the four cell lines treated with GS+sor which is surprisingly not addressed in the manuscript. Knowing that sorafenib regulates translation by inhibiting EIF4E phosphorylation, the effect of sorafenib could be explained by an inhibition of translation enhanced by glucose starvation. Is there any evidence to discredit the inhibition of translation by sorafenib in these experiments?

We thank the reviewer comment. This is a good question that has been addressed and discussed in the manuscript. Nevertheless, we want to clarify that the mechanism described in response to G.S is independent of the apparently degradation of CRAF promoted by sorafenib under G.S. In relation to this, several years ago it was demonstrated that dephosphorylation of Ser259 in CRAF regulated its association to the membrane¹. Sorafenib treatment inhibited the G.S.-induced phosphorylation of Ser259. Thus, as showed in the below figure and Supplementary Fig 2f-g, treatment of cells with sorafenib under G.S (1h and 4h) promoted the accumulation of CRAF in the cell lysates insoluble fraction. Furthermore, CRAF appears not to be degraded via the proteasome, since treatment of cells with proteasome inhibitor MG-132 led to the accumulation of CRAF in control conditions as well as, 1h or 4h after G.S. However, in the presence of sorafenib under G.S. (1h and 4h) CRAF amounts decreased. Thus, the disappearing of CRAF in G.S. condition in the presence of sorafenib is not due to CRAF degradation, but its translocation to the membrane and association to the non-soluble fraction of cell lysates. Interestingly, this is also true for PFKFB2, which becomes

associated to the insoluble fraction of the cell lysates upon G.S.+Sor. This will also explain the decreased amounts of PFKFB2 observed in G.S.+Sor. samples. The mechanism mediating the translocation of PFKFB2 to the membrane is unknown and under investigation.

(a) *SkMel103* cells were subjected to the indicated treatments, G.S. (glucose starvation) Sor. (sorafenib 10 μ M) in the presence or absence of the proteasome inhibitor MG-132. Western blot shows the amounts of the indicated proteins. (b) Western blot showing the indicated proteins in the soluble and insoluble lysate fractions upon the indicated treatments.

Nevertheless, as suggested by the reviewer, we also checked the possible inhibition of p-EIF4E by sorafenib. However, as showed in the below figure, while G.S. promotes the phosphorylation of EIF4E (most likely through the activation of ERK1/2) the amount of p-EIF4E in G.S.+sor-treated cells do not differ from control cells. Thus, despite the obvious contribution of EIF4E inhibition by sorafenib to the final CRAF amounts at 4hrs (short timing), it is unlikely that this mechanism would contribute to CRAF disappearance.

6. Figure 2f: The changes in CRAF and BRAF phosphorylations and the in vitro kinase activities suggest a switch in RAF isoform upon glucose starvation. However, because the phosphorylation of RAF on these sites is not a definitive marker of its activity and because the kinase activity is realised in vitro, the switch in RAF isoform need to be confirmed using siRNA to disrupt BRAF or CRAF and evaluate the effect on ERK phosphorylation.

This is a good suggestion; this data has been added as new Figs. 2f-g. In these experiments the depletion of CRAF did not impede the hyperphosphorylation of ERK1/2 promoted by G.S., however, either depletion of BRAF or both CRAF/BRAF decreased the hyperphosphorylation of ERK1/2 induced by G.S., confirming the participation of BRAF in the mechanism and the switch in RAF isoforms.

7. Figure 4c: It is not clearly indicated which alterations have been evaluated. Did the authors consider only genetic alterations (mutations, copy-number alterations) or included mRNA expression? What is the fraction of melanoma presenting alteration of PFKFB2 in the Firehorse Legacy study used.

The data the reviewer refers to was showed on old supplementary Figs. 3c-3d, new supplementary Figs. 5c-d. As showed in this supplemental figure we considered all genetic alterations including mRNA expression regulation. The type of alteration is indicated in the

legend below the graph, which include: in frame mutations, splice mutations, truncating mutations, missense mutations of unknown significance, missense mutations (putative drivers), amplifications, deep deletions, mRNA upregulation and mRNA downregulation. PFKFB2 was altered (including regulation of mRNA) in 12% of the samples and mutated or amplified in 7% of the samples.

8. Figure 5A : Induction of PFKFB2 phosphorylation by glucose starvation is only seen in two out of four cell lines making it impossible to draw a general conclusion on PFKFB2 phosphorylation induced by glucose starvation.

We understand the reviewer's concern that is why we tried to be cautious in the original version of the manuscript stating "PFKFB2 was constitutively phosphorylated at Ser⁴⁸³ in both NRAS^{Q61} and BRAF^{V600E} mutant cells, however, G.S. induced the further phosphorylation of this residue, in a time-course and cell line-dependent manner". It is true that the phosphorylation of PFKFB2 in response to G.S. it is seen more clearly in certain cell lines. There are cell lines that have a higher basal amount of p-PFKFB2, where the increase of phosphorylation in response to G.S is minimal or hardly noticed. Even within the same cell line, tumor cells are metabolically heterogeneous, according to the genetic alterations and their particular needs at a certain time point (i.e: nutrient availability, cell cycle phase, intracellular ROS amounts...). Thus, the metabolic synchronization of cells in respect to the posttranslational modifications of enzymes involved in the metabolic rewiring is extremely difficult, if not virtually impossible. This situation makes difficult to observe strong responses frequently, however, it does not mean that they are not occurring (i.e G.S-induced phosphorylation can be observed in SKMel103 Fig. 6a-d, SKMel147 Fig. 6b and SKMel28 Fig. 6a-d -> old Fig. 5). Furthermore, there are indirect ways to observe that this is the case. For instance, it is known that Ser483 phosphorylation induces the recruitment and binding of 14-3-3 proteins to p-PFKFB2¹⁴. In Fig 7a-b it can be observed that 14-3-3 is recruited to PFKFB2 immunocomplexes in response to G.S. Moreover, the increased phosphorylation of residue Ser483 can also be observed (including in UACC903 cells) in the identified phosphopeptides (semiquantitative). Nevertheless, we have added new cell lines and experiments subjected to G.S. (figure below and Supplementary Fig 6b) and quantified the p-PFKFB2 amounts in all experiments showed (the new ones and the original ones). We hope that this explanation and the new results would satisfy the reviewer concerns.

(a) SKMel103 and SKMel147 cells were subjected to glucose starvation for 1h in the absence or the presence of increasing concentration of sorafenib. Western blot shows the amount of p-PFKFB2 and PFKFB2 in the soluble cell lysate fraction. GAPDH is shown as loading control. As expected upon G.S. sorafenib decreased the total PFKFB2 protein levels. (b) Melanoma cell lines were subjected to G.S. for the indicated time. Western blot show p-PFKFB2 and PFKFB2 amounts upon treatment. Numbers indicate fold induction of phosphorylation in respect to the control (C).

9. Figure 5E: Although the in vitro phosphorylation of PFKFB2 by BRAF is convincing (figure 6f) the decrease in PFKFB2 phosphorylation with BRAF siRNA is not. The decrease in PFKFB2 phosphorylation parallels the decrease in PFKFB2 total protein. Therefore, the ration p-PFKFB2/ PFKFB2 does not seem to vary upon BRAF inhibition making it impossible to conclude on the effect of BRAF on PFKFB2 phosphorylation ex vivo.

The two points raised above show that this work does not clearly support the claim that there is a connection between glucose starvation, phosphorylation of PFKFB2 and BRAF. Additional evidences are needed to draw this conclusion.

The reviewer is right. We have repeated the experiment, added a new panel on Fig 6e showing the effects of G.S. on PFKFB2 phosphorylation in BRAF depleted cells and a new supplementary Fig 6c showing the same kind of experiments in a patient-derived cell line. However, we have to bear in mind that as observed for BRAF downstream signaling molecules (i.e. ERK1/2), depletion of 90% of BRAF in response to G.S. led to a partially decrease in the hyperactivation of the pathway (p-ERK1/2). These still BRAF acting molecules would be able to regulate PFKFB2. In addition to this, as observed with CRAF, inhibition of RAS-RAF pathway upon G.S. promoted the association of non-phosphorylated PFKFB2 to the insoluble lysate fraction (above figure at point 5 and Supplementary Fig 2g), challenging the interpretation.

Nevertheless, we believe that the connection between G.S., phosphorylation of PFKFB2 and BRAF is strong and solid. First, we show convincing data supporting the switch in the isoform use from CRAF to BRAF after G.S. We also have showed that hyperactivation of the pathway in response to G.S. is NRAS oncogene activation-dependent (upstream RAF). A number of evidence obtained by different techniques in several cell lines, showed that PFKFB2 becomes phosphorylated in response to G.S (WB, pull downs, mass spectrometry), and Fig 7b and 7d show the binding of BRAF to PFKFB2 in cell lysates. In addition to the effects of G.S. on PFKFB2 phosphorylation in BRAF depleted cells showed in new Fig 6e, PFKFB2 phosphorylation is inhibited upon G.S. by a RAS inhibitor (salirasib), RAF inhibitors (sorafenib , regorafenib and CTT (specific pan-RAF inhibitor)) and not by MEK1/2 inhibitors (Trametinib nor U0126) (Fig 6d), pointing out once more to BRAF contribution. Moreover, in vitro experiments have also demonstrated the binding of BRAF to either PFKFB2 or the heterodimer/heterotetramer PFKFB2/3,(Figs 7e and 7h). Finally, we show the in vitro phosphorylation of PFKFB2 by BRAF and not by CRAF, neither ARAF (Fig 7f). Altogether we think that this data strongly supports that BRAF is responsible for PFKFB2 phosphorylation after G.S.

10. Figure 6k: The data presented connect PFKFB2 and PFKFB3 expressions to ERK phosphorylation upon glucose starvation. There is however no explanation in the text and in Figure 8 to explain how PFKFB2 and PFKFB3 can regulate ERK activity upon glucose starvation.

Rather than PFKFB2 and PFKFB3 expression, is their regulation upon G.S what is connected to ERK1/2 phosphorylation. Briefly, it is known that *NRAS* mutant melanoma cells use CRAF instead of BRAF (the isoform used by normal melanocytes) to signal. Among others, this is promoted by the increase in phosphodiesterase activity, which degrades cAMP thereby preventing inhibition of CRAF by PKA¹⁵. G.S would promote the inactivation of CRAF by inducing PKA and/or AKT activity, both involved in the inactivation of CRAF^{15, 16, 17} and the latter also involved in the phosphorylation of BRAF^{S445}¹⁸. *NRAS* mutant cells appear to be dependent on glucose metabolism and are not very flexible using other sources of energy in the absence of glucose. Under G.S. BRAF phosphorylates PFKFB2^{Ser483} which increases the kinase activity of PFKB2 that was found to be forming heterodimers/heterotetramers with PFKFB3, an isoform with an enhance kinase activity. This in turn, helps to sustain glucose metabolism by the production of Fru-2,6bisP an allosteric activator of PFK1, the main regulator of glycolysis leading to the production of Fru-1,6bisP. Few years ago, it was discovered that Fru-1,6-bisP couples glycolytic flux to the activation of RAS, by activating SOS1¹⁹. We have showed that addition of Fru-1,6-bisP under G.S regulate ERK1/2 phosphorylation, and SOS1/2 depletion impedes ERK1/2 hyperactivation by Fru-1,6-bisP under G.S. This data links BRAF activation after G.S. with glycolysis, connecting the glycolytic flux with NRAS activation, which we demonstrated is necessary for ERK1/2 hyperactivation upon G.S.

This point has been addressed accordingly.

Minor points:

Figure 2c : The western blot is not described in the text. apoptosis

This figure has become supplemental Fig. 2a. The western blots have been substituted for new ones showing PARP1 clv-caspase-6 and BAX and cited in the text accordingly

Figure 6b: Was CRAF present in the His-PFKFB2 pull down?

No, we did not detect CRAF in the His PFKFB2 pull down.

Figure 7b : The western blot is not described in the text. Apoptosis

We added new data supporting an AIF-dependent necroptosis promoted by sorafenib either in G.S. or in the presence of 2DG (new Fig. 8c-d). These results have been cited accordingly in the main text.

Figures 6e/h lack control with GST alone or IgG.

The reviewer is right these controls have been added to the Fig. 7 (new Fig 7e and 7h).

Figure 6k: The opposite effects on ERK phosphorylation and ERK expression is striking. How can the authors explain the presence of phosphorylated ERK but the absence of total ERK proteins in some samples?

We are sorry for this presentation of the data. When we check the activated form of ERK1/2 we always do the phospho-antibody first, so the blotting cannot interfere with any previous blotted antibody. In the case of p-ERK1/2, when the signal is too strong, the stripping of the p-ERK1/2 antibody sometimes is no complete and interferes with the total either ERK1/2 or ERK2 antibody. In fact, most of the times, this is an inverted specular image of the degree of ERK1/2 activation. The more activation (p-ERK1/2) the less total ERK1/2 or ERK2 is detected. The blots have been re-stripped and reblotted and improved quality images have been added.

Reviewer #3 (Remarks to the Author):
This is an interesting study from McGrail et al., that shows that NRAS-Q61 mutated melanomas rely on glucose metabolism, and that inhibition of glycolysis through inhibitors or glucose deprivation sensitize these cells to sorafenib treatment. In particular, glucose deprivation of NRAS-Q61 mutated melanomas triggers a switch from CRAF to BRAF signalling, resulting in PFKFB2 phosphorylation and sustained glycolytic activity. A feedback loop hyperactivates the RAS-ERK1/2 pathway, which in turn sensitizes cells to sorafenib treatment.

Some major considerations:

1) The seahorse data (Fig. 3) indicate lower levels of both ECAR and OCR in NRAS-Q61 cells as compared to BRAF-V600E cells. Based on these data, the authors speculate that glucose metabolism is particularly relevant for NRAS-Q61 cells, because it is used in pathways that branch off from glycolysis. It would be important to quantify glucose uptake in both types, e.g. using fluorescent NBD-glucose, to estimate how much glucose both cell types use. To substantiate the claim that glucose is used to generate macromolecule intermediates or reducing power, the authors should perform a tracing experiment using [U-13C] -glucose to show remodeling (and not just dampening) of glucose

metabolism. Preferable, such tracing experiments are performed both in the presence and absence of sorafenib.

This is a good observation. We believed that the uptake of glucose might or might not be very different among the cell lines. What it is different is the dependency of these cells on glucose, not only because of the catabolic use of this molecule but also, because they seem to be less flexible using other fuel resources in the lack of glucose. We have measured the uptake of glucose in four melanoma cell lines (2 *NRAS*-mutated+2 *BRAF*-mutated). Under steady growing conditions in serum free medium, uptake of glucose increases overtime. SKMel103 and SKMel147 did not showed big differences with UACC903 cells. However, in SKMel28 the uptake was higher than in the other cell lines, which correlated with the larger size of these cells compared with the other three cell lines. However, while treatment with sorafenib diminished the uptake of glucose in *BRAF* mutated cells, it increased the uptake of glucose in *NRAS* mutated cells. Notably, when cells were subjected to G.S for 1h and then the medium was reconstituted with glucose (10 mM), *NRAS* mutated cells showed significant higher rates in the uptake of glucose compared to *BRAF* mutant cells, supporting both their glucose dependency and the lack of flexibility using other fuel sources. This new data has been added as a new Fig. 3e-3f.

(a) Graphs showing the uptake of glucose overtime in two *NRAS* mutant and two *BRAF*-mutant cell lines under normal growing conditions (C) and the presence of sorafenib (10µM) (Sor.). (b) Uptake of glucose overtime in two *NRAS* mutant and two *BRAF* mutant cell lines after 1h in G.S

As suggested by the reviewer we performed a tracing experiment using [U-13C6]-glucose to show remodeling (and not just dampening) of glucose metabolism. We performed the experiment in *NRAS*-mutated and *BRAF*-mutated cells under three different conditions: normal growing conditions (C), in the presence of sorafenib (10 µM)(Sor.) and after 1h of G.S. Labeling time was short (40 min) to guarantee the direct labeling and avoid ulterior reuse and the broad distribution of labeled molecules. The results show both the percentage of the different isotopologues and a quantitative measure of the detected molecules. The data suggest that *NRAS* mutant cells (SKMel103) showed a clear glycolytic flux that appears to be greater than in UACC903 cells (*BRAF* mutated). In agreement with the above results *BRAF*-mutant cells appeared to be more sensitive than *NRAS* mutant cells to sorafenib in inhibiting the glycolytic flux. Despite the short labeling period of time, *NRAS* mutant cells and not *BRAF* mutant cells derivate glucose to serine (one carbon pathway) and aspartate (through the pyruvate carboxylase) and most likely to alanine and valine from pyruvate. Nevertheless, a great amount of glucose was also transformed into lactate. Finally, in agreement with the proposed mechanism and the results, according to the absolute amounts of pyruvate, lactate, serine and most likely alanine and valine, *NRAS* mutant cells either sustain and/or increase glycolytic flux after 1h of glucose starvation, while *BRAF* mutant cells decreased the glycolytic flux. These results are showed in the new Fig 4 and new supplementary Fig. 4.

Supplementary Figure 4: Simplified schematic of ¹³C-labeling patterns after the metabolism of [U-¹³C]glucose (40min of labeling) via glycolysis and the TCA cycle. Empty circles, ¹²C; black filled circle, ¹³C; black filled circles with red ring ¹³C derived from pyruvate decarboxylase (PC). Graphs in white background represent the identified isotopologues derivatives of the indicated intermediates. Graphs in grey background show absolute amounts of derivatives (sum of all isotopologues) at the indicated conditions: C (clear blue and clear red)=Basal conditions, Sor (mild blue and mild red)=10μm Sorafenib in complete medium. After 1h G.S. (dark blue and dark red)= labeling after 1h of glucose starvation. Experiment were performed in quadruplicates. * p<0.05, **p<0.01, *** P<0.001, (p-value, unpaired two-sided t-test).

2) From Figure 5, the authors conclude that NRAS-Q61 but not BRAF-V600E melanomas sustain PFK1 activity rates. Are the differences in figure 5f significant? Please indicate. Also, the differences between BRAF and NRAS mutated melanoma types seem to be driven by differences in PFK1 activity under basal conditions (30 versus 20 mU/ug protein, respectively). Upon glucose starvation, the residual activity appears more or less equal (15 mU/ug protein in both types), suggesting that glucose starvation is reducing PFK1 activity to similar levels in both types. Can the authors please comment on this?

The statistical significance between the controls and the treated samples for the different cell lines has been added.

In respect to the differences in PFK1 activity between the *BRAF* and *NRAS* mutated cell lines, we believe that the important issue is how relevant is glucose metabolism for cells, and/or how flexible are cells rewiring metabolism to sustain homeostasis in the absence of glucose. We believe this is a critical issue to explain the dependency on glucose of *NRAS* mutant melanoma cells. Our interpretation is that, in the case of *BRAF* mutant cells, the higher basal activity of PFK1 correlates with the higher ECAR showed compared to *NRAS* mutant cell lines. SKMel28 cell line, these cells are larger in size than the other ones, express the higher amounts of PFK1 among the cell lines tested (Figure below) and show the higher activity of PFK1. The data also suggest that in *BRAF* mutant cells, the amount of glucose consumed does not necessarily reflect the dependency of cells on this fuel source, but the convenience. In fact, *BRAF* mutant cells are more flexible using other fuel resources in the absence of glucose. Upon G.S., they experiment a drop of PFK1 activity up to 50% compensating the lack of glucose with other sources such as glutamine or fatty acids (Fig 3d and Supplemental Fig. 3). In the case of *NRAS* mutated cell lines, the consumption of glucose appears to be crucial for their viability because they are less flexible using other fuel source/s. We believe that is this dependency on glucose, what makes these cells to sustain the functionality of the pathway. This dependency and lack of flexibility agrees with the significant increased uptake of glucose of *NRAS* mutant cells compared with *BRAF* mutant cells after 1h of G.S (figure above and below). It also correlates with the smaller decrease in F2,6BP after G.S. compared with *BRAF* mutant cells (Supplementary Fig 6d or below Figure in point 3 answer), that would help to sustain PFK1 activity. Thus, what it seems to be relevant for *NRAS* mutant cells is to sustain the functionality of the pathway as close as possible to their physiological levels to cover their needs. The similar residual activity of the pathway upon G.S., in *BRAF* and *NRAS* mutant cells might reflect the lack of room response in *NRAS* mutant cells vs. *BRAF* mutant cells or just a coincidence due the number of cell lines analyzed.

Expression of PFK1 in the indicated melanoma cell lines

Uptake of Glucose after 1h G.S. $p < 0.0001$ (****)

3) The authors claim from supplementary Figure 4B that the maintenance of PFK1 activity is accompanied with a sustained production of F2,6-BP and F1,6-BP. However, F2,6-BP levels are lowered by glucose starvation in all cell lines tested. To nonetheless make the claim that F2,6-BP

production is sustained, the authors should include mutant BRAF cells in these experiments as a control.

As suggested by the reviewer, we measured the amounts of both F1,6-BP and F2,6-BP in *BRAF* mutant cells. Results are showed in supplemental Fig. 6d-e. In agreement with the proposed mechanism, *BRAF* mutant cells showed an average 75% decrease in the amounts of F2,6-BP after G.S, while *NRAS* mutant cells decreased only about 25-30% the F2,6-BP amount. In agreement with this, *BRAF* mutant cells showed more than 80% decreased in F1,6-BP production. while *NRAS* mutant cells sustained between 75-80% of F1,6-BP amounts measured in normal conditions.

Graphs show the fold change in the amount of F2,6BP (d) and F1,6BP (e) in response to 15 and 30 minutes of glucose starvation in three BRAF-mutated and three NRAS-mutated cell lines. Western blot (f) shows ERK phosphorylation in response to 15 and 30 minutes of glucose starvation (G. S.)

4) The authors link PFKFB2/3 to the ability of NRAS-Q61 melanomas to induce ERK1/2 phosphorylation under conditions of glucose starvation (Fig 6k). The link between PFKFB2/3 phosphorylation and sorafenib sensitivity is however less convincing. If PFKFB2/3 are indeed key regulators of this process, one would expect that PFKFB2 and/or PFKFB3 overexpression sensitizes cells to sorafenib treatment. The authors should show this.

We are not sure about the reviewer proposed experiment. Our observations indicate that under glucose starvation there is a switch in the RAF isoform use, from CRAF to BRAF. Under this condition cells need to sustain the glycolytic flux for survival. To do so, BRAF phosphorylates PFKFB2/3 complex regulating its kinase activity leading to the production of F2,6-BP, an allosteric regulator of PFK1, which helps to sustain glucose metabolism. All this would foster the production of F1,6-BP that only in G.S., in turn will activate SOS1 the RAS-GEF¹⁹ activating new NRAS mutated molecules that will signal through BRAF. Thus, under this condition sorafenib will block both BRAF>ERK1/2 downstream activation and PFKFB2/3 phosphorylation, (Ser483) contributing to the inhibition of survival signals and the uncoupling of RAS pathway and the glycolytic flux (PFKFB2/3>PFK1), resulting in cell death. Since the contribution of PFKFB2/3 related to BRAF activity appear to be restricted to G.S. conditions and concomitant post-translational modification/s, we are not sure how PFKFB2/3 overexpression will sensitize cells to sorafenib.

1) How is ATP-linked respiration calculated in figure 3a? ATP-linked respirations is defined as the difference between the basal respiration and the proton leak, and appears larger in BRAF-V600E cells compared to NRAS-Q61 cells (judging from the left graphs).

The reviewer is absolutely right, we have corrected this mistake

2) In Figure 4b, the authors zoom in on glycolysis. The reason for this is not well explained in the text. Arguably, many other metabolic and signalling pathways are also critical for the response to glucose starvation, and studying these may identify other key regulators of the response to metabolic stress. Can the authors provide additional explanation to justify their choice?

It is true that many signaling and metabolic pathways are critical for the response to glucose starvation, and metabolic stress, which is a broad concept affecting many faces of cell biology. Here, we observed that the lack of glucose was triggering a different response in the two most frequent molecularly differentiated melanoma tumors (*NRAS*- and *BRAF*-mutated). The fact that was the lack glucose and not other metabolites, insults or drugs the cause of the differential response (metabolic stress) was the first observation. Then, the reversible phenotype (recovery of basal levels of p-ERK and the resistance to sorafenib) by addition glucose and not pyruvate nor glutamine, reinforce the direct effect of glucose in the response and indirectly its metabolization (glycolysis). We also had pursued other pathways and processes in the investigation and make other interesting observations but they are not related to glycolysis and are out of the scope of this manuscript.

3) Several subfigures in Figures 5 and 6 are not correctly referred to in the text, e.g 5f, 5g, 5h and 6d. Please check.

These mistakes have been corrected accordingly

4) In figure 5h, does adding F2,6-BP to cells induce similar phosphorylation of ERK1/2 under glucose deprivation conditions?

This is an interesting question. However, we did no perform this experiment. Phosphate esters are negatively charged and water soluble, so they do not readily penetrate lipid-rich cell membranes. In the case of F1,6-BP although the exact mechanism by which this molecule gets into the cell is unknown, it has been repeatedly published over the years by many groups the exogenous treatment using F1,6BP *in vitro* and *in vivo*^{19, 20, 21, 22, 23}. In addition to this our experiments demonstrate that only in the absence of glucose the production of F1,6-BP induced the hyperphosphorylation of the ERK1/2 pathway through the activation of SOS1/2. Thus, in the case that cells could capture F2,6-BP the effects probably will be only observed in the absence of glucose, a situation where the kinase activity of PFKFB2 is increased and consequently the production of F2,6-BP.

5) Figure 6b/6c: the differential effects of G.S. + Sor treatment on BRAF and NRAS melanomas seem more pronounced on western blot compared to the phosphoproteomics results: on western blot PFKFB2-S483 phosphorylation is completely inhibited in NRAS cells, but hardly affected in BRAF cells. Using phosphoproteomics, peptides containing phosphorylated S483 seem substantially downregulated in both cell lines. Can the authors comment on this apparent difference?

This is an interesting observation. We believe that this is related to the techniques used in both cases. In old Fig 6b (new Fig 7b) His-PFKFB2 was overexpressed and pulled down to

isolate the complexes. The amount of His-PFKFB2 expressed magnified the effect previously described in Fig. 6b and 6c. In old Fig 6b (new Fig 7b) due to the amount of PFKFB2 and the affinity of the phospho-antibody, the lower effects of sorafenib inhibiting PFKFB2 phosphorylation under G.S conditions in *BRAF* mutant cells are exaggerated. On the other hand, phospho-proteomic quantification is semiquantitative. Fig 7c we show the fold induction in the proportion between phosphorylated/unphosphorylated detected peptides.

Reviewer #4 (Remarks to the Author):

This study investigates the response of NRAS mutant melanoma cells to glucose starvation as compared with BRAF mutation. The results show that glucose starvation induced a rapid hyper-phosphorylation of ERK1/2 in NRAS mutant cell lines, while not in BRAF mutant cells. The authors suggested that glucose starvation induced a NRAS mutant-BRAF-ERK-PFKFB2/3-PFK1-NRAS activation feedback loop to support melanoma cell survival. They also showed that glucose starvation or 2-DG sensitizes NRAS mutant cells to sorafenib treatment. It is a interesting study. However, there are many issues for this study. The conclusion of glucose starvation inducing the switch of CRAF to BRAF downstream of NRAS mutant and following feedback loop is mainly based on consumption and hypothesis, lacking experimental validation. There are also many inconsistency in figures, and missing and wrong-labeled information. Overall, the study is premature and conclusion cannot be supported by provided data.

Main issues:

1. It is unclear how glucose deprivation triggers the CRAF/BRAF isoform switch downstream NRAS mutant. The study stated that NRAS mutant activates BRAF, but there is no direct data evidence showing that NRAS mutant binds to BRAF. Fig. 2a pull-down assay used CRAF-binding domain to pull down NRAS, which showed that dramatically increased upon glucose starvation, opposite to the switch claim. BRAF-binding domain should be tested.

This is a very good question and the answer is sustained in previously described mechanisms. It is known that glucose starvation (G.S.) promotes the activation of other relevant pathways linked to metabolism regulation such as PKA^{24, 25}, and PI3K-AKT^{26, 27}. It is also known that PKA and AKT inactivate CRAF through the phosphorylation of residue Ser259^{16, 28} (which increases in *NRAS* mutant melanoma cells after G.S). Additionally, it has been demonstrated that AKT increases BRAF^{Ser445} phosphorylation, which contributes to BRAF activation by elevating basal and consequently RAS-stimulated activity¹⁸. Thus, G.S. will promote CRAF inactivation and simultaneously BRAF activation by the already above described mechanisms. We have added a Supplementary Fig. 2b-c validating this mechanism using both PKA and AKT inhibitors. In addition to this, as suggested by the reviewer we have added a new Fig. 2c showing the immunoprecipitation of CRAF and BRAF at early time points after G.S. This figure not only shows the initial phosphorylation of CRAF at the inactivating residues (Ser259, and Ser289/296/301) and the phosphorylation of BRAF at residue Ser445 but, it also shows the recruitment of NRAS to BRAF and not to CRAF molecules.

In respect to the criticism about the RAS binding domain (RBD) used in the experiment, we wanted to clarify that the purpose of this experiment (Fig. 2a), together with Fig. 2b, is to demonstrate that the activation of NRAS^{Q61} mutant molecules is crucial to sustain the downstream observed effect on ERK1/2 hyperactivation (not the participation of neither CRAF or BRAF). Nevertheless, Ulf R. Rapp lab has documented essential differences between full length proteins BRAF and CRAF with respect to association with Ras, however they also showed that sequence alignment of the regulatory domains of both kinases displayed regions of high homology, particularly in RBD and CRD (Cysteine Rich Domains), concluding that is

unlikely that RBD or CRD are responsible for differences in the Ras binding properties of BRAF and CRAF, blaming on the first 98 aminoacids of BRAF the observed differences²⁹. Thus, we believe that using RBD domains to prove the participation of BRAF or CRAF *in vitro* would not be the appropriate experiment to blame on a particular RAF isoform. Furthermore, the K_D of the binding of H-Ras to CRAF-RBD (21.2 nM) is slightly higher than the binding of H-RAS to BRAF-RBD (11.2 nM)²⁹, meaning that BRAF-RBD will have higher affinity for RAS and the differences could be more pronounced

Sequence alignment of the N-terminal fragments of A, B and CRAF containing RBD and CRD regions²⁹.

2. The study claims that NRAS mutant through activation of BRAF to support melanoma cell growth. in Fig. 2d, could siBRAF mimic Sorafenib cell killing effects in glucose starvation condition?

While BRAF appears to be an important piece in the response to G.S. coupling RAS pathway and the glycolytic flux, sorafenib-mediated cell killing under G.S seems to be associated to a more complex mechanism. As showed in the figure below, BRAF knock down SKMe103 cells did not show a significant increase in cell death under G.S conditions. Our results showed that only sorafenib and regorafenib, two very similar multikinase type II inhibitors, promoted cell death under this condition. Blocking the activation of the RAS pathway using MEK inhibitors (Trametinib or U0126, Fig 1d) or other BRAF inhibitors (CCT) did not promoted cell killing as sorafenib did. We neither observed the effects exerted by sorafenib under G.S when we used other RTK inhibitors targeting the sorafenib inhibited RTKs and not the RAF proteins (Avastin, Axitinib, Lenvatinib). Thus, the cell killing appeared to be specifically associated somehow to the inhibitor/s (combination of targets or mode of action) and not only to the inhibition of BRAF.

Histograms showing the FACS analysis of dead cells (PI + Annexin-EGFP) under the indicated conditions (C=Control Sor Sorafenib treatment(10 μ M) 4h; G.S.=glucose starvation) in control cells and BRAF depleted cells with two different siRNAs. Graphs below show the percentage of death cell under the different conditions. Three first graphs they are duplicate with different Y-axis scales.

3. Fig. 2d, p-BRAF S445 band shows very minor effects in response to glucose starvation, not consistent with p-ERK1/2 levels. Why?

This is an interesting observation. There are considerable differences between the activation of the RAF proteins. In the case of CRAF maximal activity is seen only when Ser338 and Tyr341 are phosphorylated. While BRAF lacks a tyrosine phosphorylation site equivalent to Tyr341 of CRAF, Ser445 of BRAF is equivalent to Ser338 of CRAF. Phosphorylation of Ser445 in BRAF is constitutive, however, Ser445 phosphorylation still contributes to BRAF activation by elevating basal and consequently Ras-stimulated activity³⁰. As above stated, it also has been demonstrated that AKT might increase BRAFSer445 phosphorylation¹⁸. Thus, in our case Ser445 is showed as a surrogate marker of BRAF activation (with its obvious limitations). The differences the reviewer refers to, are mainly due to two different causes. First, there is a technical issue. The affinity of the antibodies is totally different so trying to compare the intensities of two different antibodies in a western blot would be very complex, not only because the quality of the antibody per se, but because of the relative amounts of the target in the extracts in respect to the other proteins of the lysates. Second, and more relevant, is a signal transduction cascade where the signal is amplified downstream the pathway. In this matter, several models have suggested that the RAF concentration in this cascade is significantly lower than MEK1/2 and ERK1/2. Thus, even with similar reagents it would be unlikely to find similar signal in both cases.

Fig. 2C should check caspase 3/6/9 cleavage and PRAP for apoptosis marker. Cell death necroptosis'???

Old Fig. 2c has become supplemental Fig. 2a. Following the reviewer suggestion, we have changed the western blot and added cleaved-caspase 6 and PARP1 to the figure to support apoptosis. We did not detect any changes in p-MLKL and p-RIPK1/3 (necroptosis markers).

4. Fig. 4. Many panels showing the changes of p-PFKFB2 levels upon glucose starvation are very minors, difficulty to conclude that glucose starvation activates PFKFB2 in NRAS mutant cells. Fig 5e, p-PFKFB2 levels change very minors upon siNRAS or siBRAF.

We understand the reviewer's concern. It is true that the phosphorylation of PFKFB2 in response to G.S. it is seen more clearly in certain cell lines. There are cell lines that have a higher basal amount of p-PFKFB2, where the increase of phosphorylation in response to G.S is minimal or hardly noticed. Even within the same cell line, tumor cells are metabolically heterogeneous, according to the genetic alterations and their particular needs at a certain time point (i.e: nutrient availability, cell cycle phase, intracellular ROS amounts...). Thus, the metabolic synchronization of cells in respect to the posttranslational modifications of enzymes involved in the metabolic rewiring is extremely difficult, if not virtually impossible. This situation makes difficult to observe strong responses frequently, however, it does not mean that there are not occurring (i.e G.S-induced phosphorylation of PFKFB2 can be observed in SKMe103 Fig. 6a-d, SKMe147 6b and SKMe28 Fig 6a-d). Furthermore, there are indirect ways to observe that this is the case. For instance, it is known that Ser483 phosphorylation induces the recruitment and binding of 14-3-3 proteins to p-PFKFB2¹⁴. In Fig 7a and 7b it can be observed that 14-3-3 is recruited to PKFKB2 immunocomplexes in response to G.S. Moreover, the increased phosphorylation of residue Ser483 can also be observed (including in UACC903 cells) in the identified phosphopeptides (semiquantitative). Nevertheless, as suggested by the reviewer, we have added new cell lines and experiments subjected to G.S. (figure below) and quantified the p-PKFB2 amounts in all experiments showed (the new ones and the original ones). These results have been added to Supplementary Fig. 6a.

(a) *SKMel103* and *SKMel147* cells were subjected to glucose starvation for 1h in the absence or the presence of increasing concentrations of sorafenib. Western blot shows the amount of p-PFKFB2 and PFKFB2 in the soluble cell lysate fraction. GAPDH is shown as loading control. As expected upon G.S. sorafenib decreased the total PFKFB2 protein levels. (b) Melanoma cell lines were subjected to G.S. for the indicated time. Western blot shows p-PFKFB2 and PFKFB2 amounts upon treatment. Numbers indicate fold induction of phosphorylation in respect to the control (C).

In respect to the levels of pPFKFB2 in old Fig. 5e, (new Fig. 6e) the reviewer is right. We have repeated the experiment and added a new panel on Fig 6e showing the effects of G.S. on PFKFB2 phosphorylation in BRAF depleted cells. Additionally, we repeated this experiment knocking down NRAS and BRAF in MMLN9 patient-derived cells which is showed in supplementary Fig 6c. However, we wanted to explain that knocking down the components of the RAS pathway in a RAS-dependent cell line to study the effects on RAS signaling pathway is not trivial. There are intrinsic difficulties involving the subjects of the study (cells and the pathway). Due to the oncogene addiction of cells, depletions of the pathway components reaching 95-100% are mostly lethal. Additionally, this signaling modules are in charge of collecting and funnel different upstream stimuli that result in a cell biological response. These molecules (modules) are located at different subcellular localizations to timely accomplish the cell requirements. Thus, just a very small fraction of them become activated in response to a certain stimulus or at basal level. For example, up to 3% of cellular CRAF can be found in association with Ras and just between 2-5% participate in response to a certain stimulus^{10, 11, 12}. It is also known that the scaffold protein KSR1 only binds less than 5% of endogenous CRAF¹³, indicating that KSR1 affects only a subset of RAF functions, and RAF members might be present in other protein complexes. Thus, when it comes to knockdown these molecules, depletion of 80-90% of the protein might not be enough to affect the basal state of the pathway or even the response to certain stimulus, since just 3-5% of the total protein pools participate in the response (see the example below). So, to demonstrate the effects of BRAF and NRAS depletion on PFKFB2 phosphorylation might be challenging, because the downregulation levels must allow survival but also compromise the signaling. We hope this explanations and experiments will satisfy the reviewer concerns.

Western blot shows ERK1/2 phosphorylation status in response to glucose deprivation after knockdown of BRAF (left) or CRAF (right) in *SKMel103* cells (*NRAS*-mutated).

5. Fig. 6a, there is no BRAF protein shown in PFKFB2 pull-down protein list, which do not support their direct association. Not detected in MS.

We thank the reviewer comment. It is true that detection of the protein of interest by MS would be a confirmation of the western blot data. However, it is not uncommon that proteins detected by WB cannot be detected by MS. Even though mass spectrometry can detect small amounts of proteins (in the range of 0.2-1 fmol), western blots are generally more sensitive. Besides the protein amount present in a sample, other factors can also influence their detectability by mass spectrometry, such as the ability to produce, upon digestion with trypsin, peptides that are easily ionizable and well-behaved in the chromatographic separation previous to MS. The complexity and dynamic range of the sample are also critical for LC-MS detectability. The fact that the low-abundance of BRAF protein is not detected in the MS analysis, does not rule out at all its presence in the pull-down protein mixture.

Fig. 6d, there is no increase of PFKFB2 and BRAF binding upon glucose starvation compared with control. Constitutive binding, supported by the *in vitro* experiments using recombinant proteins mostly unmodified.

We understand the reviewer concern. It is true that there is a constitutive binding of BRAF to PFKFB2. However, due to the number of BRAF molecules that become involved as part of the mechanism in response to the stimuli (2-5%), it results very difficult to observe differences in total BRAF immunoprecipitations. As above exposed phosphorylation at BRAF^{S445} contributes to BRAF activation, thus, active BRAF molecules must be phosphorylated at Ser445. When we immunoprecipitate phospho-BRAF^{S445} we enrich the population of molecules participating in the mechanism (Fig. 7d). Under these circumstances, we could observe an increased amount of PFKFB2 bound to phospho-BRAF^{S445} after G.S. We are aware of the limitations of the *in vitro* assays, however, the binding and kinase activity assay were performed with an active recombinant human BRAF (according to its activity phosphorylating MEK1-KD). Thus, there is no need for protein modification for the activity of the recombinant BRAF. Furthermore, other active forms of different RAF isoforms (ARAF or CRAF) failed to bind and phosphorylate PFKFB2 and none of the RAF isoforms phosphorylated PFKFB3 (data not showed), supporting the rest of evidence and the proposed mechanism.

Fig. 6h, lacking negative pull-down control.

These controls have been added accordingly

6. Fig. 7a, should also test PFKFB2 levels upon 2-DG treatment.

Fig. 7b should test caspase 3/6/9 cleavage. There is no explaining for what is the mean for BIM, MCL1, BAX levels in the text. Change apoptosis, necroptosis

This is a good observation. The levels of p-PFKFB2 and their quantifications have been added to new Fig. 8a (old Fig. 7a). New western blots have been added supporting AIF-PARP1 dependent necroptosis cell death (Fig. 8c). Supporting this mechanism, Fig 8d shows the inhibition of cell death by Necrostatin-I and not z-VAD-FMK in G.S.+Sor condition.

7. Fig. 8. The proposed model is not strongly supported by provided data, there are too many mechanistic gaps for the feedback loop, lacking experimental data support the connection between PFK1/SOS1/NRAS mutant activation and following pathway.

We understand the reviewer's concern and we agree that we do not provide large amount of experimental data supporting this last part of the model. It has been suggested that F1,6bisP activation of Ras constitutes a key mechanism through which the Warburg effect

might stimulate oncogenic potency¹⁹, which could be particularly relevant in *NRAS* mutated cells due to their dependency on glucose metabolism. The purpose of this final issue within the manuscript was to connect an already known and demonstrated mechanism (conserved from yeast to mammalian cells) (Peeters et al., Nat. Commun. 2017) with our discoveries. We demonstrated that the effect of G.S. on ERK1/2 activation is *NRAS*^{Q61} activation-dependent. Furthermore, *NRAS* mutant cells tend to sustain F1,6-BP concentration under G.S.(supplementary Fig 6e). Since F1,6-BP promotes the activation of *NRAS* molecules through the binding to *SOS1*¹⁹, we wanted to validate the possible participation of this mechanism by confirming the hyperactivation of the pathway by the addition of F1,6-BP under G.S. conditions (mimicking the sustained situation in *NRAS* mutant cells under the same conditions). Of course, we do not exclude the contribution of other possible mechanism/s (i.e. activation of RTKs after G.S.) in the activation of *NRAS*^{Q61} molecules, however we considered important to show possible contribution of this mechanism. In addition to this, we have added new supplementary Fig 6h (Fig. below, left panel) showing that in the presence of F1,6-BP the hyperactivation of ERK1/2 becomes to be detected as early as 15'-30' after G.S. Moreover, in new Fig 6i knockdown of *SOS1/2* avoided the activation of ERK1/2 by F1,6-BP under G.S (Figure below, right panel). However, for obvious reasons, a deeper analysis of this mechanism is out of the scope of this manuscript.

Western blots showing the activation of ERK1/2 after glucose starvation (G.S.) in the presence of F1,6-BP (10mM) (on the left), and by the addition of F1,6-BP in *SOS1/2* depleted cells on the right.

8. The study should specific indicate glucose starvation, instead to use metabolic stress in whole study as metabolic stress can include many different types of stress.

This suggestion has been incorporated into the manuscript

There are also many mis-information:

1. In supplementary figure 1a, “*NRAS*Q61 mut” and “*BRAF*V600E mut” should be switched.

This has been corrected

2. In supplementary figure 1b, could you explain why *DUSP1* mainly expresses in nucleus in *NRAS*Q61 mutant cells and only expresses in cytoplasm in *BRAF*V600E mutant cells.

The reviewer is right, these WBs were wrong labeled. We are sorry for this mistake we have modified the figure and added new blots showing *DUSP1* accordingly.

4. In the main text, there are too much errors in the citations of figures. Some samples are as follows, please check:

cited in main text Supposed version
Supplementary Fig. 1c Supplementary Fig. 1d

Fig.2e Fig.2d
Fig.2f Fig.2e
Fig.2g Fig.2f
Figure 4c after Fig.5b Fig.5c
Fig.4d after Fig.5b Fig.5d
Fig.4e after Fig.5b Fig.5e
Fig. 5c Fig.5f
Fig. 5d Fig.5g
Fig 5e Fig.5h
Second Fig. 6c Fig. 6d

We apologize for these typos. All these errors have been revised and corrected accordingly

5. There is no Fig. 2g.

This typo has been corrected accordingly.

6. please check Fig. 5e and result statement regarding F1,6-BP promoted the phosphorylation

We are sorry for this error. This information has been added to the manuscript

7. Almost all the inhibitors, such as sorafenib, salirasib, regorafenib and salirasib, are not labelled with the dosage in the figure caption.

We are sorry for this error. This information has been added to the manuscript

References

1. Kubicek M, Pacher M, Abraham D, Podar K, Eulitz M, Baccarini M. Dephosphorylation of Ser-259 regulates Raf-1 membrane association. *J Biol Chem* **277**, 7913-7919 (2002).
2. He C, *et al.* MiR-21 mediates sorafenib resistance of hepatocellular carcinoma cells by inhibiting autophagy via the PTEN/Akt pathway. *Oncotarget* **6**, 28867-28881 (2015).
3. Hong TH, *et al.* Application of self-assembly peptides targeting the mitochondria as a novel treatment for sorafenib-resistant hepatocellular carcinoma cells. *Sci Rep* **11**, 874 (2021).
4. Liang C, *et al.* Hypoxia induces sorafenib resistance mediated by autophagy via activating FOXO3a in hepatocellular carcinoma. *Cell Death Dis* **11**, 1017 (2020).
5. Llobet D, *et al.* The multikinase inhibitor Sorafenib induces apoptosis and sensitises endometrial cancer cells to TRAIL by different mechanisms. *Eur J Cancer* **46**, 836-850 (2010).
6. Rahmani M, Nguyen TK, Dent P, Grant S. The multikinase inhibitor sorafenib induces apoptosis in highly imatinib mesylate-resistant bcr/abl+ human leukemia cells in association with signal transducer and activator of transcription 5 inhibition and myeloid cell leukemia-1 down-regulation. *Mol Pharmacol* **72**, 788-795 (2007).
7. Salvi A, *et al.* Effects of miR-193a and sorafenib on hepatocellular carcinoma cells. *Mol Cancer* **12**, 162 (2013).

8. McCubrey JA, *et al.* Roles of the Raf/MEK/ERK pathway in cell growth, malignant transformation and drug resistance. *Biochim Biophys Acta* **1773**, 1263-1284 (2007).
9. Marais R, Light Y, Paterson HF, Mason CS, Marshall CJ. Differential regulation of Raf-1, A-Raf, and B-Raf by oncogenic ras and tyrosine kinases. *J Biol Chem* **272**, 4378-4383 (1997).
10. Hallberg B, Rayter SI, Downward J. Interaction of Ras and Raf in intact mammalian cells upon extracellular stimulation. *J Biol Chem* **269**, 3913-3916 (1994).
11. Matallanas D, *et al.* Raf family kinases: old dogs have learned new tricks. *Genes Cancer* **2**, 232-260 (2011).
12. Grammatikakis N, Lin JH, Grammatikakis A, Tschlis PN, Cochran BH. p50(cdc37) acting in concert with Hsp90 is required for Raf-1 function. *Mol Cell Biol* **19**, 1661-1672 (1999).
13. Joneson T, Fulton JA, Volle DJ, Chaika OV, Bar-Sagi D, Lewis RE. Kinase suppressor of Ras inhibits the activation of extracellular ligand-regulated (ERK) mitogen-activated protein (MAP) kinase by growth factors, activated Ras, and Ras effectors. *J Biol Chem* **273**, 7743-7748 (1998).
14. Pozuelo Rubio M, Peggie M, Wong BH, Morrice N, MacKintosh C. 14-3-3s regulate fructose-2,6-bisphosphate levels by binding to PKB-phosphorylated cardiac fructose-2,6-bisphosphate kinase/phosphatase. *EMBO J* **22**, 3514-3523 (2003).
15. Dumaz N, *et al.* In melanoma, RAS mutations are accompanied by switching signaling from BRAF to CRAF and disrupted cyclic AMP signaling. *Cancer Res* **66**, 9483-9491 (2006).
16. Dhillon AS, Pollock C, Steen H, Shaw PE, Mischak H, Kolch W. Cyclic AMP-dependent kinase regulates Raf-1 kinase mainly by phosphorylation of serine 259. *Mol Cell Biol* **22**, 3237-3246 (2002).
17. Zimmermann S, Moelling K. Phosphorylation and regulation of Raf by Akt (protein kinase B). *Science* **286**, 1741-1744 (1999).
18. Hong SK, Jeong JH, Chan AM, Park JI. AKT upregulates B-Raf Ser445 phosphorylation and ERK1/2 activation in prostate cancer cells in response to androgen depletion. *Exp Cell Res* **319**, 1732-1743 (2013).
19. Peeters K, *et al.* Fructose-1,6-bisphosphate couples glycolytic flux to activation of Ras. *Nat Commun* **8**, 922 (2017).
20. Bickler PE, Kelleher JA. Fructose-1,6-bisphosphate stabilizes brain intracellular calcium during hypoxia in rats. *Stroke* **23**, 1617-1622 (1992).
21. Hassinen IE, *et al.* Mechanism of the effect of exogenous fructose 1,6-bisphosphate on myocardial energy metabolism. *Circulation* **83**, 584-593 (1991).
22. Liu J, Hirai K, Litt L. Fructose-1,6-bisphosphate does not preserve ATP in hypoxic-ischemic neonatal cerebocortical slices. *Brain Res* **1238**, 230-238 (2008).
23. Veras FP, *et al.* Fructose 1,6-bisphosphate, a high-energy intermediate of glycolysis, attenuates experimental arthritis by activating anti-inflammatory adenosinergic pathway. *Sci Rep* **5**, 15171 (2015).
24. Deng Z, *et al.* Selective autophagy of AKAP11 activates cAMP/PKA to fuel mitochondrial metabolism and tumor cell growth. *Proc Natl Acad Sci U S A* **118**, (2021).
25. Depry C, Mehta S, Li R, Zhang J. Visualization of Compartmentalized Kinase Activity Dynamics Using Adaptable BimKARs. *Chem Biol* **22**, 1470-1479 (2015).

26. Izuishi K, Kato K, Ogura T, Kinoshita T, Esumi H. Remarkable tolerance of tumor cells to nutrient deprivation: possible new biochemical target for cancer therapy. *Cancer Res* **60**, 6201-6207 (2000).
27. Salvadori G, Zanardi F, Iannelli F, Lobefaro R, Vernieri C, Longo VD. Fasting-mimicking diet blocks triple-negative breast cancer and cancer stem cell escape. *Cell Metab* **33**, 2247-2259 e2246 (2021).
28. Subramaniam S, *et al.* Insulin-like growth factor 1 inhibits extracellular signal-regulated kinase to promote neuronal survival via the phosphatidylinositol 3-kinase/protein kinase A/c-Raf pathway. *J Neurosci* **25**, 2838-2852 (2005).
29. Fischer A, Hekman M, Kuhlmann J, Rubio I, Wiese S, Rapp UR. B- and C-RAF display essential differences in their binding to Ras: the isotype-specific N terminus of B-RAF facilitates Ras binding. *J Biol Chem* **282**, 26503-26516 (2007).
30. Mason CS, Springer CJ, Cooper RG, Superti-Furga G, Marshall CJ, Marais R. Serine and tyrosine phosphorylations cooperate in Raf-1, but not B-Raf activation. *EMBO J* **18**, 2137-2148 (1999).
31. Bellou S, *et al.* VEGF autoregulates its proliferative and migratory ERK1/2 and p38 cascades by enhancing the expression of DUSP1 and DUSP5 phosphatases in endothelial cells. *Am J Physiol Cell Physiol* **297**, C1477-1489 (2009).
32. Boulding T, *et al.* Differential Roles for DUSP Family Members in Epithelial-to-Mesenchymal Transition and Cancer Stem Cell Regulation in Breast Cancer. *PLoS One* **11**, e0148065 (2016).
33. Fan MK, *et al.* Siglec-15 Promotes Tumor Progression in Osteosarcoma via DUSP1/MAPK Pathway. *Front Oncol* **11**, 710689 (2021).
34. Lin YC, *et al.* DUSP1 expression induced by HDAC1 inhibition mediates gefitinib sensitivity in non-small cell lung cancers. *Clin Cancer Res* **21**, 428-438 (2015).

REVIEWER COMMENTS

Reviewer #2 (Remarks to the Author):

My concerns have been addressed in the revised manuscript.

Reviewer #3 (Remarks to the Author):

The revised manuscript by McGrail et al. is substantially improved. The new data provided strengthen the conclusions and substantiate the proposed mechanism.

The authors have provided glucose uptake measurements as well as performed a glucose tracing experiment, as requested. I agree with the authors that these data support the observed glucose dependency and glucose flexibility. Points regarding these added data:

1. Valine is to the best of my knowledge an essential amino acid, that cannot be synthesized from alanine by human cells. Showing valine in figure 4 is therefore confusing. Please remove valine from figures 4b and 4e, supplementary figure 4 and the manuscript text.
2. In line 243 the authors conclude that glucose metabolism is important to generate reducing power (amongst other things). Technically speaking, the authors do not show this in this experiment (metabolites such as GSH are not shown). I suggest removing the statement about reducing power from the results section and/or mention it in the discussion instead.
3. The data in figure 4e are interesting as they hint towards a more sustained glycolytic flux in NRAS Q61 mutant melanomas. However, most of the changes in metabolite levels are not significant, the data show trends at best and strong conclusions cannot be drawn from these data in my opinion. I would suggest changing the accompanying text (lines 237-241) to state more clearly that this is a trend only.

Regarding major point 2 and 4: I thank the authors for their further explanation of the proposed mechanism (that is now also clearer from the manuscript itself). I completely agree with the authors that the key issue in NRAS Q61 cells is their lack of flexibility, which forces cells to switch RAF isoform use upon conditions of glucose starvation only, in turn sensitizing cells to sorafenib. The previously suggested experiment is indeed not required to prove this mechanism.

Major point 3 and other minor points are sufficiently addressed by the authors in the rebuttal and/or the revised manuscript.

Reviewer #4 (Remarks to the Author):

The authors' responses and revision have satisfied my criticisms. The revision has significantly improved and is appropriate for the journal.

Reviewer #5 (Remarks to the Author):

As referee 1 is not available to assess the authors response, I have reviewed the authors response to this referee, but not to the other referees.

As such I have restricted my review to Figures 1, 2 and 3A since I feel it would be unfair to undertake a full review of the manuscript as a 4th referee at this stage.

Although the authors have gone some way to responding to referee 1, there remain a number of concerns.

1. Although a point not raised by the referee 1, what is really going on with pERK levels in the BRAF mutant lines versus those with NRAs mutations? Is it really that there is 'hyperactivation' of pERK on glucose starvation in the NRAS mutant lines, or is it that the BRAF lines already exhibit elevated basal levels of pERK so that further activation in response to glucose limitation is not

- possible. Since the results from each cell line are presented in different panels, no direct comparison can be made, but looking at Figure 1A, it would seem that in the BRAF mutant cells the basal level of pERK is very high while in the NRAS cells it is low and glucose limitation just brings the level of pERK up to that in the BRAF cells without glucose starvation. The authors therefore need to do one WB where the basal level of pERK and that low glucose is directly compared on the same blot with a similar exposure time and similar loading of total ERK.
2. Suppl. Figure S1A, the levels of Metformin used are very high at 250 μ M. How does this compare to the levels required to affect AMPK activity in other manuscripts?
 3. In supplemental Fig. S1B H.G. and L.G. presumably mean High and Low glucose but this should be defined in the legend.
 4. In Figure S1D, the BRAF mutant line SKmel28 does show a relatively robust activation of pERK in low glucose, in contrast to Figure 1A. Why the variation?
 5. In Supplemental Figure 1D, Lenvatinib and Salirasib both increase basal level of p-ERK so induction doesn't happen. This is potentially very similar to what is happening in the BRAF melanoma lines where the basal level of pERK appears very high. Does this mean that in BRAF melanomas, the pathways inhibited by Lenvatinib and Salirasib are not operating and this is the reason for the difference between the NRAS and BRAF lines? The authors at least should comment on the effects of these drugs.
 6. In Figure S2A the effects on Cleaved Caspase 6 and Bax are not obvious in the SKmel147 cells. I see no effect. The authors might show instead cleaved PARP, since total PARP does decrease. Otherwise, I don't think this western blot is useful as it is.
 7. The disagreement between the authors and Referee 1 regarding whether the rewiring of signaling from CRAF to BRAF is mediated by a change in activity or loss of CRAF protein is largely immaterial to the conclusion.

In the response to referee 1 the authors state that CRAF is not degraded, but rather relocated to the insoluble fraction. However, it is difficult to ascertain the relative amounts of the proteins in each fraction since the WB will be performed using aliquots of fractions that are not normalized to cell number. What happens if cells are just lysed using SDS loading buffer and then analysed with no fractionation/cell extraction?

The authors also use MG132 to block proteasome-mediated degradation, but there are non-proteosomal pathways that degrade proteins, ER stress for example can trigger such degradation. This experiment therefore cannot rule out CRAF degradation.

In addition I suspect it more likely that CRAF is no longer translated under low glucose conditions when global translation is suppressed via phosphorylation of eIF2 α , and translation of factors required to resolve the stress (such as ATF4 and possibly BRAF) is upregulated or maintained. This can easily be tested. If the authors treat cells with the eIF2 α phosphatase inhibitor salubrinal, this will induce translation reprogramming and I expect will silence CRAF leading to rewiring of MAPK signalling via BRAF. This would be a particularly exciting result since it is known that translation reprogramming de-differentiates melanoma and drives invasiveness.

In reality it doesn't matter whether CRAF protein levels are decreased or not, the key result is that BRAF is activated and CRAF activity and/or levels are reduced. But to properly resolve this issue the authors need to do three things:

- a. Check total CRAF levels in cells that are not extracted/fractionated under conditions where CRAF appears to be lost
- b. Test the effects of Salubrinal (or even tunicamycin) on CRAF, BRAF and pERK.
- c. Adjust the text to state that irrespective of whether CRAF is inactivated, relocated to the insoluble fraction, degraded via a non-proteosomal pathway, or silenced by translation reprogramming, signaling is now redirected away from CRAF toward BRAF. Of course if the salubrinal expt works, the authors can definitely state the mechanism underlying the switch from CRAF to BRAF.

8. In Figure 3A the NRAS mutant line MMLN10 appears to behave more like the BRAF mutant lines than the other NRAS lines. It also exhibits a somewhat reduced induction of pERK on glucose

deprivation in Fig. 1A. Is the basal level of ERK higher in this cell line than other NRAS lines? And does this mean that NRAS vs BRAF status is not the sole determinant of the metabolic rewiring that occurs on glucose limitation?

Minor point:

Typographical errors remain in the text eg in legend to Suppl Figure 1 eg nucleous. The manuscript should be carefully reviewed.

RESPONSE TO REVIEWERS' COMMENTS

Reviewer #2 (Remarks to the Author):

My concerns have been addressed in the revised manuscript.

Reviewer #3 (Remarks to the Author):

The revised manuscript by McGrail et al. is substantially improved. The new data provided strengthen the conclusions and substantiate the proposed mechanism.

The authors have provided glucose uptake measurements as well as performed a glucose tracing experiment, as requested. I agree with the authors that these data support the observed glucose dependency and glucose flexibility. Points regarding these added data:

1. Valine is to the best of my knowledge an essential amino acid, that cannot be synthesized from alanine by human cells. Showing valine in figure 4 is therefore confusing. Please remove valine from figures 4b and 4e, supplementary figure 4 and the manuscript text.

We completely agree. The figures and text have been corrected accordingly

2. In line 243 the authors conclude that glucose metabolism is important to generate reducing power (amongst other things). Technically speaking, the authors do not show this in this experiment (metabolites such as GSH are not shown). I suggest removing the statement about reducing power from the results section and/or mention it in the discussion instead.

The suggestion has been addressed

3. The data in figure 4e are interesting as they hint towards a more sustained glycolytic flux in NRAS Q61 mutant melanomas. However, most of the changes in metabolite levels are not significant, the data show trends at best and strong conclusions cannot be drawn from these data in my opinion. I would suggest changing the accompanying text (lines 237-241) to state more clearly that this is a trend only.

As suggested we have softened the previous statement

Regarding major point 2 and 4: I thank the authors for their further explanation of the proposed mechanism (that is now also clearer from the manuscript itself). I completely agree with the authors that the key issue in NRAS Q61 cells is their lack of flexibility, which forces cells to switch RAF isoform use upon conditions of glucose starvation only, in turn sensitizing cells to sorafenib. The previously suggested experiment is indeed not required to prove this mechanism.

Major point 3 and other minor points are sufficiently addressed by the authors in the rebuttal and/or the revised manuscript.

Reviewer #4 (Remarks to the Author):

The authors' responses and revision have satisfied my criticisms. The revision has significantly improved and is appropriate for the journal.

Reviewer #5 (Remarks to the Author):

As referee 1 is not available to assess the authors response, I have reviewed the authors response to this referee, but not to the other referees.

As such I have restricted my review to Figures 1, 2 and 3A since I feel it would be unfair to undertake a full review of the manuscript as a 4th referee at this stage.

Although the authors have gone some way to responding to referee 1, there remain a number of concerns.

1. Although a point not raised by the referee 1, what is really going on with pERK levels in the BRAF mutant lines versus those with NRAs mutations? Is it really that there is 'hyperactivation' of pERK on glucose starvation in the NRAS mutant lines, or is it that the BRAF lines already exhibit elevated basal levels of pERK so that further activation in response to glucose limitation is not possible. Since the results from each cell line are presented in different panels, no direct comparison

can be made, but looking at Figure 1A, it would seem that in the BRAF mutant cells the basal level of pERK is very high while in the NRAS cells it is low and glucose limitation just brings the level of pERK up to that in the BRAF cells without glucose starvation. The authors therefore need to do one WB where the basal level of pERK and that low glucose is directly compared on the same blot with a similar exposure time and similar loading of total ERK.

This is an interesting observation and we thank the reviewer for the comment. Of course, the level of pERK activation has a limit according to the molecules available within the cell and the status of the regulatory mechanisms involved in the pathway. It is also known that mutated BRAF^{V600E} protein is a powerful and potent activator of its downstream effectors independently of the participation of the upstream components of the pathway (RAS). This, hampers many of the regulatory negative feedback mechanisms controlling the activation of the RAS pathway, (i.e. RAF inactivation, Sos1...), which contributes to the elevated amounts of pERK1/2 in resting conditions(1). Thus, it is known that BRAF mutant melanoma cells have higher basal levels of pERK1/2 than NRAS mutant cells, and this, determines the room of response that BRAF mutant cells have to hyperactivate the pathway by other stimuli (2). It is also known, that the amounts of pERK1/2 in BRAF mutant cells is also subjected to regulation (3,4). Thus, even though the amounts of pERK1/2 are very high in BRAF mutant cells compared with cells harboring either, wild type BRAF or NRAS mutations, the amount of pERK1/2 found in a particular BRAF mutant cell line may vary overtime or according to the conditions, which is going to determine its response and capabilities to hyperactivate the pathway. Nevertheless, we have demonstrated that the hyperactivation of ERK1/2 in NRAS mutant cells is NRAS^{Q61} oncogene dependent (Figure 2 manuscript), which is upstream of BRAF^{V600E} signaling. In other words, the mechanism needs the engagement of the RAS molecules, that would ultimately activate wild-type BRAF. This would indicate that most of the signal detected in BRAF^{V600E} mutant cells is RAS independent. However, due to above mentioned regulation of the signal, under certain circumstances these cells could be a bit responsive to simultaneous stimuli. Thus, the hyperactivation of the pathway it can be observed in not saturated systems (NRAS mutant cells and much less frequent or absent in BRAF^{V600E} (as stated in the main text (line 109))), but always in a RAS dependent-manner, which, appears to be critical to understand the differences (and the mechanism) between these two melanoma entities. As suggested by the reviewer, we have added a WB showing the basal level of p-ERK1/2 in BRAF^{V600E} and NRAS^{Q61} mutant melanoma cells.

Figure 1: WB showing the amount of pERK1/2 in basal (C) and glucose starvation conditions (G.S.) in BRAF and NRAS mutated cells .

2. Suppl. Figure S1A, the levels of Metformin used are very high at 250 μ M. How does this compare to the levels required to affect AMPK activity in other manuscripts?

We appreciate the reviewer concern. However, as far as we know metformin is regularly used at mM concentrations. In fact, we and others had published metformin treatments in melanoma and other tumor cells that range from 0.5 to 30-40 mM concentrations of metformin (5-9). In this case, we selected a suboptimal concentration of Metformin (250 μ M) to test the synergic effect in combination with sorafenib.

3. In supplemental Fig. S1B H.G. and L.G. presumably mean High and Low glucose but this should be defined in the legend.

We are sorry for this typo. Following the first review we substituted all the low glucose (LG) by glucose starvation (G.S.) and apparently we did not do it in this figure. The typo was corrected accordingly.

4. In Figure S1D, the BRAF mutant line SKmel28 does show a relatively robust activation of pERK in low glucose, in contrast to Figure 1A. Why the variation?

As mentioned in point 1 , due to the BRAF^{V600E}-induced signal regulation, under certain circumstances these cells, could be a bit responsive to other simultaneous stimuli. Additionally, upon G.S., ERK2 seems to be sometimes accumulated (stabilized??, less degraded, not happening in all treatments) that might also contribute to exacerbate the observation. This, can be better/also observed in the right WB panel of Fig S1D. Nevertheless, the majority of the signal will be coming from oncogenic BRAF^{V600E} independently of RAS, furthermore, as showed later in the manuscript the effect of mutated BRAF^{V600E} on PFKFB2/3 regulation upon metabolic stress, is not the same as the one exerted by BRAF wild type in NRAS mutant cells.

5. In Supplemental Figure 1D, Lenvatinib and Salirasib both increase basal level of p-ERK so induction doesn't happen. This is potentially very similar to what is happening in the BRAF melanoma lines where the basal level of pERK appears very high. Does this mean that in BRAF melanomas, the pathways inhibited by Lenvatinib and Salirasib are not operating and this is the reason for the difference between the NRAS and BRAF lines? The authors at least should comment on the effects of these drugs.

This is a good observation. Lenvatinib and Salirasib are a multikinase inhibitor (RTKs) and an inhibitor of prenylated protein methyltransferase (RAS) respectively, both acting upstream RAF proteins, and probably, affecting negative feed-back loops in *NRAS*^{Q61} mutant cells, which are not working in *BRAF*^{V600E} mutant cells that signal independently of RAS activation. Thus, as the reviewer suggested the pathway molecules targeted by these inhibitors most likely are not operating in *BRAF*^{V600E} mutant cells because they are affecting to BRAF-upstream acting molecules.

6. In Figure S2A the effects on Cleaved Caspase 6 and Bax are not obvious in the SKmel147 cells. I see no effect. The authors might show instead cleaved PARP, since total PARP does decrease. Otherwise, I don't think this western blot is useful as it is.

We understand the reviewer concerns and we agree that the data appear to be less clear in one of the cell lines (SKMel147). The purpose of this figure was to support the cell death FACS analysis showed on the left part of the figure. We have eliminated the Bax data and added the quantification of PARP and Clv.Caspase 6 bands.

7. The disagreement between the authors and Referee 1 regarding whether the rewiring of signaling from CRAF to BRAF is mediated by a change in activity or loss of CRAF protein is largely immaterial to the conclusion.

We completely agree with the reviewer. Not only because of the conclusion, but also because the rewiring occurs in response to G.S. where no changes in the amount of either CRAF or BRAF have been ever observed or showed in the manuscript (This was not well understood by referee 1 in the first version of the manuscript). The re-localization (not disappearance or loss) is observed under G.S., yes, but only in the presence of Sorafenib. In the first reviewed version we showed the mechanism mediating the switch between RAF isoforms (CRAF to BRAF) and why CRAF relocates to the membrane in G.S.+ sorafenib conditions. Briefly, G.S. will promote the activation of PKA and AKT which are going phosphorylate the inhibitory residues S259 and S289/296/301 of CRAF and also S445 residue in BRAF favoring its activity(10-13). This will result in the switch of use isoform (CRAF to BRAF) that will have consequences regulating glucose metabolism in *NRAS* mutant cells. Sorafenib is a multikinase inhibitor that inhibits among other things the phosphorylation of CRAF S259 (Figure 2), otherwise induced by G.S through PKA and AKT activation (Figure S2). As demonstrated 20 years ago by Baccharinis's lab the inhibition of S259 phosphorylation localize CRAF to the membrane (14), this is why under this condition (G.S.+Sorafenib treatment) CRAF relocates to the membrane. Again, this only occurs only in the presence of sorafenib. Thus, we agree that the controversy of CRAF re-location under this condition (G.S.+Sorafenib) is not relevant for the rewiring mechanism induced by G.S.

In the response to referee 1 the authors state that CRAF is not degraded, but rather relocalized to the insoluble fraction. However, it is difficult to ascertain the relative amounts of the proteins in each fraction since the WB will be performed using aliquots of fractions that are not normalized to cell number. What happens if cells are just lysed using SDS loading buffer and then analysed with no fractionation/cell extraction?

We understand the reviewer concerns. However, in these experiments between 75-85% of the soluble protein of same number of cells was loaded in the soluble fraction and the whole lysate pellet was dissolved in SDS buffer and loaded for further analysis of the insoluble fraction. Thus, the experiments were normalized by cell number in all the cases. Variation in CRAF re-localization was observed as early as 1h after G.S.+Sorafenib treatment due to the mechanism above explained, which occurs upon sorafenib treatment in G.S. conditions and not only in response to G.S. In the figure below following the reviewer recommendations extracting the protein samples using an SDS lysis buffer, we show that CRAF is not degraded under G.S.+Sorafenib conditions, including in the presence of salubrinal.

The authors also use MG132 to block proteasome-mediated degradation, but there are non-proteosomal pathways that degrade proteins, ER stress for example can trigger such degradation. This experiment therefore cannot rule out CRAF degradation.

We completely agree with the reviewer that there are other non-proteosomal degradation mechanisms occurring within the cell. However, the purpose of these experiments in the first review was to add some supportive data about the stability of CRAF (not degradation) under these conditions (G.S.+Sorafenib) in addition to the re-localization experiments.

In addition I suspect it more likely that CRAF is no longer translated under low glucose conditions when global translation is suppressed via phosphorylation of eIF2a, and translation of factors required to resolve the stress (such as ATF4 and possibly

BRAF) is upregulated or maintained. This can easily be tested. If the authors treat cells with the eIF2a phosphatase inhibitor salubrinal, this will induce translation reprogramming and I expect will silence CRAF leading to rewiring of MAPK signalling via BRAF. This would be a particularly exciting result since it is known that translation reprogramming de-differentiates melanoma and drives invasiveness.

Again, the rewiring mechanism occurs as early as 30', and the longer period of G.S tested in the manuscript is 4h. In any case, at this time points under G.S. CRAF is not degraded. It only becomes translocated to the membrane upon G.S+ Sorafenib, (which represent the therapeutic suggested approach) and due to the mechanism above explained(14). As showed in Figure below, CRAF proteins are very stable overtime (half-life of CRAF is $t_{1/2} \sim 8-10$ h) and in cells lysed using SDS lysis buffer, CRAF is present in any condition even in the presence of salubrinal. Thus, even if only G.S. would affect immediately to global translation (which is not relevant in this case because we are discussing an effect only observed under G.S.+Sorafenib), the times at which the experiments were performed in the manuscript would not justify the absence of CRAF by this mechanism.

Fig 2: Graph showing the CRAF degradation in SKMel103 cells treated with cycloheximide for the indicated time points. On the right WB showing the amounts of the indicated proteins extracted with SDS lysis buffer under glucose starvation (G.S. 4h) and the indicated treatments Sorafenib (Sor. 10µM) and Salubrinal (Salubr. 25µM).

In reality it doesn't matter whether CRAF protein levels are decreased or not, the key result is that BRAF is activated and CRAF activity and/or levels are reduced. But to properly resolve this issue the authors need to do three things:

- Check total CRAF levels in cells that are not extracted/fractionated under conditions where CRAF appears to be lost
- Test the effects of Salubrinal (or even tunicamycin) on CRAF, BRAF and pERK.
- Adjust the text to state that irrespective of whether CRAF is inactivated, relocated to the insoluble fraction, degraded via a non-proteosomal pathway, or silenced by translation reprogramming, signaling is now redirected away from CRAF toward BRAF. Of course if the salubrinal expt works, the authors can definitely state the mechanism underlying the switch from CRAF to BRAF.

We believe that we have answered these points above.

8. In Figure 3A the NRAS mutant line MMLN10 appears to behave more like the BRAF mutant lines than the other NRAS lines. It also exhibits a somewhat reduced induction of pERK on glucose deprivation in Fig. 1A. Is the basal level of ERK higher in this cell line than other NRAS lines?

The answer is yes. This cell line showed one of the highest if not the highest amounts of p-ERK at basal level among the all NRAS mutated cell that we managed. Melanoma cells usually harbor large numbers of mutations. MMLN10 are patient derived cells very naive. Besides BRAF or NRAS status we do not have further genetic information (exome seq) on these cells. To explain this, on top of NRAS^{Q61R} mutation we have to consider the possible contribution of other mutations and/or expression regulation of genes affecting the RAS pathway activation. Thus, even though generally NRAS mutant cells display lower levels of p-ERK than BRAF mutant cells, there is a gradient of activated ERK among the NRAS mutant cells, that of course, will also have influence in the response.

And does this mean that NRAS vs BRAF status is not the sole determinant of the metabolic rewiring that occurs on glucose limitation?

This is a good question. We completely agree that NRAS and BRAF oncogenes would not be the sole determinant of the response to glucose limitation. What we can tell is that NRAS mutated cells independently of the preferred source of energy for each cell line according to their own molecular features, appear to be less flexible than BRAF mutant cells using other alternative fuel energy sources in the absence of glucose. Is this caused by the oncogene itself? Probably not, however the fact that you acquired an either NRAS or BRAF mutation will determine the selection of other genetic alterations and/or gene expression regulation, that finally will contribute to the observed phenotype in different ways. Nevertheless, one of the critical parts of the described mechanism in response to G.S., is the engagement of BRAF in cells preferentially using the other isoform CRAF, that ultimately will modify important enzymes that regulate glycolysis, generating a feed-back loop that will end up promoting the activation of new RAS proteins (in this case mutated). This part of the mechanism is not working in BRAF mutated cells,

where instead of CRAF, is mutated BRAF the isoform signaling independently of RAS. Furthermore, in Figure 7b we show evidence for the binding of BRAF wild type and not mutated BRAF^{V600E} to PFKFB2 (the glycolytic enzyme regulated by BRAF). This suggests the absence of this regulatory mechanism in BRAF mutated cells, which appears to be critical to minimize the effects of glucose limitations in cells lacking the flexibility to compensate with other fuel sources. Thus, although NRAS oncogene seems not to be sufficient, it seems to be necessary for the observed response.

Minor points

Typographical errors remain in the text eg in legend to Suppl Figure 1 eg nucleous. The manuscript should be carefully reviewed.

This issue has been properly revised.

1. Yao, Z., Torres, N. M., Tao, A., Gao, Y., Luo, L., Li, Q., de Stanchina, E., Abdel-Wahab, O., Solit, D. B., Poulidakos, P. I., and Rosen, N. (2015) BRAF Mutants Evade ERK-Dependent Feedback by Different Mechanisms that Determine Their Sensitivity to Pharmacologic Inhibition. *Cancer Cell* **28**, 370-383
2. Garnett, M. J., Rana, S., Paterson, H., Barford, D., and Marais, R. (2005) Wild-type and mutant B-RAF activate C-RAF through distinct mechanisms involving heterodimerization. *Mol Cell* **20**, 963-969
3. Houben, R., Vetter-Kauczok, C. S., Ortmann, S., Rapp, U. R., Broecker, E. B., and Becker, J. C. (2008) Phospho-ERK staining is a poor indicator of the mutational status of BRAF and NRAS in human melanoma. *J Invest Dermatol* **128**, 2003-2012
4. Gutierrez-Prat, N., Zuberer, H. L., Mangano, L., Karimaddini, Z., Wolf, L., Tyanova, S., Wellinger, L. C., Marbach, D., Griesser, V., Pettazoni, P., Bischoff, J. R., Rohle, D., Palladino, C., and Vivanco, I. (2022) DUSP4 protects BRAF- and NRAS-mutant melanoma from oncogene overdose through modulation of MITF. *Life Sci Alliance* **5**
5. Niehr, F., von Euw, E., Attar, N., Guo, D., Matsunaga, D., Sazegar, H., Ng, C., Glaspy, J. A., Recio, J. A., Lo, R. S., Mischel, P. S., Comin-Anduix, B., and Ribas, A. (2011) Combination therapy with vemurafenib (PLX4032/RG7204) and metformin in melanoma cell lines with distinct driver mutations. *J Transl Med* **9**, 76
6. Pereira, F. V., Melo, A. C. L., Low, J. S., de Castro, I. A., Braga, T. T., Almeida, D. C., Batista de Lima, A. G. U., Hiyane, M. I., Correa-Costa, M., Andrade-Oliveira, V., Origassa, C. S. T., Pereira, R. M., Kaech, S. M., Rodrigues, E. G., and Camara, N. O. S. (2018) Metformin exerts antitumor activity via induction of multiple death pathways in tumor cells and activation of a protective immune response. *Oncotarget* **9**, 25808-25825
7. Tomic, T., Botton, T., Cerezo, M., Robert, G., Luciano, F., Puissant, A., Gounon, P., Allegra, M., Bertolotto, C., Bereder, J. M., Tartare-Deckert, S., Bahadoran, P., Auberger, P., Ballotti, R., and Rocchi, S. (2011) Metformin inhibits melanoma development through autophagy and apoptosis mechanisms. *Cell Death Dis* **2**, e199
8. Tseng, H. W., Li, S. C., and Tsai, K. W. (2019) Metformin Treatment Suppresses Melanoma Cell Growth and Motility Through Modulation of microRNA Expression. *Cancers (Basel)* **11**
9. Klubo-Gwiedzinska, J., Jensen, K., Costello, J., Patel, A., Hoperia, V., Bauer, A., Burman, K. D., Wartofsky, L., and Vasko, V. (2012) Metformin inhibits growth and decreases resistance to anoikis in medullary thyroid cancer cells. *Endocr Relat Cancer* **19**, 447-456

10. Zimmermann, S., and Moelling, K. (1999) Phosphorylation and regulation of Raf by Akt (protein kinase B). *Science* **286**, 1741-1744
11. Hong, S. K., Jeong, J. H., Chan, A. M., and Park, J. I. (2013) AKT upregulates B-Raf Ser445 phosphorylation and ERK1/2 activation in prostate cancer cells in response to androgen depletion. *Exp Cell Res* **319**, 1732-1743
12. Dumaz, N., Hayward, R., Martin, J., Ogilvie, L., Hedley, D., Curtin, J. A., Bastian, B. C., Springer, C., and Marais, R. (2006) In melanoma, RAS mutations are accompanied by switching signaling from BRAF to CRAF and disrupted cyclic AMP signaling. *Cancer Res* **66**, 9483-9491
13. Dhillon, A. S., Pollock, C., Steen, H., Shaw, P. E., Mischak, H., and Kolch, W. (2002) Cyclic AMP-dependent kinase regulates Raf-1 kinase mainly by phosphorylation of serine 259. *Mol Cell Biol* **22**, 3237-3246
14. Kubicek, M., Pacher, M., Abraham, D., Podar, K., Eulitz, M., and Baccharini, M. (2002) Dephosphorylation of Ser-259 regulates Raf-1 membrane association. *J Biol Chem* **277**, 7913-7919

REVIEWERS' COMMENTS

Reviewer #5 (Remarks to the Author):

I believe that in the latest revision the authors have adequately addressed the referees' concerns

RESPONSE TO REVIEWERS' COMMENTS

Reviewer #5 (Remarks to the Author):

I believe that in the latest revision the authors have adequately addressed the referees' concerns

We thank the reviewer for her/his inputs to the manuscript